# Universal structures for adaptation in biochemical reaction networks

Robyn P. Araujo [1] & Lance A. Liotta[2]

At the molecular level, the evolution of life is driven by the generation and diversification of adaptation mechanisms. A universal description of adaptation-capable chemical reaction network (CRN) structures has remained elusive until now, since currently-known criteria for adaptation apply only to a tiny subset of possible CRNs. Here we identify the definitive structural requirements that characterize all adaptation-capable collections of interacting molecules, however large or complex. We show that these network structures implement a form of integral control in which multiple independent integrals can collaborate to confer the capacity for adaptation on specific molecules. Using an algebraic algorithm informed by these findings, we demonstrate the existence of embedded integrals in a variety of biologically important CRNs that have eluded previous methods, and for which adaptation has been observed experimentally. This definitive picture of biological adaptation at the level of intermolecular interactions represents a blueprint for adaptation-capable signaling networks across all domains of life, and for the design of synthetic biosystems.

The capacity for biological systems to adapt to variable and unpredictable conditions, and to maintain certain key survival-requisite properties within tight tolerances, is fundamental to life itself. This ubiquitous property has been studied under a variety of guises, including robust homeostasis[1] and absolute concentration robustness (ACR)[2,3], all of which are special cases of the keystone phenomenon known as robust perfect adaptation (RPA)[4]. RPA encompasses two essential features: a baseline reference signal, or 'setpoint', established by the concentrations of one or more key molecules, and which allows the system to distinguish high/increasing signals from low/decreasing signals; and an actuator signal (the adaptation), which serves as a memory trace for the altered conditions or stimuli to which the system has been exposed over time[4–6]. RPA has been ubiquitously observed at all scales of biological organization from homeostatic control of plasma mineral concentrations[7] to the regulation of cellular signal transduction networks[8]; from sensory adaptation[9] to neuronal excitation regulation[10]; from the orchestration of cellular stress responses[11] to the coordination of chemotaxis in single-celled organisms[12,13], and is

thought to play a critical role in robust patterning during organism development[14,15].

Importantly, RPA corresponds to a special case of a defining problem in classical automatic control – namely, the robust asymptotic tracking of a desired trajectory (the system's setpoint), while rejecting unwanted disturbances. In the 1970s, the landmark studies of Francis and Wonham[16,17] investigated the necessary controller structures to achieve such robust tracking, and established what is now known as the internal model principle (IMP)[18] (Fig. 1). In simple terms, the IMP states that a dynamical system, $\Sigma$, regulated by some exosystem-generated stimulus or disturbance can be decomposed, via a coordinate change if necessary, into two subsystems, $\Sigma_0$ and $\Sigma_{IM}$, where $\Sigma_{IM}$ constitutes an output-driven internal model (Fig. 1a). Only the system output, $O(t)$, can act as an input to this internal model $\Sigma_{IM}$, which cannot be directly regulated by the remainder of the system $\Sigma_o$, nor by the stimulus/disturbance, $I(t)$, generated by the exosystem. In this way the IMP allows a control system to reject exogenous stimuli or disturbances by incorporating within itself a model of the dynamic structure of the stimulus or disturbance. In the face of persistent

[1]School of Mathematical Sciences, Queensland University of Technology, Brisbane, QLD 4000, Australia. [2]Center for Applied Proteomics and Molecular Medicine, George Mason University, Manassas, VA 20110, USA. e-mail: r.araujo@qut.edu.au

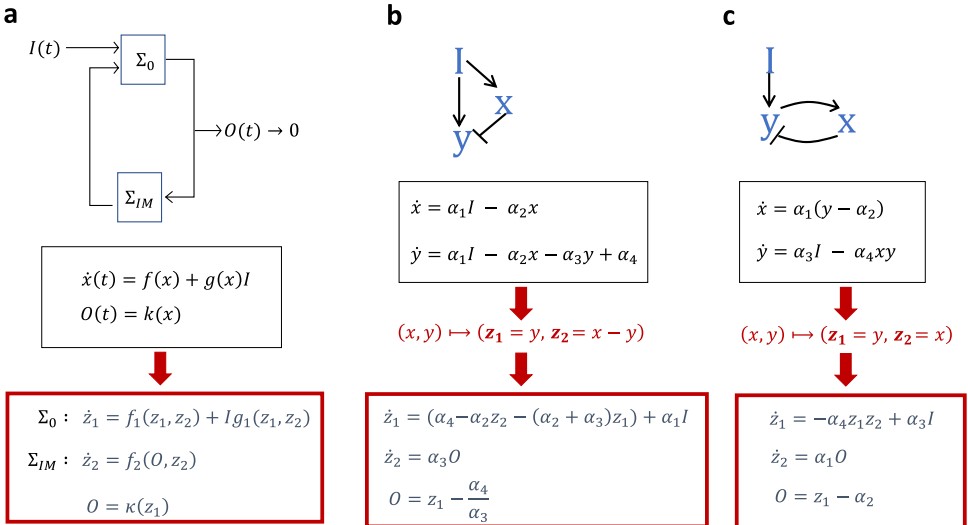

**Fig. 1 | The internal model principle (IMP) and its application to RPA-capable CRNs.** The class of constant disturbances ($I(t) = $ const) is generally the disturbance class of most fundamental interest to the study of biological systems. **a** In order to exhibit RPA (i.e. adapt to constant disturbances), the dynamical system $\Sigma$ should be decomposable, via a coordinate transformation if needed, into an 'output-driven internal model', $\sum_{IM}$ (generating all the constant signals corresponding to solutions of $\dot{z}_2 = 0$), and the remainder of the system, $\sum_0$. The variable $z_2$ thereby computes the integral of the output error. **b** A suitable coordinate change should be able to recast an RPA-capable system into integral feedback form, even if there is no feedback present in the network. As shown, a linear transformation is sufficient to identify an output-driven internal model for this particularly simple incoherent feedforward motif (Balancer module); $y = \alpha_4/\alpha_3$ (setpoint) at steady-state. **c** A model that employs feedback is frequently simpler to recast in 'integral feedback' form, with an output-driven internal model; here $y = \alpha_2$ (setpoint) at steady-state. Note that the reaction rates selected for illustrative purposes in (**b**) or (**c**) cannot be induced, under the law of mass action, by any CRN[20].

(constant) disturbances such as a mutation, an altered external environment, or a new network stimulus, the internal model must produce constant signals and is equivalent to the requirement for integral control[18,19] (Fig. 1b, c). The internal model integrates the adaptation error in some distinguished output variable of the system, constraining it to asymptotically track a fixed value, or setpoint, at steady-state.

Integral control can readily be implemented in engineering design problems by incorporating special integral-computing components into feedback loops. By contrast, signaling networks that evolve in living systems are dynamically assembled via the physical interactions – involving collisions, binding events, and chemical modifications – among discrete entities, or molecules, which must constitute both the signals and their own controllers. Moreover, many collections of biochemical reactions have been identified[2] which impose concentration robustness at steady-state in the absence of any feedback loops or any exosystem-driven disturbances, and in response to alterations in total molecular abundances alone. Robust asymptotic tracking of molecular setpoints in the absence of exogenous network inputs or disturbances constitutes special type of RPA generally referred to as 'absolute concentration robustness' (ACR), corresponding to robustness with respect to the system's initial conditions. How can these complex self-organizing collections of chemical reactions manage to embed the integral-computing structures required for adaptation? Note, in particular, that the reaction rates selected for illustrative purposes in Fig. 1b, c cannot be induced, under the law of mass action, by any chemical reaction network (CRN)[20], highlighting the challenge of identifying the general properties of RPA-capable CRN reaction structures, and universal implementations of integral control via intermolecular interactions.

Until now, integral-computing molecular interactions have only been identified in exceedingly simple chemical reaction networks (CRNs), such as the antithetic integral control motif[6,21], and highly simplified versions of bacterial metabolic circuits and phosphorelays[3], where the requisite integral can be identified via linear change of coordinates. But many adaptation-capable CRNs have been identified (see, for example, Cappelletti et al.[3]), for which no such linear transformation can reveal an adaptation-conferring integral structure,

highlighting the fact that complex nonlinear transformations may be required to detect the presence of integral control for most adaptation-capable CRNs in nature[22]. Although the universal topological principles required for RPA, and for the implementation of integral control, at the level of the network macroscale are now understood in complete generality[4], how these principles could be realized by the intricate intermolecular interactions which comprise CRN structures at the network microscale has been entirely unclear. Previous attempts to identify RPA-capable CRNs, and in some cases, their underlying integral control strategies, have only provided partial answers for special cases[2,3,18,23–25].

Here we identify the universal principles by which all possible instances of RPA-capable CRNs – in all living systems on Earth, as well as in synthetic biology – construct internal models[16–18,25] of any possible stimulus or disturbance, or change in total molecular abundances, thereby allowing the CRN to implement integral control.

As we will show in the sections to follow, a mathematical transformation may always be applied to the reaction rates of any RPA-capable CRN to produce a special two-variable invariant called an 'RPA polynomial'. This distinguished algebraic invariant encodes the robust asymptotic tracking of a molecular setpoint, no matter how complex or intricate the intermolecular interactions, nor how vast the network of interacting molecules. Unlike the nonlinear coordinate transformations invoked in the standard IMP, where a global coordinate transformation (unique to each RPA-capable network) is required to identify a single internal model, the nonlinear transformation we identify here has a special almost-linear structure (and, in special cases, exactly linear) and is decomposable into a topologically organized collection of linear integral controllers, each with its own independent internal model. In this way, we are able to identify the fundamental building blocks of all possible RPA-capable CRNs, thereby revealing definitive structural requirements that characterize all adaptation-capable collections of interacting molecules.

Importantly, these universal adaptation-promoting structural requirements lead to a definitive and well-defined algorithmic test for RPA within networks of chemical reactions. We provide code in the open-source software *Singular* (www.Singular.uni-kl.de) to implement

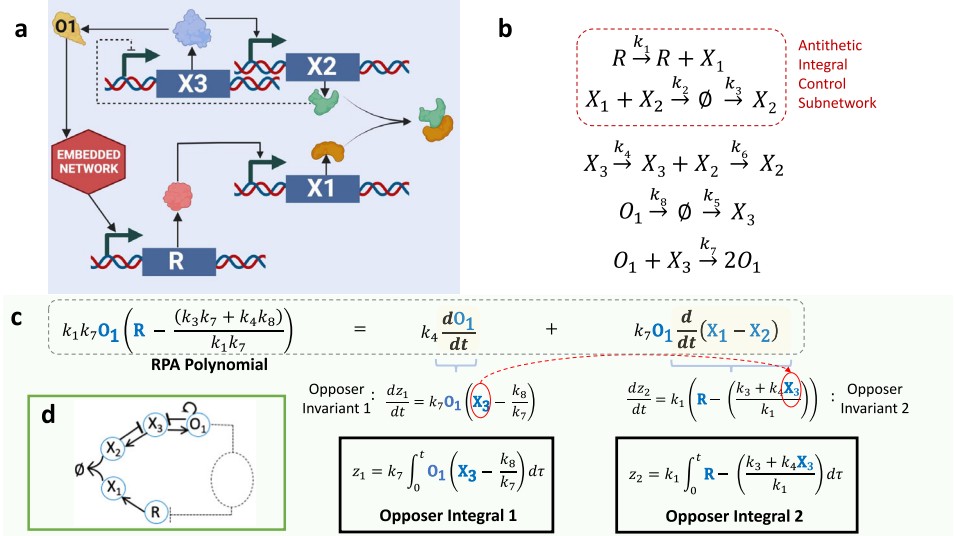

**Fig. 2 | A CRN containing the antithetic integral control motif as a subnetwork.**
**a** Closed-loop system for an antithetic integral controller ($X1$, $X2$), along with an interconnected auxiliary controller ($X3$, $O1$). Created with BioRender.com.
**b** Chemical reactions for the CRN. **c** The CRN implements integral control via two independent internal models, corresponding to two independent polynomial invariants, each obtained by a linear change of coordinates. These are combined nonlinearly (through the concatenating monomial $O1$ applied to invariant 2), to obtain the RPA polynomial, which reveals the setpoint of the molecule $R$.
**d** Topologically, the CRN is an Opposer module; the controller architecture is a two-node opposing set (see SI, Section S4.2.2).

this test for all examples considered here and in our Supplementary Information (SI). This code can be tailored to the study of any CRN.

## Results

### Structural principles for all adaptation-capable CRNs

As a prelude to presenting the universal structural principles by which any CRN can orchestrate a robustly adaptive response (see also SI Section S1), we first briefly describe two simple examples that have eluded all previous systematic methods[2,3,18,23,26,27] to detect RPA, and the presence of integral control, and which exemplify the essential structural principles that are common to all RPA-capable CRNs.

First, we consider an RPA-promoting CRN (Fig. 2a) known as *antithetic integral control* (Fig. 2b)[6,21] - a controller structure that has been identified in the form of sigma/anti-sigma factors in a range of bacterial strains, including E. coli and Salmonella[21], and has also been implemented in synthetic networks[6]. In the simplest possible version of this control mechanism (Fig. 2b, highlighted reactions), two proteins $X_1$ and $X_2$ bind with very high affinity (i.e. irreversibly, thus annihilating each other). One of these proteins ($X_1$) is synthesized at a rate proportional to the concentration of a transcription factor ($R$), while the other protein ($X_2$) is constitutively produced at a constant rate. From the law of mass action, whereby reaction rates are proportional to the concentrations of their reactant molecules, this simple scheme produces the two reaction rates

$$\dot{X}_1 = k_1 R - k_2 X_1 X_2, \tag{1}$$

$$\dot{X}_2 = k_3 - k_2 X_1 X_2. \tag{2}$$

A linear change of coordinates, $\dot{z} = \dot{X}_1 - \dot{X}_2 = k_1 R - k_3$, suffices to identify an internal model, with integral variable $z = k_1 \int_{t_0}^{t} (R(\tau) - \frac{k_1}{k_3}) d\tau$, for any possible persistent disturbance to the system, thereby establishing the capacity for RPA in the molecule $R$ with setpoint $k_1/k_3$ (see Cappelletti et al.[3]). But suppose that a modification to the CRN is introduced during evolution whereby the production of $X_2$ is no longer independent of other signaling activity, but is now under the control of another network protein $X_3$, a transcription factor, as depicted in Fig. 2a, b. The reaction rate for $X_2$ now becomes

$$\dot{X}_2 = k_4 X_3 + k_3 - k_2 X_1 X_2. \tag{3}$$

This perturbed controller structure can no longer impose RPA on $R$ unless $X_3$ participates in additional regulatory interactions, whose structure satisfies very strict constraints (see SI Sections S1.5 and S4). In Fig. 2a, b we provide an example of a suitable auxiliary controller structure for $X_3$, which now includes an additional protein $O_1$. In Fig. 2c and SI Section S4.4.2 we demonstrate that for this expanded CRN, one internal model with integral variable $z_1 = O_1 = k_7 \int_{t_0}^{t} O_1(\tau)(X_3(\tau) - \frac{k_8}{k_7}) d\tau$ imposes RPA on $X_3$, provided that $O_1$ maintains a non-zero concentration – a concept known as constrained integral control[28]. Having thus imposed a steady state value (setpoint) of $k_8/k_7$ on $X_3$, a second internal model, with an integral variable $z_2 = X_1 - X_2 = k_1 \int_{t_0}^{t} (R(\tau) - (\frac{k_3 + k_4 X_3(\tau)}{k_1})) d\tau$, can now impose RPA on $R$. In this way, two internal models - or polynomial invariants (see Fig. 2c) - may be defined, each obtained through (at most) a linear coordinate change. From these two separate polynomial invariants, an 'RPA polynomial' of the form

$$\rho_1 = k_1 k_7 O_1 \left( R - \left( \frac{k_3 k_7 + k_4 k_8}{k_1 k_7} \right) \right) \tag{4}$$

can now be constructed through nonlinear combination, via 'concatenating monomials' (Fig. 2c, and SI Sections S3 and S4), from which it is clear that the network has the capacity for RPA in $R$ with setpoint $(k_3 k_7 + k_4 k_8)/k_1 k_7$.

We demonstrate (SI Sections S2 and S4.4.2) that the CRN depicted in Fig. 2 conforms to the topological principles of an Opposer Module[4], see Fig. 2d. More specifically, the controller structure of this CRN exhibits the special architecture known as a two-node opposing set (see Araujo et al.[4]). Each linear combination of CRN reaction rates that produces an opposer invariant, corresponding to an internal model, thereby constructs an independent opposer integral (see Fig. 2c). The two separate integrals together confer RPA on the sensor molecule R, and ultimately, the entire embedded network (SI Section S4.4.2). This special topological structure, with three feedback loops, and two

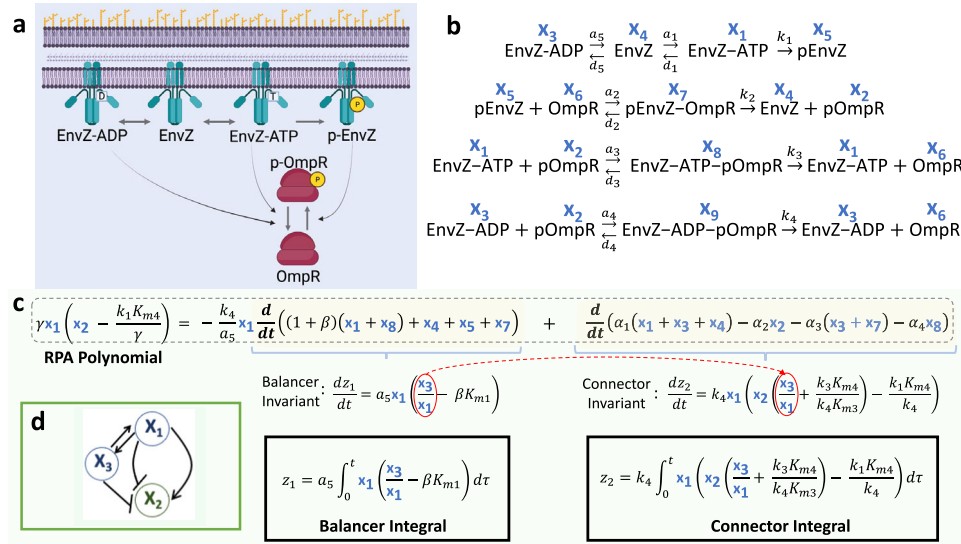

**Fig. 3 | The EnvZ-OmpR bacterial two-component system employing both ATP and ADP as cofactors, as originally proposed by Shinar and Feinberg[2].** **a** Schematic of the EnvZ-OmpR phosphorelay system. Created with BioRender.com. **b** The corresponding CRN. **c** The CRN implements integral control via the computation of two independent polynomial invariants – a Balancer invariant, and a Connector invariant – each obtained by a linear coordinate change. These are combined nonlinearly through the concatenating monomial $x_1$ (EnvZ-ATP) applied to the balancer invariant. The resulting RPA polynomial thereby reveals the setpoint of the molecule pOmpR. Code provided for this calculation in SI (Section S5); Parameters: $K_{mi} = (d_i + k_i)/a_i$, $\beta = d_5/a_5$, $\gamma = k_3(K_{m4}/K_{m3}) + k_4\beta K_{m1}$. **d** Topologically, this CRN is a Balancer module (see SI, Section S3.2).

independent opposer invariants, is directly constructed by the CRN's 'deficiency'[29] of three (see SI Section S4.4.2 for a detailed analysis). Deficiency is a key integer invariant associated with a CRN[29], as we discuss in greater detail in the sections to follow (see also SI Sections S1.2 and S1.3), whose fundamental consequences for the implementation of integral control in CRNs are identified in the present study.

Second, we consider a CRN for the EnvZ-OmpR osmoregulation system in E·coli (Fig. 3). A simplified model of this network with a deficiency[29] of one was first analyzed by Shinar and Feinberg[2], and could be shown to exhibit absolute concentration robustness (ACR) – a type of RPA (SI Section S1.4) – by the Shinar-Feinberg theorem[2]. Crucially, the Shinar-Feinberg theorem applies only to deficiency-one CRNs, which severely limits its ability to detect ACR (and thus RPA) in most molecular networks of biological interest, since the deficiencies of known genome-scale signaling networks (e.g. in metabolism) in even the simplest organisms frequently exceed one hundred[26]. In Fig. 3a, b we consider the more detailed version of the EnvZ-OmpR CRN, which eludes the Shinar-Feinberg theorem, having a deficiency of two, but which can be shown to exhibit RPA (ACR) in the phosphoform, pOmpR (Supplementary Materials of Shinar and Feinberg[2]).

The CRN in Fig. 3 conforms to the topological principles of an RPA-conferring Balancer Module (see SI Section S3.2), with three (incoherent) parallel pathways governing the interconversion between phosphorylated and unphosphorylated OmpR; this fundamental structure is directly controlled by the CRN's deficiency of two (see SI Sections S3.2, S4.4.1). All Balancer modules are characterized by a collection of parallel pathways emanating from a diverter node (here, EnvZ-ATP) and culminating in a connector node (here, pOmpR)[4]. One or more balancer nodes (here, EnvZ-ADP) may be embedded within the parallel pathways[4]. In Fig. 3c we identify an integral variable $z_1 = a_5 \int_{t_0}^{t} EnvZ-ATP(\tau)(\frac{EnvZ-ATP_{(\tau)}}{EnvZ-ATP_{(\tau)}} - \beta K_{m1})d\tau$ that first confers RPA on the concentration ratio EnvZ-ATP/EnvZ-ADP – a key invariant known as a balancer invariant (see Araujo et al.[4]). Having established a setpoint of $\beta K_{m1}$ for this balancer invariant, a second integral variable, $z_2$, constructs a connector invariant that imposes RPA on pOmpR (see Fig. 3c). Crucially, both the balancer invariant and the connector invariant are obtained via linear combinations of the CRN's mass action equations. These two polynomial invariants together produce an RPA polynomial of the form

$$\rho_2 = \gamma.EnvZ - ATP\left(pOmpR - \frac{k_1 K_{m4}}{\gamma}\right), \tag{5}$$

through nonlinear combination via concatenating monomials (see Fig. 3c, and SI Section S3 and S4), thereby explicitly highlighting the CRN's capacity for RPA in pOmpR, with setpoint $\frac{k_1 K_{m4}}{\gamma}$.

Our transformative step is to prove that all RPA-capable CRNs, regardless of size, complexity, or deficiency[2,29], necessarily conform to the general principles encapsulated by these two illustrative examples (see SI Section S4). In fact, all RPA-capable CRNs are characterized by a topological hierarchy of polynomial invariants – a key phenomenon foreshadowed by our earlier framework[4], before general principles were known as to how these invariants might be achieved by CRN graph structures. Here we show that each such subsidiary polynomial invariant is necessarily identifiable via a linear combination of the CRN's rate equations, and corresponds directly to a topological feature (a balancer node, a connector node, or an opposer node) of the overarching network structure. We further show that the presence of these key invariants in the row span of the system (see SI Sections S3 and S4) is fundamentally controlled by the deficiency of the CRN. Moreover, CRN deficiency is generally distributed across multiple algebraically independent subnetworks, which correspond to a decomposition of the reactions into topological modules (see SI Section S4). Thus, the integral-computing properties of CRNs at the network microscale presented here, together with the topological principles that are known to hold at the network macroscale for all RPA-capable networks[4], constitute definitive design criteria that unify all possible RPA-capable CRNs.

There are two distinct but interrelated components to this central result, which we delineate in turn in the sections to follow: (i) We identify the universal algebraic condition that is satisfied by all RPA-capable CRNs, and which is codified in what we call the Two-Variable Kinetic Pairing Theorem (SI Section S1.5); (ii) We show that this algebraic condition always admits a decomposition into a collection of

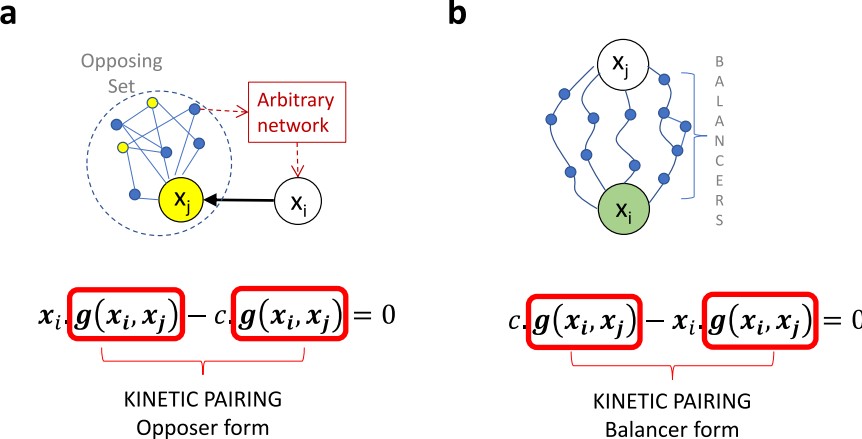

$$x_i \cdot \boxed{g(x_i, x_j)} - c \cdot \boxed{g(x_i, x_j)} = 0$$

KINETIC PAIRING
Opposer form

$$c \cdot \boxed{g(x_i, x_j)} - x_i \cdot \boxed{g(x_i, x_j)} = 0$$

KINETIC PAIRING
Balancer form

**Fig. 4 | The principle of Kinetic Pairing and its topological consequences. a** An Opposer module has a feedback structure, and contains either a single opposer node (not shown) or an organized collection of opposer nodes (known as an 'opposing set', as shown), embedded into the feedback segment of the circuit. Opposer nodes are indicated in yellow. The regulator of this feedback segment, $x_i$, is the sensor molecule for the controlled network, and thus regulates $x_j$ (directly, or indirectly). RPA is achieved at $x_i$ when the upregulating and dowregulating

contributions of $x_j$ to its own reaction rate are exactly matched (paired) through the pairing function, $g(x_i, x_j)$. **b** A Balancer module has a feedforward structure, whereby two or more distinct pathways connect the diverter node, $x_j$, to the downstream connector node, $x_i$ (indicated in green). RPA is achieved at $x_i$ when the upregulating and downregulating contributions of $x_j$ to the reaction rate for $x_i$ are exactly matched via the pairing function $g(x_i, x_j)$.

---

linear problems. The discovery of this previously unrecognized fundamental structure of RPA-capable CRNs exploits the fact that CRNs reactions can generally be partitioned into algebraically independent subsets, each with its own linearly independent stoichiometric subspace (SI Section S1.2), and each with its own deficiency (SI Section S1.3) that governs the formation of RPA-promoting algebraic invariants within the associated subnetwork. Together, these two mathematical results reveal an integral control implementation that holds for all possible RPA-capable CRNs, however large, complex or nonlinear in their dynamics, and whatever their deficiency.

**Adaptation relies on a CRN design strategy called *Kinetic Pairing***
First, we prove (see Theorem 1, SI Section S1.5) that for all RPA-capable CRNs, with interacting molecules $x_1, \ldots, x_n$, and corresponding mass-action rate equations $f_1, \ldots f_n$, there always exist polynomials $\{h_1, \ldots, h_n\} \subset \mathbb{R}[x_1, \ldots, x_n]$ such that

$$h_1 f_1 + \ldots + h_n f_n = g(x_i, x_j)(x_i - c) = \rho, \qquad (6)$$

where $\rho = g(x_i, x_j)(x_i - c)$, in its lowest order form, is the RPA polynomial of the CRN, $x_i$ is any RPA-capable variable of the CRN, and $x_j$ is any variable that does not exhibit RPA (i.e., an actuator variable, or a molecule regulated by an actuator variable). In this context 'variable' has quite a specific meaning, which we define carefully in SI Section S1.5 (Definition 1). The system setpoint, $c$, is a rational function of biochemical parameters (see also Remark 3 in SI Section S1.5).

From this new mathematical vantage point, we can now recognize that the special structure of the RPA polynomial specified by Theorem 1, being a function of exactly two variables, imposes fundamental limitations on the flow of biochemical information through RPA-capable CRNs, and encodes the cardinal principle that we call *kinetic pairing* (see Fig. 4 and SI Section S2). In particular, the functional form of $\rho$ suggests two possible topological interpretations of Theorem 1, depending on whether the RPA-capable variable, $x_i$, is a regulating variable (for $x_j$) or a regulated variable (by $x_j$). As depicted schematically in Fig. 4a, if $x_i$ regulates $x_j$ (the non-RPA-capable variable), then the upregulating and downregulating contributions of $x_j$ to its own reaction rate must be precisely matched, or paired, via the pairing function $g$. At steady state, this form of $\rho$ satisfies the condition $\frac{\partial \rho}{\partial x_j} = 0$ referred to in Araujo et al.[4] as opposer kinetics, and must therefore be embedded in an overarching

feedback loop that gives rise to the topological structure of an Opposer module. Our previous exhaustive analysis on Opposer module topologies has established that the feedback segment of such modules may contain multiple opposer nodes, each with its own opposer kinetics, and each contributing to a collection of embedded interlinked feedback loops known as an opposing set (Fig. 4a).

If, on the other hand, $x_i$ is regulated by $x_j$, then the upregulating and downregulating contributions of $x_j$ to the reaction rate for $x_i$ must likewise be precisely paired via $g$.

As illustrated in Fig. 4b, the pairing function naturally induces a Balancer topology[4] on the CRN in this case (see SI Section S2), with $x_j$ performing a diverter function[4]. For this topological structure, the steady-state condition $\frac{\partial \rho}{\partial x_j} = 0$ corresponds to connector kinetics[4], provided that additional constraints (referred to as balancer kinetics in Araujo et al.[4]) can be satisfied for the reactions embedded into any parallel pathways linking $x_j$ to $x_i$.

Theorem 1 holds for any choice of RPA-capable variable, $x_i$, and non-RPA-capable variable, $x_j$, in the CRN, even if the two variables contribute to distinct independent CRN subnetworks (SI Section S1.3) corresponding to distinct topological modules. Algorithmically, the choice of two variables within a single independent subnetwork, corresponding to a single topological module, simplifies the elimination polynomials $h_1, \ldots, h_n$ in Eq. 6 (SI Section S1.6 for a fully analyzed example). In this context, we recognize that Theorem 1 identifies an independent RPA polynomial specific to the topological module in question and that there exists a natural choice of $x_j$ with respect to the topological structure of the module: a diverter variable (in the case of a Balancer module), or an opposer variable (in the case of an Opposer module). For such a judicious choice of $x_j$, the pairing function $g$ is frequently zero-order in $x_i$, except in the special case of an autoregulatory role for $x_i$. In the case of the antithetic integral control motif[21], the pairing function is zero-order in both $x_i$ and $x_j$, giving rise to unconstrained integral control[3,28].

**RPA-permissive topological features are encoded by CRN deficiency**
Second, we consider a decomposition of the nonlinear algebraic condition for RPA (Eq. 6) into its component linear contributions, to reveal the general mechanism through which kinetic pairing is transacted in CRNs. Indeed, by identifying the connection between the deficiency of

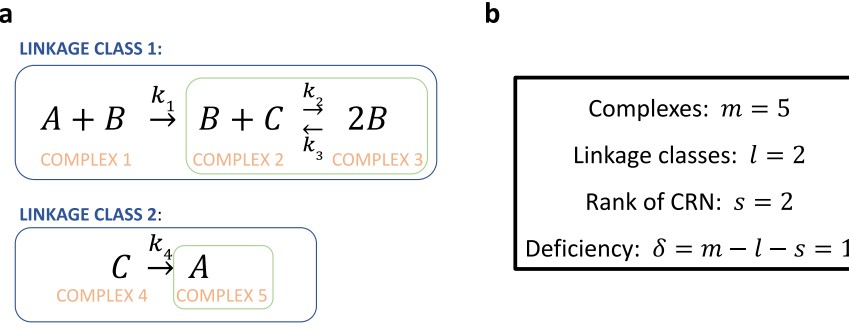

**Fig. 5 | Deficiency is a key integer invariant for a CRN. a** The linkage classes of a CRN are the connected components of the CRN's graph. The complexes are the vertices of the graph, while the reactions are the directed edges. Strong-linkage classes are the maximal strongly-connected subgraphs of the CRN. A terminal strong-linkage class (noted in green) is one in which no complex reacts to a complex in a different strong linkage class. Complexes belonging to terminal strong- linkage classes (complexes 2, 3, and 5 in this case) are terminal complexes; all other complexes are non-terminal complexes. See SI Section S1.2 for a complete technical overview. **b** Deficiency is calculated from the number of complexes, linkage classes and the rank of the CRN, as shown. The rank of the CRN is the number of linearly independent reactions, ie. the dimension of the stoichiometric subspace of the CRN (see SI Section S1.2).

an RPA-capable CRN and the presence of feedback loops and/or feedforward segments (SI Section S1.2), we prove that the RPA polynomial of a CRN can always be decomposed into a collection of subsidiary polynomial invariants, each corresponding to a component of the CRN's topological structure, and each residing in the rowspan of the CRN's reaction rates (SI Sections S3, S4). In other words, each such subsidiary invariant may be obtained via a linear transformation of the system's reaction equations, $f_1, \ldots f_n$.

The deficiency of a CRN (Fig. 5) is a non-negative integer that encapsulates the extent to which the individual reactions of the CRN are linearly independent given their distribution into linkage classes[29] (see SI Sections S1.2 and S1.3 for a detailed exposition of this fundamental concept). With the exception of the trivial RPA-capable CRN consisting only of an isolated connector node, which has a deficiency of zero (see SI Section S4.4.2), all (non-trivial) RPA-capable CRNs require a deficiency of at least one. The Shinar-Feinberg Theorem[2] (see Theorem 2 in SI) pertains to CRNs with a deficiency of exactly one, and states that any such CRN containing two distinct complexes that differ in a single species, $S$, and admitting a steady-state in the positive orthant, necessarily exhibit ACR (and therefore RPA) in the species $S$. This key theorem follows from Shinar and Feinberg's more general result that the steady-state ratio of any two monomials associated to non-terminal complexes (see Fig. 5) is independent of the system's initial conditions[2]. We show that Shinar and Feinberg's arguments may be extended to prove even stronger results (SI Theorems 3 and 4) – namely, that all deficiency-zero and deficiency-one CRNs contain binomials in the rowspan of their reaction rates, of the form $\alpha_1 \psi_i(\boldsymbol{x}) - \alpha_2 \psi_j(\boldsymbol{x})$, where $\psi_i(\boldsymbol{x})$ and $\psi_j(\boldsymbol{x})$ are any two mass-action monomials corresponding to non-terminal complexes (of a deficiency-one CRN), or complexes of a single terminal strong linkage class (of a deficiency-zero CRN), and where $(\alpha_1, \alpha_2) \in \mathbb{R}^2$ is a pair of rational functions of the CRN rate constants. In other words, there exists some $(h_1, \ldots, h_n) \in \mathbb{R}^n$ such that $h_1 f_1 + \ldots + h_n f_n = \alpha_1 \psi_i(\boldsymbol{x}) - \alpha_2 \psi_j(\boldsymbol{x})$.

Crucially, we extend the mathematical framework for Theorems 3 and 4 (on low-deficiency CRNs, $\delta \leq 1$) to a general method for identifying rowspan polynomials in CRNs of arbitrary deficiency ($\delta > 1$). In fact, CRNs can generally be decomposed into a set of algebraically-independent subnetworks (SI Section S1.3), which correspond topologically to independent RPA-capable-modules, each with its own independent RPA polynomial. The deficiency that characterizes each independent subnetwork, and the associated topological module, governs how RPA-relevant subsidiary polynomial invariants are constructed within the rowspan of the CRN. In Section S4.4, we demonstrate these fundamental principles through detailed analyzes of several RPA-capable CRNs with $\delta > 1$ for a single module (see also Fig. 6).

Our method for analyzing these examples makes clear that although an RPA polynomial associated to an RPA-capable CRN generally requires a nonlinear transformation of the reaction equations (Eq. 6), there always exist linear transformations that can extract the special polynomial building blocks, each corresponding one-to-one with a topological feature of the overarching network structure, from the CRN's reaction equations. In particular, all RPA-capable CRNs of Balancer type contain a connector polynomial, corresponding to the connector node, and one or more balancer polynomials, corresponding to balancer node(s), in their rowspans. CRNs of the Opposer type, on the other hand, contain one or more opposer polynomials (corresponding to opposer node(s)) in their rowspans.

## Integral control and the 'passing' of invariants

Until now strategies for identifying an internal model, and an associated integral, via a nonlinear coordinate change have only been applicable to exceedingly simple CRNs[18]. By contrast, our approach identifies a well-defined nonlinear map between reaction rates of the model variables $f_1, \ldots f_n$, and a defining algebraic invariant, $\rho$ (Eq. 6), which exists for all adaptation-capable CRNs. In contrast to all prior control theoretic approaches, this alternative viewpoint decomposes all RPA-capable CRNs into a constellation of linear integral control problems, each with an associated invariant (i.e. internal model). These are distributed across well-defined RPA-permissive CRN subnetworks, corresponding to RPA-permissive topological basis modules[4], and construct an RPA polynomial, $\rho$, through the process of invariant passing (Fig. 7; see also SI Sections S3 and S4).

Invariant passing describes the process by which polynomial invariants corresponding to adjacent topological features of the CRN's overarching network structure are combined so as to systematically eliminate model variables, and ultimately obtain an RPA polynomial for each independent subnetwork, corresponding to an RPA-conferring topological module of the CRN. As illustrated schematically in Fig. 7a for CRNs of Opposer type, opposer invariants are passed from the distal opposer node to the proximal opposer node; for CRNs of Balancer type (Fig. 7b), invariants are passed from the diverter node to the sequence of balancer nodes within each parallel pathway, culminating at the connector node.

Crucially, we demonstrate that the ability or inability of these invariants to 'pass' within the rowspan of the system is a question of stoichiometric independence of the individual chemical reactions contributing to successive invariants. We outline the significance of this key concept through the analysis of a simple illustrative example (Fig. 8). In Fig. 8a, we depict a deficiency-one CRN comprising three interacting molecules, $A$, $B$ and $C$, which exhibits RPA (and, more

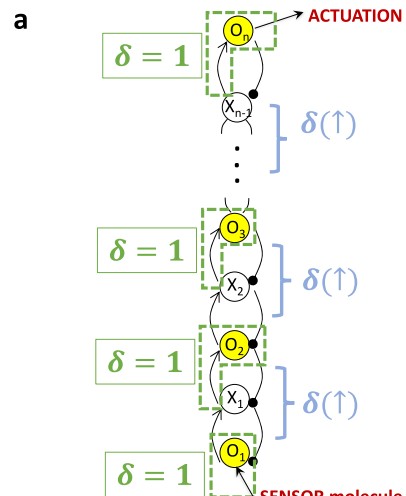

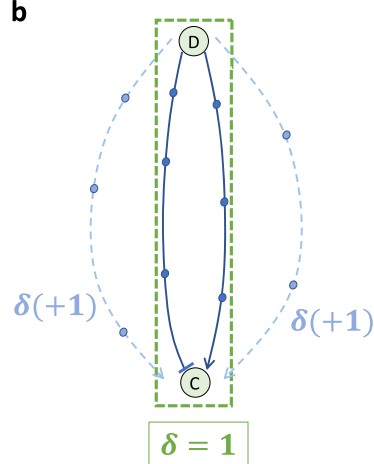

**Fig. 6 | Deficiency-increasing topological features of RPA-capable CRNs. a** An opposing set, containing multiple interconnected feedback loops involving opposer nodes (indicated in yellow), all embedded together into the feedback segment of an Opposer module. As shown, each single opposer cycle contributes a deficiency of one to the CRN. The interspersed feedback loops that connect the individual opposer nodes further increase deficiency (SI Section S4.4.2), as indicated. **b** A Balancer module, containing multiple feedforward segments between the diverter molecule (D) and the connector molecule (C). A Balancer module with just two (incoherent) feedforward segments, without any embedded feedback loop, has a deficiency of one, as shown. Each additional feedforward segment increases deficiency by one, as shown. Any feedback loop embedded into a feedforward segment also increases deficiency by one (not shown).

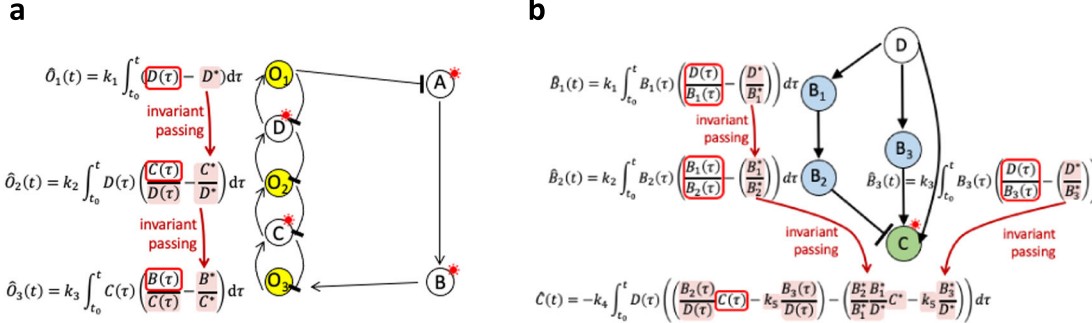

**Fig. 7 | The principle of Invariant Passing.** Rowspan invariants (obtained via a linear coordinate change applied to the model variables) within a single algebraically independent subnetwork of the CRN, are combined nonlinearly (via a concatenating monomial, as needed) to produce the RPA polynomial of the subnetwork. **a** In CRN subnetworks of opposer type, polynomial invariants in the system rowspan are passed from the 'distal' opposer invariant to the 'proximal' opposer invariant. Opposer nodes indicated in yellow. **b** In CRN subnetworks of balancer type, polynomial rowspan invariants are passed from the diverter node, along parallel pathways, to the connector invariant. Balancer nodes indicated in blue, connector node indicated in green.

specifically, ACR) in the molecule $A$. It is easy to show (SI Section S3.1) that this CRN is topologically a Balancer module, where $A$ is the connector, $B$ is the diverter, and $C$ is a balancer. In Fig. 8b, we present a modified version of this CRN that preserves both its topology and its deficiency of one. Both CRNs contain an identical connector polynomial in their rowspans. In addition, both CRNs contain a balancer polynomial in their respective rowspans, as expected. In the original CRN (Fig. 8a), however, the balancer and connector polynomials are stoichiometrically independent, in the sense that the variable to be eliminated in the process of invariant passing (ie. $C$) derives from the reactant complex $B + C$ in the balancer polynomial, and from the reactant complex $C$ in the connector polynomial. Therefore, the connector polynomial must be multiplied by the concentration of molecule $B$ (a concatenating monomial) in order for the balancer polynomial to pass to it, and thereby construct the RPA polynomial. By contrast, the single reactant complex $C$ contributes to both subsidiary polynomials for the modified CRN (Fig. 8b), guaranteeing their stoichiometric dependence.

It is striking to note that the original form of the CRN (Fig. 8a) eludes the Shinar-Feinberg theorem, even though the CRN exhibits ACR and has a deficiency of one. It is thereby clear that, even for the special case of deficiency-one CRNs, the Shinar-Feinberg theorem cannot provide a comprehensive description of ACR (and hence RPA). By contrast, the modified form of the CRN (Fig. 8b), with stoichiometrically-dependent balancer and connector polynomials, does satisfy the Shinar-Feinberg theorem. With the stoichiometric dependence of all subsidiary polynomials now delivering the all-important RPA polynomial to the system's rowspan, this modified CRN necessarily contains two non-terminal complexes ($A + B$, and $B$) that differ in the single ACR-exhibiting molecule, $A$ – thereby satisfying the conditions of Shinar and Feinberg's theorem.

We illustrate the control diagram corresponding to our decomposition into a topological hierarchy of linear controllers in Fig. 9 for the particular case of a single Opposer module, since all Opposer modules necessarily incorporate an overarching feedback structure, and are thus easily described using standard control diagrams. In principle, there should always exist some single nonlinear coordinate change to extract a single output-driven internal model (Fig. 9a) from the system's rate equations, corresponding to a single integral of the system's tracking error (Fig. 9b)[18,22,27]. But in our representation, each

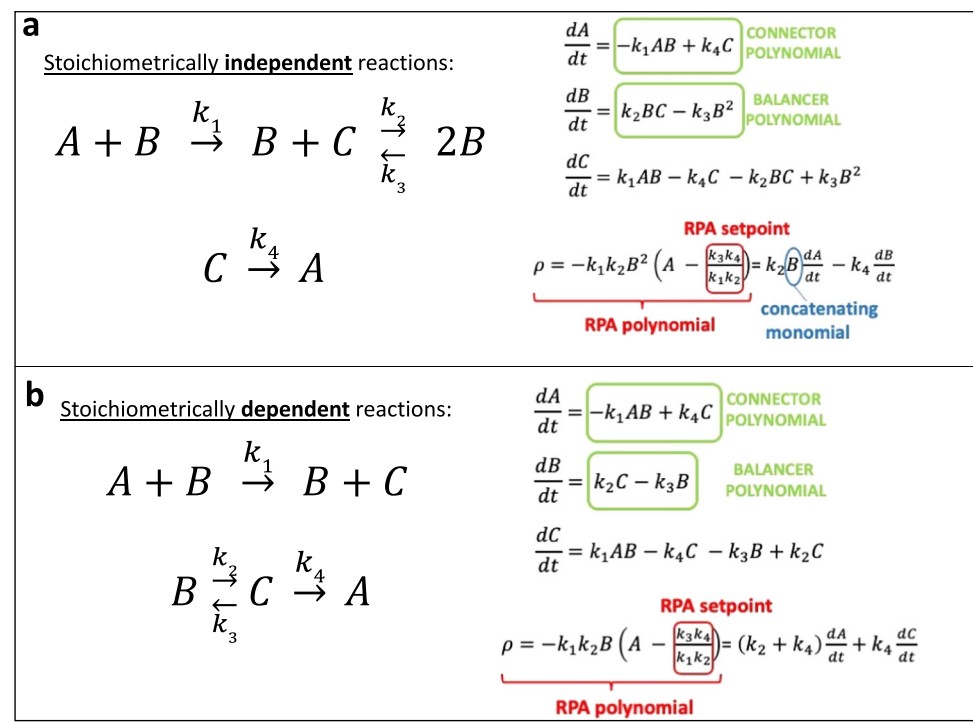

**Fig. 8 | Stoichiometric independence of chemical reactions, and its consequences for invariant passing. a** A simple CRN which is topologically a Balancer module (see SI Section S3.1), involving interactions among three molecules: A (connector), B (diverter), and C (balancer). The connector polynomial (dA/dt) and the balancer polynomial (dB/dt) are stoichiometrically independent, since the CRN complexes that contribute the molecule C (which must be eliminated) are 'C' for the connector polynomial, and 'B + C' for the balancer polynomial. A concatenating monomial (B) is therefore required to reconcile the two invariants to construct the RPA polynomial. **b** A modified version of the CRN, with identical underlying topology. In this case the connector polynomial (dA/dt) and the balancer polynomial (dB/dt) are stoichiometrically dependent, since the complex 'C' contributes to both invariants, and is thereby eliminated within the rowspan of the system. As a consequence, the RPA polynomial resides in the system's rowspan.

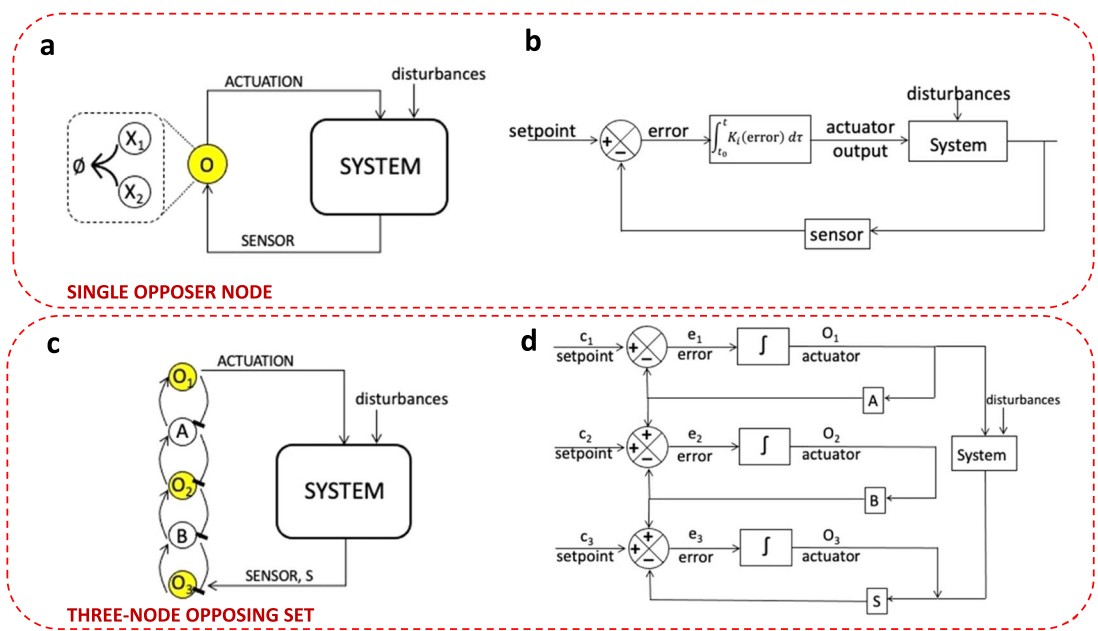

**Fig. 9 | The distribution of internal models into a topological hierarchy. a** A network with a single integral controller. **b** A standard integral feedback control diagram, in which the error between the setpoint and sensor for the system is integrated in a single integral, corresponding to a single internal model. **c** In CRNs, a universal description of integral control is obtained by decomposing the internal model into multiple subsidiary internal models, each corresponding to a linear coordinate change, and each corresponding to an (algebraically) independent subnetwork of the CRN, and an associated topological feature of the underlying network. Here, we depict an Opposer module featuring a three-node opposing set. **d** Since all Opposer modules have a feedback architecture, their feedback control diagram can explicitly incorporate the multiple independent integral-computing elements. The feedback control diagram shown corresponds to the opposer module in (**c**). Opposer nodes are indicated in yellow.

subsidiary polynomial invariant corresponds to an independent internal model, each with its own independent setpoint. For a CRN constituting a three-node opposing set (Fig. 9c), for instance, there will exist three independent internal models and three corresponding opposer integrals, each conferring RPA on a different variable (Fig. 9d). These three independent linear control systems collaborate to confer RPA on the sensor variable of the CRN, and ultimately, the entire embedded network.

**A general algorithm for adaptation detection in complex CRNs**
It is clear from the illustrative example of the EnvZ-OmpR osmoregulatory motif (Fig. 3; see also additional CRN examples in SI) that even for exceedingly simple CRNs, constituting a single RPA module, the integral-computing polynomial invariants may be deeply concealed within the chemical reaction structures, and cannot generally be identified by inspection. For this, we introduce here a general algorithmic method for establishing the RPA capacity of a CRN, which can identify the subsidiary polynomial invariants automatically – even for large and/or high-deficiency CRNs, comprising multiple RPA-promoting topological modules.

Our method is a direct consequence of the fact that the RPA polynomial of any adaptation-capable CRN is a function of *two* variables, thereby converting the question or RPA capacity to a well-defined elimination problem. This elimination problem corresponds geometrically to the projection of the system onto just two variables (one RPA-capable, and one RPA-incapable) – a task that can accomplished via computation of the Gröbner basis of $\langle f_1, \ldots f_n \rangle = \{h_1 f_1 + \ldots + h_n f_n | h_i \in \mathbb{R}[x_1, \ldots, x_n]\}$ with suitable monomial ordering (see SI Section S5). Remarkably, although the problem of computing a Gröbner basis (e.g. by Buchberger's algorithm[30]) for general systems of polynomials is well-known to be NP-Hard[31], the special almost-linear structure of RPA capable CRNs (as described above, see also SI Section S4) allows any RPA-capable CRN to yield to this approach in polynomial time.

Although this method can be applied with any choice of two variables – one RPA-capable and one non-RPA-capable - we provide guidance in our SI on the decomposition of CRNs into independent subnetworks that can be analyzed individually as to their RPA-capacity, and for which particularly judicious choice of two variables can be made. In fact, analysis of deficiency in such independent subnetworks can confirm the inability of a CRN to exhibit RPA, if such is the case, since large and complex CRNs may otherwise require a long (and potentially indeterminate) timeframe for the algorithm to terminate. We provide a fully analyzed example of a non-RPA-capable CRN, to illustrate these principles, in SI Section S4.5. Moreover, the fact that the RPA polynomial is a function of (at most) two variables provides additional opportunities for computational efficiency via the choice of a fast elimination ordering on monomials – e.g. a block monomial ordering, involving two blocks (one for the two variables, and the other for the remaining variables to be eliminated). We provide full details of this method, along with code in the open-source software Singular (www.Singular.uni-kl.de) in our Supplementary Information (see SI Section S5), where we also provide a selection of fully-annotated illustrative examples. This code can readily be applied to any CRN.

## Discussion
Identification of a definitive test for the capacity of a network of chemical reactions to exhibit RPA has been the subject of a long quest, and most attempts have considered only the special case of RPA known as Absolute Concentration Robustness (ACR). These diverse attempts have drawn from a range of different mathematical frameworks, which can be broadly divided into two main categories: the chemical reaction network theory (CRNT) viewpoint[2,26,29], and the engineering control theory viewpoint[16–19].

The pinnacle of CRNT approaches is the Shinar-Feinberg theorem[2] which identifies a sufficient condition for ACR in CRNs of deficiency one[2,29]. It is now well known that most CRNs in nature have a deficiency much greater than one[26] and the Shinar-Feinberg theorem is silent on all such CRNs. In addition, the Shinar-Feinberg theorem cannot reveal the setpoint of any ACR-exhibiting molecules as a function of system rate constants, nor how the existence of ACR corresponds to the presence of integral control. Karp et al.[24], developed an alternative systematic method to identify 'complex linear' polynomial invariants, which require only linear combinations of the mass-action equations of a CRN. Using this method, the two subsidiary polynomial invariants given in Fig. 3c could be identified in an ad-hoc manner. But without recognizing these two invariants as a balancer invariant and a connector invariant, and the general relationship of such invariants to an RPA polynomial, this approach cannot make the crucial connection to the essential structure that characterizes all possible RPA-capable CRNs, and provides no connection to integral control in any such systems.

From the control theory viewpoint, Yi et al.[19] use general linear models to demonstrate the necessity for integral control in all robust asymptotic tracking problems (such as RPA) and extract the internal model for the well-known Barkai-Leibler model of bacterial chemotaxis[13] by ad-hoc (linear) algebraic manipulations. Gupta and Khammash[23] provide a universal characterization, and an explicit integral control implementation, for CRNs with the special property known as maxRPA, where the system setpoint can only depend on two biochemical rate parameters. Moreover, a recent systematic algebraic method developed by Cappelletti et al.[3] can now identify the capacity for RPA in any mass-action CRN for which an RPA polynomial exists in the rowspan of the system. This method explicitly identifies the presence of integral control in all such CRNs, and also reveals the system's setpoint as a function of biochemical rate constants, but is silent on any CRN requiring a nonlinear coordinate change to reveal an internal model.

Until now, general strategies for identifying an internal model via nonlinear coordinate transformations have remained elusive, and specific nonlinear maps have been identified only for exceedingly simple RPA-capable CRNs[18]. Although a single nonlinear diffeomorphism that maps the original model variables to a special block form - thereby explicitly revealing a single internal model - should always exist in principle[18,28], even in models for which no feedback loops are present[22,27] (see Fig. 1b), the identification of such a nonlinear map in even the most complex special cases cannot, of itself, clarify the general principles that unify all possible RPA-capable chemical reaction structures.

By contrast, our approach identifies a well-defined nonlinear map – distinct from the coordinate transformations considered in previous control theoretic approaches[18,19] – between the reaction rates of the individual molecules, $f_1, \ldots f_n$, and a key CRN invariant known as the RPA polynomial. This transformation holds for all RPA-capable CRNs, and constitutes a geometric projection of the full set of molecular concentrations onto a particular subset of the model variables, comprising one RPA-capable molecule and one non-RPA-capable molecule (recognizing that all RPA-capable networks require a minimum of one such variable to constitute the adaptation[4]). The innovative leap that we make from this mathematical cornerstone is to show that this nonlinear map can always be decomposed into a constellation of linear maps, each existing within a topological hierarchy[4] associated to the CRN's underlying flow of biochemical information, and each corresponding to an independent linear control problem. The integrals that are formulated by these independent subsidiary control systems thereby collaborate to confer RPA on one or more molecules in the CRN. Our approach unifies both the control theory and CRNT viewpoints, and extends prior results on the macroscale topologies[4] of RPA-capable

networks to the microscale level of intermolecular interactions within CRNs. The space of RPA-capable CRNs that we identify in this way extends our current understanding of the intricate biochemical interactions that support RPA beyond simple special cases that have been considered in previous work – deficiency-one ACR-capable CRNs[2], ACR-capable CRNs with a linear constrained integral controller[3], maxRPA networks[23], etc. – to a truly general framework that can encompass all forms of RPA, in any CRN.

The combination of a definitive algebraic condition with the special almost-linear structure of the underlying control system provides the essential ingredients for a simple algorithmic test for RPA capacity, that is applicable even to large, high-deficiency CRNs. The only algorithmic method capable of handling CRNs of arbitrary deficiency prior to this work was the necessary condition for ACR identified by Eloundou-Mbebi et al.[26] Being a necessary condition, the Eloundou-Mbebi method can identify a collection of molecules that certainly couldn't exhibit ACR (and therefore RPA), and thereby reduces the number of molecules that must be analyzed in detail for their ACR (RPA) capacity, eg. via extensive numerical simulation. But the Eloundou-Mbebi method is unable to identify, definitively, which molecules *do* exhibit ACR (RPA) since it fails to capture the essential structural characteristics common to all RPA-capable networks. Indeed, the Eloundou-Mbebi method characteristically overestimates the space of molecules that could potentially exhibit ACR/RPA quite significantly[26]. For the deficiency-two model of the EnvZ-OmpR phosphorelay (Fig. 2), for instance, all nine species satisfy the Eloundou-Mbebi condition, even though only *pOmpR* can actually exhibit ACR/RPA (as we can easily prove by the method we present here). And in common with other CRNT-based approaches, the Eloundou-Mbebi condition makes no connection to integral control, and cannot identify the setpoint of any RPA-capable species as a function of biochemical parameters.

Although RPA-conferring CRN structures are known to be quite subtle[32,33], the framework we present here delineates the fundamental principles fully. Only a complete and truly general picture of the integral control problem in CRNs, as we present here, can demarcate the evolutionary trajectories along which complex adaptation-capable biological networks can arise from simpler building blocks, and provide a roadmap for either preserving or disrupting the RPA property in natural, diseased or synthetic networks through design alterations or pharmacological interventions.

The quest to uncover the fundamental design principles that constrain complex signaling networks in nature to implement biologically important functions is considered to be one of the most important and far-reaching grand challenges in the life sciences[34–39]. On the basis of the present study, along with our earlier topological study at the network macroscale[4], RPA currently stands alone as a keystone biological response for which there now exists a universal explanatory framework – one that imposes strict and inviolable design criteria on arbitrarily large and complex networks, and one that now accounts for the subtleties of intricate intermolecular interactions at the network microscale. These universal RPA-permissive design principles now represent a launching-point for future explorations of more complex phenotypes - including some classes of embryonic patterning problems, for instance, where integral control is known to play a role in promoting adaptation of segmentation boundaries to variations in organism size[14,15]. The identification of universal design principles for many other complex phenotypes, such as Turing patterns[37,40,41] and multistability/switching-responses[42,43], is likely to prove more challenging - due, in part, to the central role of equilibrium stabilities, or instabilities, in generating these responses. These grand challenges remain open, and we hope that our study will inspire bold new mathematical thinking in these vitally important directions.

## Methods

For the purposes of obtaining the greatest possible generality and universality, we define RPA from the most general possible viewpoint in this study. In particular, we consider a CRN to exhibit RPA in the concentration, $x$, of some network molecule (directly or indirectly subjected to some disturbance or perturbation) exactly when $x$ maintains a constant steady-state value, $c$, for all steady-states of the system. The setpoint, $c$, is a function of some subset of CRN parameters. Moreover, the CRN exhibits RPA (in $x$) in response to any perturbation or disturbance that leaves the collection of setpoint parameters unaltered (see SI Section S1). We make no assumptions a priori as to which type(s) of network perturbation or disturbance might affect a particular CRN, nor do we impose any restrictions on which (or how many) parameters determine the setpoint.

In our Supplementary Information (SI), we provide full details of all mathematical theorems that support our results, along with their corresponding proofs. The central result of our study, the Two-Variable Kinetic Pairing Theorem (Theorem 1), is discussed fully, and proved, in SI Section S1, along with a full discussion of the RPA polynomial – the fundamental object at the heart of Theorem 1 – in addition to a range of other essential concepts, such as CRN deficiency, partitions of CRN reactions into algebraically independent subsets/subnetworks, matrix decomposition of CRN mass-action equations, and boundary variables. The second pivotal result of this study concerns the nature of the nonlinear transformation (of the CRN rate equations) required to yield the all-important RPA polynomial, and the fact that this transformation is always decomposable into a topological hierarchy of linear maps (and hence, linear integral controllers, and their associated internal models). Our analysis (SI Sections S2, S3, S4) highlights the crucial notion that the deficiency of the CRN fundamentally controls the formation of polynomial invariants that are obtainable through linear coordinate changes. All theorems related to these concepts are discussed in detail, along with their corresponding proofs and a set of fully-analyzed illustrative examples (SI Section S4) and annotated *Singular* code (SI Section S5).

### Reporting summary

Further information on research design is available in the Nature Portfolio Reporting Summary linked to this article.

## Data availability

No datasets were generated or analyzed during this study.

## Code availability

All code used for the analysis of illustrative CRN examples in this study is provided in the Supplementary Information (SI) file. See SI Section S5 for fully annotated code listings, as well as complete listings of code output for all examples considered in this study. All code is written in the open-source software *Singular* (www.singular.uni-kl.de). Detailed guidance on running this code, and adapting the code to the analysis of any CRN, is provided in SI Section S5.

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

## Acknowledgements

R.P.A. is supported by an Australian Research Council (ARC) Future Fellowship (project no. FT190100645) from the Australian Government.

## Author contributions

R.P.A. conceived the study and secured funding, proposed all mathematical concepts and definitions, proved all theorems, undertook all analysis and wrote Singular code. R.P.A. and L.A.L. wrote the paper.

## Competing interests

The authors declare no competing interests.
