## [Peer Review File · Nature Communications]

Review: Universal structures for embedded integral control in biological adaptation

Robyn P. Araujo and Lance A. Liotta

Overview: Development of adaptation mechanisms is crucial for survival across biological scales. Understanding how such mechanisms are generated through biomolecular interactions is a fundamentally important problem? This issue has underpinnings in control theory and it is known that adaptation requires an embedded integral control, which, if identified, can offer insights into how a reaction network achieves adaptation and how robust this adaptation is. The aim of this paper is to construct an algebraic procedure to identify the hidden integral controller in adaptation-capable networks and connect these controllers to the structure of the networks.

Recommendation: Even though the paper studies an important problem, I do not find the analysis presented in this paper to be convincing enough to warrant publication in *Nature Communications*. The manuscript has several shortcomings which I list below:

- **Lack of sufficient novelty:**

- The general principles for RPA have been developed by (slightly) extending previous results by the authors for deficiency zero and deficiency one CRNs. These low-deficiency networks are related to ‘balancer’ and ‘opposer’ modules which are known to form a topological basis for all RPA networks [1]. This connection is explained with two illustrative examples and general principles are formulated by relying on the theory in [1]. Hence, given the earlier results in [1], the incremental novelty in this paper is not significant.
- Many researchers have exploited Gröbner basis and ideal construction for obtaining steady-states in terms of parameters and in particular for studying ACR. For example, see [6, 5, 3] and references therein. It seems that the authors are unaware of these works and in order to assess the novelty of the methods in this paper, these existing works must be cited and discussed.
- More importantly, the central analysis in this paper crucially depends on the result for deficiency one networks which is stated as Theorem 3 in the Supplement. However, this result is not new, as it follows from Theorem D.1 in the Supplement of [2] (see the proof in [2]).

- **Definition of RPA:** The notion of RPA is not mathematically defined. In other words, it is not clear what disturbances are allowed and what are the biochemical parameters on which the output set-point is allowed to depend. On page 11 in the Supplement it is written that for RPA the set-point c should be a rational function of

the biochemical rate constants. However, this would be the case for any steady-state and so what constitutes RPA is not clear.

- **Algorithm may not terminate:** The Gröbner basis algorithm to find the RPA polynomial may not terminate. It is mentioned that failure to terminate for a chemical reaction network (CRN) is a *prima-facie* evidence that the CRN does not exhibit RPA. However this is not mathematically shown. In particular, no proof is given showing that non-termination implies no RPA. Moreover, non-termination of a method is not something that can be ascertained with any certainty by running the code. Accordingly, the authors cannot claim that they have a full 'characterisation' of the RPA property.
- **Form of the RPA polynomial:** The whole approach hinges on the RPA polynomial having the form

$$\rho = g(x, y)(x - c)$$

where $g(x, y) \neq 0$ and y is a non-RPA capable variable which forms the *kinetic pair* to x . It is unclear why $g(x, y)$ can only be a function of one additional variable, apart from the output variable. The proof given on page 12 in the Supplement does not satisfactorily explain why the ideal $I_f \cap \mathbb{R}[\bar{x}]$ will contain polynomials in x_j and x_m that are not in $I_f \cap I_p$.

As an example consider the following network:

Furthermore assume that species $\mathbf{X}_2, \dots, \mathbf{X}_5$ participate in several reactions that do not involve \mathbf{X}_1 but can be catalysed by it. In this scenario the RPA polynomial would be $\rho = x_2 x_3 (\frac{c_1}{c_2} - x_1)$. Hence the function g depends on two non-RPA variables x_2 and x_3 . Please explain how this example is consistent with the form of $g(x, y)$ stated above.

Moreover, it is not immediately clear how existence of multiple opposer/balancer modules translates into existence of corresponding RPA polynomials. This has not been explained in sufficient detail.

- **Connection to existing works:** In [4] the authors consider RPA systems that are maximally robust and find simple linear-algebraic structural conditions that characterise this property in both deterministic and stochastic settings. How do the results in this paper connect to the results in [4]?

References

- [1] R. P. Araujo and L. A. Liotta. The topological requirements for robust perfect adaptation in networks of any size. *Nature communications*, 9(1):1–12, 2018.
- [2] D. Cappelletti, A. Gupta, and M. Khammash. A hidden integral structure endows absolute concentration robust systems with resilience to dynamical concentration disturbances. *Journal of the Royal Society Interface*, 17(171):20200437, 2020.

- [3] G. Craciun, A. Dickenstein, A. Shiu, and B. Sturmfels. Toric dynamical systems. *Journal of Symbolic Computation*, 44(11):1551–1565, 2009.
- [4] A. Gupta and M. Khammash. Universal structural requirements for maximal robust perfect adaptation in biomolecular networks. *bioRxiv*, 2022.
- [5] M. Pérez Millán, A. Dickenstein, A. Shiu, and C. Conradi. Chemical reaction systems with toric steady states. *Bulletin of mathematical biology*, 74(5):1027–1065, 2012.
- [6] A. Sadeghimanesh and E. Feliu. Gröbner bases of reaction networks with intermediate species. *Advances in Applied Mathematics*, 107:74–101, 2019.

Reviewers' Comments:

Reviewer #1:

Remarks to the Author:

Reviewer #2:

Remarks to the Author:

Araujo and Liotta develop a set of precise criteria that a system needs to fulfil in order to perfect adaptation.

This is remarkable and an extremely valuable contribution to the literature in three ways: (i) the search for "design principles" is taking centre stage in synthetic (but also e.g. developmental systems) biology; the first author has proven a leader in distilling precise mathematical criteria for robust perfect adaptation, and I hope that this study will inspire more work in this area, especially in considering the mathematical properties of chemical reaction networks to guarantee different types of behaviour; (ii) in Mathematical biology there are few instances where the mathematical statements can be so precise that biology will have to "obey" these statements. This is one such instance. Finally (iii) the maintenance and control of robust adaptation is of considerable and far reaching biological relevance. I like this manuscript for each of these three points a lot.

The most interesting mathematical aspects of the work have, unfortunately but predictably been relegated to the supplementary information. The results in Figure 2, for example, are very clearly explained and easy to follow in the SI. I personally enjoyed the SI a lot, and while the discussion e.g. in lines 207-215 is clear, it may be too terse for some readers to follow. The SI by contrast was very clear. Fig 3 does a nice job, however, to get the message of kinetic pairing across.

I think it would help readers to understand how translatable to other phenotypes this type of analysis is, or if RPA is particular in this regard of allowing such general mathematical statements to be derived.

Small points:

- much of chemical reaction network theory is obscure to most readers. I would like to see a clear and easy definition of integral control that is accessible to non-expert audiences. Deficiency is maybe another such concept and it would be good to define it on line 146.
- The authors show that their criteria hold for all instances of RPA. I was wondering how easily in practice these mechanisms could be lost? Is it easy to identify points that disrupt RPA?
- there is a vast literature on design principles which could be touched upon at least in passing (limit cycles, multistability, switch-like behaviour, Turing patterns), especially if there is scope for applying similar concepts in these contexts.

Response to Reviewers (R.P. Araujo and L.A. Liotta; NCOMMS-22-15241).

Reviewer #1

Overview: *Development of adaptation mechanisms is crucial for survival across biological scales. Understanding how such mechanisms are generated through biomolecular interactions is a fundamentally important problem. This issue has underpinnings in control theory and it is known that adaptation requires an embedded integral control, which, if identified, can offer insights into how a reaction network achieves adaptation and how robust this adaptation is. The aim of this paper is to construct an algebraic procedure to identify the hidden integral controller in adaptation-capable networks and connect these controllers to the structure of the networks.*

Recommendation: *Even though the paper studies an important problem, I do not find the analysis presented in this paper to be convincing enough to warrant publication in Nature Communications. The manuscript has several shortcomings which I list below:*

Our response:

We would like to sincerely thank the reviewer for his/her significant investment of time in considering our work, and would like to acknowledge that the reviewer – on the basis of the content in his/her feedback – is clearly an expert in the RPA field, which we both respect and value. The reviewer has given our work a rather harsh review, but we are very grateful to have the opportunity to defend our work, and to make additional improvements and clarifications after thoughtfully considering the reviewer’s feedback. We thank the reviewer in advance for his/her generosity and willingness to re-consider our work in light of these detailed responses and clarifications. We have noted all revisions to the main manuscript, and to our SI, with **red text**, so that the reviewer can readily identify which parts of our article have been updated since the original submission.

We do wish to respectfully point out at the outset that the primary aim of this work is **not** “to construct an algebraic procedure to identify the hidden integral controller in adaptation-capable networks” as the reviewer suggests. Rather, our **primary** goal is to provide a **comprehensive and universal description of all possible RPA-capable chemical reaction networks (CRNs), that could ever exist** – recognising (in contrast to Shinar/Feinberg, for example) that **all** instances of RPA, including ACR, must implement “some” form of integral control. In so doing, we wish to also propose a definitive criterion, or test, that captures the presence of the RPA property in any CRN that possesses it. The nature of this universal characterisation of RPA at the level of intermolecular interactions, as codified in the most general terms by our Theorem 1, does indeed suggest an algorithmic method for testing specific CRNs as to their RPA capacity, and for identifying their hidden integral controllers and setpoints. That this algorithmic method could terminate in polynomial time for any RPA-capable CRN, comparable to Gaussian elimination, is a consequence of the previously unrecognised relationship we identify between **increases in CRN deficiency** (ie. beyond a deficiency of **one**) and the **associated modifications to**

polynomial invariants in the rowspan of the chemical reaction rates. For this reason we are able to identify, for the first time, the special '**almost linear**' **structure** of the mathematical transformation (coordinate change) required to reveal the CRN's "RPA polynomial" – a central concept in our work - if such exists for the CRN in question. We thereby "solve" the integral control problem for all RPA-capable CRNs, in complete generality, at the microscale level of intermolecular interactions, with their associated mathematical graph structures. In so doing, we demonstrate how RPA may be accommodated by CRNs of arbitrary deficiency (not just "low-deficiency" CRNs, as the reviewer goes on to claim). Our two main outcomes are thus inextricably related: the **special CRN structures** that hold for all possible RPA-capable networks allow the system's reaction rates to be projected, mathematically, onto **two** variables via a process that does not differ substantially from Gaussian elimination – thereby revealing a computationally feasible algorithmic test for the presence of RPA.

Importantly, our new and unexpected findings on the relationship between the **deficiency** of a CRN and its overarching **topological structure** allow us to make a new and crucial connection to our previous work on RPA at the network macroscale [1], thereby enabling us to identify, definitively, the one-to-one correspondence between the subsidiary **linear** control problems and key topological features of the CRN. The fact that we can connect these new results to our previous general (macroscale) topological solution to the RPA problem is, without question, **extremely powerful**. But it is entirely untrue that we simply **impose** our previous results on RPA topologies at the network macroscale onto the microscale of intermolecular interactions in a CRN, leading to a result of "incremental novelty" as the reviewer goes on to suggest. In fact, no previously published work to our knowledge has been able to identify the general relationship between deficiency and linear complex invariants in RPA-capable networks. Cappelletti et al. [2], for instance, do note that an "RPA polynomial" (as we call it here) can exist in the rowspan of a CRN for deficiencies greater than one – **if the RPA polynomial in question happens to be in the rowspan of the CRN, which is not generally the case** – BUT fail to identify **how** any added deficiency (beyond unity) allows this crucially important "RPA polynomial" to reside in the rowspan. As a consequence, even the very simplest RPA-capable CRNs whose RPA polynomial does **not** reside in the rowspan (e.g. the toy model discussed by those authors, that we analyse as Example 1 in our SI (Section S3.1)) simply elude the Cappelletti et al. [2] approach. No previous work has been able to characterise, in complete generality, the **nonlinear coordinate changes** that are needed to reveal the capacity for RPA in CRNs in an entirely universal manner.

We are truly grateful for the reviewer's fresh set of eyes on our work, and have carefully revised our manuscript (and SI) to ensure that the novelty and scientific importance of our findings have been clearly and accurately communicated. We acknowledge, in particular, that the preamble to our Supplementary Information (i.e. the Supplementary Text prior to Section S1) could have been organised much better, with the aims and novel contributions of our work expressed much more explicitly. We acknowledge that our original submission mentioned the development of a

definitive ‘RPA test’ first in this section, before describing the overarching goal of demonstrating the truly universal principles of RPA in CRNs, and the relationship of this universal description of RPA to integral control. This section has now been carefully revised to read as follows:

The overarching goal of this work is to provide a comprehensive and universal description of all possible Chemical Reaction Network (CRN) structures that can implement the keystone biological function known as Robust Perfect Adaptation (RPA), and in so doing, to propose a definitive algebraic condition that captures presence of the RPA property in any CRN that possesses it. We also recognise here that all forms of RPA, including the special case known as Absolute Concentration Robustness (ACR), must implement some form of integral control. Our universal characterisation of RPA at the level of intermolecular interactions thereby provides an algorithmic method not only for detecting the RPA property, but also for identifying the hidden integral controllers, as well as the ‘setpoints’ of any RPA-exhibiting molecules.

As we show in the pages to follow, providing such an all-encompassing description of RPA at the microscale-level of biological networks - that is, in CRNs, which account for the mathematical graph structure of intermolecular interactions - has required us to reconcile a wide range of disparate mathematical viewpoints, ranging from Chemical Reaction Network Theory (CRNT), control theory and the internal model principle, and algebraic geometry. The essential structure of our mathematical development is as follows:

1. In Section S1 we identify a universal ‘kinetic pairing’ principle, which captures the essence of RPA in all CRNs that possess the property, and which is codified by our Two-Variable Kinetic Pairing Theorem (Theorem 1, Section S1.4). This theorem guarantees that, for all RPA capable CRNs, the geometric projection of the CRN's steady-state locus onto a state space defined by just *two variables* is captured by a distinguished algebraic invariant that we call an *RPA polynomial*. In this context the term ‘variable’ has a particular meaning, that we define carefully in Section S1.4.

In this Section we also introduce readers to the fundamentally important notion of *algebraically independent subnetworks* - a simple yet powerful concept introduced by Martin Feinberg (see Appendix A.6 in [9] for a detailed overview), whose implications for the decomposition of CRNs into independent subnetworks, and ultimately *topological modules*, that are compatible with RPA are recognised for the first time in the present work. This key concept provides a vital ingredient which contributes to a definitive characterisation of RPA-capable CRNs of arbitrarily high deficiency, far beyond the simple deficiency-one ACR-capable CRNs covered by the Shinar-Feinberg theorem [7].

2. In Section S2 we briefly discuss the topological implications of Theorem 1 and, for later reference, make a preliminary connection with previous work on the topological structures that are known to hold, at the network macroscale, for all RPA-capable networks [1]. Importantly, the universal topological solution to the RPA problem [1] was developed considering only a single external input or disturbance, and did not account for the possibility of *no external inputs*, with alterations in total abundances of the interacting elements (as specified by the initial conditions of the system) constituting the only possible perturbation to the system. More importantly, *and in stark contrast to the present work*, the universal macroscale solution to the RPA problem [1] gives no concrete information as to the CRN structures (at the network microscale) that could engender RPA or satisfy its strict topological requirements.
3. Since most RPA-capable CRNs identified in prior studies identify an RPA polynomial (as we call it here) from only linear coordinate changes (see, for example, Cappelletti et al. [2]), in Section S3 we consider the algebraic properties of CRNs that allow the RPA polynomial (as expressed in Theorem 1) to reside in the rowspan of the CRNs chemical reaction rates (i.e. given by an \mathbb{R} -linear combination of the rate equations). For this purpose, we analyse two carefully chosen CRN examples in detail, and demonstrate the existence of several subsidiary polynomial invariants *in the rowspan* for each case, and note the relationship of these polynomial invariants to the overarching topology of the system. We demonstrate via deficiency-preserving transformations of the respective CRN structures that these subsidiary polynomial invariants *combine* to yield the all-important RPA polynomial *within the rowspan* exactly when the invariants are *stoichiometrically dependent*. If the invariants are *stoichiometrically independent*, on the other hand, a *concatenating monomial* is required to reconcile the invariants, thereby introducing a necessary nonlinearity to the transformation that produces the RPA polynomial. The detailed analysis of these simple examples provides the reader with an accessible overview of the general principles that characterise RPA-capable CRN structures before we develop these principles more rigorously and in complete generality in the remaining Sections.
4. In Section S4 we demonstrate, through an analysis of deficiency-increasing reactions within independent CRN subsets, that the coordinate change required to extract the all-important RPA polynomial from the CRN rate equations is always *almost linear*, in the sense that the RPA polynomial can always be decomposed into a collection of ‘complex linear invariants’. From a control theory viewpoint, these observations lead to the previously unrecognised conclusion that the ‘internal model’ for any RPA-capable CRN can always be decomposed into a collection of *linear* integral controllers, in addition to reconciling the processing of biochemical information at the network microscale (i.e. via the intricate intermolecular interactions that

comprise a CRN) with the overarching topological features at the network macroscale [1].

5. In Section S5, we provide the reader with extensively annotated code in the open-source software *Singular* (www.singular.uni-kl.de) to test the RPA capacity of a range of illustrative examples. This code can be adapted readily to the analysis of any CRN. If RPA does obtain for a particular CRN under investigation, an important consequence of the special almost-linear structure of the transformation required to identify the RPA polynomial is that the computational demands of this algorithm do not differ substantially from those of Gaussian elimination (a polynomial-time algorithm).

We clarify the nature of our scientific advance in further detail as we address each of the reviewer's individual points in turn.

- **Lack of sufficient novelty:**

- *The general principles for RPA have been developed by (slightly) extending previous results by the authors for deficiency zero and deficiency one CRNs. These low-deficiency networks are related to 'balancer' and 'opposer' modules which are known to form a topological basis for all RPA networks [1]. This connection is explained with two illustrative examples and general principles are formulated by relying on the theory in [1]. Hence, given the earlier results in [1], the incremental novelty in this paper is not significant.*

Our response: We wish to respectfully emphasize, first and foremost, that these **completely general principles** by which RPA is **always** realised **at the level of intermolecular interactions** in a CRN have **by no means** "been developed by (slightly) extending previous results (of ours) for deficiency zero and deficiency one CRNs". Our framework is certainly not limited to "low-deficiency networks", as the reviewer claims. The general principles, which we succinctly summarise for a general audience via two particularly simple illustrative examples, in order to provide as accessible an overview of the main ideas as possible at the outset, have most assuredly not been "formulated by relying on the theory in [1]".

Our previous work [1], considers the broad macroscale topological structures that are compatible with an RPA response, and delineates the full solution space to the RPA problem from this viewpoint. This approach considered an abstract network, with a single 'input stimulus', or disturbance, delivered to a distinguished input node. The network was considered to exhibit RPA exactly when a distinguished output node, not necessarily distinct from the input node, could always return to a single, fixed value at steady-state (the 'setpoint'), independently of the magnitude of the input stimulus, and independently of the choice of parameter values. Using a topological method, where the flow of biochemical information could be partitioned into the subsets of a topology that accounted for the algebraic requirements of RPA, an exhaustive description of all possible RPA-compatible network designs, induced

relative to the chosen input-output node pair, could be identified. This approach definitively established that all RPA-capable networks of this type must be decomposable into two types of well-defined topological structures, or ‘modules’ – Opposer modules and Balancer modules – and concluded that certain key ‘computational nodes’, corresponding to well-defined constraints on reaction kinetics, must be embedded into these modules. In particular, Balancer modules require one or more balancer nodes, exhibiting ‘balancer kinetics’ along with a single connector node, exhibiting ‘connector kinetics’; Opposer modules, by contrast, require one or more opposer nodes, exhibiting ‘opposer kinetics’, organised (topologically) into structures known as ‘opposing sets’.

But these conclusions, by themselves, provide *absolutely no information* on how these ‘reaction constraints’ might be realised, *in general*, by the intricate interactions among collections of molecules. Nor has it been clear until now how any of the *known* RPA-capable (or ACR-capable) collections of chemical reactions (CRNs) encode any of the topological requirements for RPA that we had identified in [1]. Indeed, in our earlier paper [1], our illustrative examples employed simple ‘invented’ functional forms (disconnected from CRN structures) which encapsulated the principles of opposer kinetics, or balancer/connector kinetics, in the most straightforward way, since the focus of that earlier work was the overarching network design principles that must hold at the network macroscale. An opposer node P_o , regulated by some node R , for instance, could be assigned a kinetics of the form $dP_o/dt = k_1R - k_2$, which is zero-order in P_o . But this contrived reaction form could certainly not be induced by a CRN for a variety of mathematical reasons, including the fact that (by the so-called ‘Hungarian Lemma’), all terms preceded by a negative sign must include the species that is the subject of the reaction equation – in this case, P_o . Indeed it has been entirely unclear, until now, how the necessary conditions for RPA could be realised *in general* by ‘real’ chemical reactions involving molecules that interact in specific and potentially intricate ways according to a mathematical graph structure (i.e. a CRN). Previous studies (e.g. Shinar and Feinberg [7], Cappelletti et al. [2], Karp et al. [8], etc.) have only provided partial answers for particular special cases.

Of course, the antithetic integral control motif is a simple and well-known reaction structure that can serve as an ‘opposer node’, and can encode opposer kinetics (and thereby implement integral control) in a fairly obvious and straightforward manner via a particularly simple linear transformation. But this mechanism by no means explains the RPA (ACR) property observed in the Shinar-Feinberg deficiency-two model of the EnvZ-OmpR network, for example. In fact, until now, it has not been at all clear how the EnvZ-OmpR motif, or virtually any other known example of an ACR/RPA-capable CRN, might relate to the topological principles we identify in [1]. That the completely general description of RPA-permissive CRN graph structures, as we introduce in the present paper, should be able to map in such a clear, direct and well-defined way with our earlier universal description of RPA *at the network macroscale* is both remarkable and immensely satisfying. In particular, the revelation that balancer ‘nodes’, opposer ‘nodes’ and connector ‘nodes’ must *necessarily* be *linear* transformations of the CRN’s molecular concentrations and

that, combining these invariants nonlinearly in the very specific manner we describe here, gives rise to a completely **general** description of ***all possible RPA-capable CRNs***, is entirely unexpected. *A priori*, there was no reason to think that there would necessarily be such a straightforward way to concretely reconcile the CRN microscale with established design principles of the network macroscale [1] on the basis of any previous work. Only in retrospect does this seem inevitable!

In any case, no previous study has made the well-defined connections we make here between CRN deficiency, network topology (encoding the flow of biochemical information through the graph structure of a CRN) and the internal model principle, in addition to the general results pertaining to the decomposition of the requisite nonlinear map (that extracts a suitable internal model) into a collection of linear maps, governed by the fundamental algebro-geometric requirements of RPA (and hence, ACR). As a consequence, no previous study has been able to provide a definitive description of *all possible* RPA-capable (and, more specifically, ACR-capable) CRNs, nor a well-defined algebraic condition (and associated algorithmic test) that captures the presence of the RPA property in any CRN that possesses it. Although there have certainly been previous ad-hoc attempts (e.g. in [5]) to demonstrate ACR/RPA capacity in a specific CRN via computation of a suitable Gröbner basis, these ad-hoc attempts, of themselves, are not generalisable (as we explain in greater detail later, in response to the reviewer's more specific queries regarding prior work). Moreover, there was no reason to think that this approach would be computationally feasible ***in general prior to our demonstration of the universal decomposition of the underlying nonlinear coordinate change into a connected sequence of linear maps***. (Again, this is what we mean when we claim that the nonlinear map is 'almost linear'). Of even greater computational concern is that fact that, prior to our proof that the RPA polynomial (as we call it here) is always obtained via geometric projection onto exactly ***two variables*** (noting that 'variable' does not *necessarily* mean 'species'!), prior ad-hoc attempts to compute RPA-relevant Gröbner bases have been forced to use the most computationally expensive monomial ordering (i.e. the lexicographic ordering), since it was not known at the outset which, or how many, variables had to be eliminated. These former potential computational challenges have all been robustly resolved by the present work.

Of course, most of the RPA-capable CRNs that have been identified at the present time exhibit a special form of RPA known as Absolute Concentration Robustness (ACR). Unlike the abstract networks we consider in [1], there is no external 'input stimulus' for these networks, whose steady-states can only be altered by varying the total abundances of the various interacting molecules, as specified by the initial conditions for the reactions. For example, the Shinar-Feinberg models of the EnvZ-OmpR motif, comprising both deficiency-one and deficiency-two versions, are known to exhibit ACR in the phosphorylated OmpR moiety, yet no previous study of this motif (including the recent analysis of the deficiency-one version by Cappelletti et al. [2]), has succeeded in connecting the ACR capacity of this particular molecule to its topological balancer structure (and certainly not in a way that allows a ***general*** connection to be made to the set of ***all possible*** RPA/ACR-capable CRNs). As we show in great detail in our newly revised manuscript, a balancer structure arises

from the fact that, for this CRN, the one unit of deficiency encodes *two* parallel pathways that regulate the phosphorylation of OmpR, whereas the two units of deficiency (in the deficiency-two version) encodes *three* such feedforward pathways that regulate the same phosphorylation cycle. Indeed, it is the deep connection we identify between deficiency and the flow of biochemical information (topology), along with the relationship to modularity of the decomposition of RPA-capable CRNs into algebraically-independent subnetworks, that feature among the major conceptual advances of the present work. In our original submission, we had devoted the entirety of sections S4.2.1 and S4.2.2 in our SI to a succinct consideration of these ideas. In our revised submission, we have extensively revised the preliminary sections of our SI, adding in significant additional discussion on the significance of deficiency (a key CRNT concept) in Section S1.2, and have added in an entire new section (Section S1.3), which explains the concept of algebraically independent subnetworks of a CRN in great detail. (This is in contrast to our original submission, in which our far briefer discussion of algebraically independent subnetworks was delayed until Section S4.2.1). In our revised SI, the crucial connections between deficiency and topology are thereby made clear and explicit from the outset. We also briefly analyse the example of the Cappelletti et al. [2] toy model from the point of view of a decomposition into algebraically-independent subnetworks, along with a careful consideration of the partitioning of deficiency and rank into these subnetworks, so that the topological discussion of this (and other) networks in later Sections, as well as the general relationship of CRN topologies to deficiency and independent subnetworks, can be more readily appreciated by the general reader.

Importantly, ACR-capable CRNs, despite not being driven by exogenous inputs, nor perturbed by exogenous disturbances delivered to specific molecules, are still subject to the topological requirements that unify the broader class of RPA-capable CRNs which *generally do* involve external inputs – a deeply important result which fundamentally governs how such CRNs respond to perturbations by *other* (new) classes of disturbances (e.g. enzyme inhibitors and other small molecules), or new exogenous inputs/disturbances distinct from those represented in the original CRN. Since the detailed reaction structures governing complex cellular signalling networks (e.g. signal transduction networks in Metazoa) remain mostly unknown, having access to such a complete and universal description of RPA-capable CRN structures ***in the abstract*** is immensely powerful, particularly for considering new possibilities for molecular-targeted therapies, for instance, and for considering evolutionary alterations to CRNs.

We do acknowledge that our earlier work [1] foreshadowed the principle of a topologically distributed internal model, where one internal model could ‘feed into’ another topologically related one. Nevertheless, no prior work has been able to reconcile the mathematical properties of dynamical systems induced by CRNs with these previously-identified principles, and our current work highlights that making the fundamental connection between macroscale (e.g. from [1]) and CRN microscale poses significant mathematical challenges. Our new study clearly delineates the full suite of new mathematical ideas required – on the relationship of subsidiary linear

invariants (polynomials) to the algebraic elimination problem; on the relationship of stoichiometric independence (which is itself a function of the algebraic structure of the vertices of the CRN graph) to the inability of polynomial invariants to ‘pass’ within the rowspan; and on the requirement for ‘concatenating monomials’ and the relationship of these to the topological structure of the CRN (for which we identify a previously unrecognised connection with CRN deficiency, as we explain in far greater detail in our revised submission).

We truly value the reviewer’s feedback, which has helped us appreciate where our original submission failed to compellingly communicate the novel aspects of this work. We are deeply grateful for the reviewer’s time and effort in re-considering our extensively revised work, and are thankful for his/her support.

- *Many researchers have exploited Gröbner basis and ideal construction for obtaining steady-states in terms of parameters and in particular for studying ACR. For example, see [6, 5, 3] and references therein. It seems that the authors are unaware of these works and in order to assess the novelty of the methods in this paper, these existing works must be cited and discussed.*

Our response: We thank the reviewer for highlighting these references, which use Gröbner bases to elucidate various aspects of CRNs and even consider ACR in one instance [5]. We value the reviewer’s suggestion that the novelty of our new work could be made clearer by citing and discussing earlier work employing Gröbner bases and ideal construction in the context of chemical reaction networks.

We certainly acknowledge that the elucidation of polynomial ideals via Gröbner basis computations is well-known to have wide-ranging applications to problems in science and engineering [10]. We do not claim that the use of Gröbner bases, in and of itself, to elucidate some property of a polynomial system is novel.

There are actually exceedingly few published papers in the literature currently that employ Gröbner basis methods, or ‘ideal construction’ to study RPA/ACR specifically, other than the article by Perez Millan et al. [5]. Those authors compute a Gröbner basis in an ad-hoc manner for the deficiency-two EnvZ-OmpR motif, and use this result to demonstrate the capacity for ACR in one of the species. But the chief focus of that paper [5], and of the Craciun et al. paper [3], is not ACR, or RPA, but CRNs that exhibit ***toric steady-states***. As our study makes clear, RPA-capable CRNs ***do not, in general***, give toric ideals, or have toric steady-states. Although the EnvZ-OmpR motif analysed by Perez-Millan et al. [5], along with several of the illustrative examples we consider in our paper ***are*** characterised by toric ideals, it is clear that this could not possibly be a ***general*** property of RPA-capable CRNs for a number of reasons, including the fact that the pairing function $g(x, y)$ (see our Theorem 1) can be a multi-term polynomial, and not simply a power product of species (monomial). In addition, even in cases where the RPA polynomial itself is a binomial, this condition does not guarantee that the steady-state ideal has a (standard) generating set composed ***entirely*** of binomials. A simple example of an RPA-capable CRN that is

not generated by binomials, and is therefore not characterised by a toric ideal, is given in Figure 2 (previously Figure 1) of our main paper. (See, in particular, SI S5.2, where we compute a number of relevant Gröbner bases for this particular CRN, thereby demonstrating that the ideal corresponding to its chemical reaction rates cannot be generated by binomials). Thus, while the class of RPA-capable CRNs has a non-empty intersection with the class of toric dynamical systems, the study of toric ideals, unfortunately, cannot capture the fundamental essence of the RPA property in CRNs.

In our extensively revised SI, we provide a detailed commentary on the computational demands of Gröbner basis-computing algorithms from a complexity standpoint (see Remark 3 following Definition 3 in Section S1.5), and on the simplifying structure afforded by RPA-capable CRNs (owing to the ‘almost-linear’ coordinate change that we demonstrate can always extract the ‘RPA polynomial’ from the CRN rate equations). At the conclusion of those explanations, we explain that

“ ... the ACR-capacity of the deficiency-two EnvZ-OmpR motif was demonstrated via computation of a Gröbner basis, in an ad-hoc manner, by Perez Millan et al. [5]. But without knowing which, or how many, variables *generally* characterise the RPA-encoding elimination ideal, and without establishing the special ‘almost linear’ structure of the mathematical transformation required to identify the RPA polynomial *in any RPA-capable CRN*, this approach does not generalise, and cannot be applied to the systematic analysis of large and complicated CRNs (e.g. in the context of metabolism [11] for which candidate RPA-capable molecules have not already been posited from experimental evidence).”

Note that this is not a criticism of the Perez-Millan et al. [5] paper in any way. Again, the focus of the Perez Millan et al paper is not ACR-capable networks, so there is nothing amiss with their ad-hoc approach to the particular CRN they consider, which happens to be characterised by a toric ideal (which is the focus of their paper); they do not claim to present a general method that will necessarily be valid (or computationally feasible) for any ACR-capable CRN. They simply show that the particular CRN they consider does exhibit ACR, and the particular Gröbner basis they compute confirms this.

The Sadeghimanesh and Feliu paper [6], on the other hand, considers the computational demands of Gröbner basis algorithms for the analysis of CRNs, and proposes a transformation of CRNs containing ‘intermediate species’ to yield a core network from which these intermediates are absent. This approach allows a more computationally efficient monomial order to be employed in Gröbner basis computations, where only the intermediates are subject to the computationally-expensive lexicographic ordering, and with faster orderings (e.g. grevlex – graded reverse lexicographic) applied to non-intermediate species. Thus, what these authors effectively achieve is the development of a particular ‘block ordering’ on

monomials, and which can potentially speed up Gröbner basis computations quite significantly in comparison with a ‘complete’ lexicographic monomial ordering. This approach can be employed, in principle, for **any** CRN containing intermediate species (as defined in their paper [6]) – not just RPA-capable CRNs.

Our paper, by contrast, focusses squarely on RPA-capable CRNs, and aims to discover their general properties. For **these** CRNs, **in contrast to** the much broader class of CRNs considered by Sadeghimanesh and Feliu [6], the computational demands confronting any Gröbner basis computations are comparatively very light. There are two reasons for this: (1) If the CRN is indeed RPA-capable, then Theorem 1 guarantees that the RPA polynomial can be obtained via elimination **of all but two** variables. As a consequence (and as we point out explicitly in Section S5 where we instruct the reader on the implementation of these algorithms in Singular, using a variety of illustrative examples), we can **always** use a ‘block ordering’ on monomials, where the two chosen projection variables are ordered ‘lower’ than the remaining variables. Having established this, the computationally expensive lexicographic ordering is not required: both of the ‘blocks’ in this special elimination order can now accommodate a fast monomial ordering (e.g. grevlex, sometimes referred to as degrevlex, which uses the syntax ‘dp’ in Singular). But there is a second, and more important reason: (2) The transformation that reveals the RPA polynomial is ‘almost linear’ in the sense that only \mathbb{R} -linear combinations of the rows of the system are required to identify the ‘fundamental’ (subsidiary) polynomial invariants (which are subsequently combined nonlinearly via concatenating monomials). For this reason, the syzygy polynomials (S-polynomials) that are calculated as part of Buchberger’s algorithm are identical to the linear row combinations that are computed during Gaussian elimination. But note that this computational ‘streamlining’ is linked inextricably to the special CRN structures that implement RPA, and establishing this key property is one of the highly novel contributions of our work. For more general CRNs, computing Gröbner bases may be computationally infeasible (especially if imposing a global lexicographic ordering at the outset, as one is often forced to do without any prior analytical guideposts); from this point of view, the approach offered by Sadeghimanesh and Feliu [6] is both interesting and potentially helpful.

We also emphasize that the existence of an algorithm that can test for the RPA-capacity in a particular CRN does not, in and of itself, reveal the definitive properties of all possible RPA-capable CRNs. Our paper is not “about” Gröbner bases as such. That we can, indeed, employ Gröbner basis-computing algorithms (e.g. Buchberger’s algorithm) to test for RPA capacity, **in general**, is a consequence of the special properties of RPA-capable CRNs that we identify for the first time in this new paper.

- *More importantly, the central analysis in this paper crucially depends on the result for deficiency one networks which is stated as Theorem 3 in the Supplement. However, this result is not new, as it follows from Theorem D.1 in the Supplement of [2] (see the proof in [2]).*

Our response: We have carefully scrutinized the proof to Theorem D.1 in the Supplement of [2], and acknowledge that the reviewer is entirely correct to note that our Theorem 3 follows directly from Theorem D.1 in [2], and that our essential mathematical argument is virtually identical to that in [2]. We have now updated our preamble to Theorem 3 (Section S4.2) to emphasize to the reader that this alternative version of Shinar-Feinberg's theorem draws on arguments developed by Cappelletti et al. [2], and provide a reference to Theorem D.1 in [2]. In the interests of a more comprehensive and scholarly exposition of the key ideas (for later use in the remaining sections of the SI), we have also taken this opportunity to add in a second version of the theorem (Theorem 4), applicable to deficiency-zero CRNs, which shows that binomials involving any pair of terminal complexes from the *same* terminal SCC must also reside in the rowspan of the CRN reaction equations. Following these two theorems, we also add a remark, again citing Cappelletti et al. [2] Proposition C.5, that the same mathematical arguments (as employed in Theorems 3 and 4) can readily be extended to demonstrate that binomials involving either a terminal complex and a non-terminal complex, or two terminal complexes from different SCCs, can never reside in the rowspan of a CRN.

We sincerely thank the reviewer for his/her careful attention to this key technical detail, and for ensuring that all mathematical concepts presented in our work are correctly attributed. We feel that the presentation of the mathematical ideas in this section are now much more comprehensive and scholarly, and the additional references to the Cappelletti et al. [2] paper are immensely valuable. We would like to respectfully emphasize, however, that the novelty of our present paper lies in not so much in our Theorem 3 (or D.1 in [2]) but in demonstrating how this result on deficiency-one (and deficiency-zero) networks *may be extended to RPA-capable networks of any deficiency*. The remainder of the SI, following Theorems 3 and 4, is devoted to precisely this goal. (The generalisations the arbitrary deficiency are also much more clearly and thoroughly communicated in our revised manuscript, with the help of the comprehensive explanations in the earlier sections of the SI (particularly Sections S1.2 and S1.3) on how deficiency governs the flow of biochemical information within a CRN (and ultimately relates to topology), as well as the decomposition of a CRN into algebraically independent subnetworks).

- ***Definition of RPA:*** *The notion of RPA is not mathematically defined. In other words, it is not clear what disturbances are allowed and what are the biochemical parameters on which the output set-point is allowed to depend. On page 11 in the Supplement it is written that for RPA the set-point c should be a rational function of the biochemical rate constants. However, this would be the case for any steady-state and so what constitutes RPA is not clear.*

Our response: The reviewer's point that we should have given a more explicit mathematical definition of RPA is well taken. We truly appreciate such helpful advice.

We now include the following text at the beginning of Section S1 in our Supplementary Information:

Definition of RPA

In this study we consider Robust Perfect Adaptation (RPA) from the most general possible viewpoint. In particular, a CRN exhibits RPA in the concentration, x , of some molecule exactly when x maintains a constant steady-state value, c (the molecule's 'setpoint'), *for all steady-states of the system*. The setpoint, c , is a function of some collection of CRN parameters. Moreover, the CRN exhibits RPA (in x) in response to *any perturbation or disturbance* that does not feature in the functional form of the setpoint, c .

With this very broad definition of RPA in mind, we recognise that there are many possible types of perturbations/disturbances that could alter the steady-state of the system:

1. One or more 'external' inputs. A disturbance of this type could arise in the form of some input molecule, I , whose concentration is given by a step function generated by an 'exosystem' ($dI/dt = 0$, not included among the CRN reaction rates).
2. Alterations in total expression levels (abundances) of the interacting molecules. If this is the only type of perturbation possible, then the CRN is considered a 'conservative network', corresponding to a 'closed' system (see [9], Chapter 4). In this scenario, the CRN captures the interconversion of molecules among a variety of possible forms. The dynamical system is thereby constrained to evolve within a particular stoichiometric compatibility class (an affine space parallel to the CRN's stoichiometric subspace – see Section S1.2) determined by the initial conditions. The orthogonal complement to the stoichiometric subspace determines the conservation relations for the CRN, which specify the constant expression levels of the various interacting molecules. Where a molecule can exhibit a fixed concentration across all steady-states when the CRN is subject to perturbations to these total expression levels, this type of RPA is referred to as Absolute Concentration Robustness (ACR, see Chapter 9 in [9]).
3. A disturbance that is encoded in a CRN parameter such as a production rate, μ , of some molecule, M , as reflected in a CRN reaction of the form $\phi \xrightarrow{\mu} M$. Note that CRN parameters determined by the intrinsic chemical properties of the interacting molecules (e.g. association/dissociation constants, catalytic constants, etc.) are very stable and not readily perturbed, except via mutation of the interacting molecules.

In the pursuit of greatest generality, we make no assumptions *a priori* as to which type of network perturbation might affect a CRN, nor do we impose any restrictions on which (or how many) parameters determine the setpoint. Of course, if the setpoint for a putative RPA-capable molecule involves a parameter μ that can be perturbed (see case 3. above) then it is clear that the molecule cannot exhibit RPA in response to *that particular disturbance*.

We can appreciate that the reviewer might have a different perspective on RPA from us, and hope that he/she is willing to accept our viewpoint and mathematical goals.

Regarding the reviewer's comment that "the set-point c (being) a rational function of the biochemical rate constants ... would be the case for any steady-state" – we agree completely with the reviewer on this point. All variables of the CRN would certainly achieve *some* steady-state that is a rational function of parameters; but *not all variables* would achieve the **same** rational function of parameters *across all possible steady-states* of the system. By noting that the setpoint is a rational function of parameters, we do not intend to convey that the steady-states of **other** variables are **not** rational functions of parameters. We are simply pointing out the mathematical nature of the setpoint (which can ultimately be computed automatically via the algorithm we propose) for the benefit of the general reader, to whom such properties may not be obvious. We did not intend this to be a major point.

- **Algorithm may not terminate:** *The Gröbner basis algorithm to find the RPA polynomial may not terminate. It is mentioned that failure to terminate for a chemical reaction network (CRN) is a prima-facie evidence that the CRN does not exhibit RPA. However this is not mathematically shown. In particular, no proof is given showing that non-termination implies no RPA. Moreover, non-termination of a method is not something that can be ascertained with any certainty by running the code. Accordingly, the authors cannot claim that they have a full 'characterisation' of the RPA property.*

Our response: We thank the reviewer for this comment and for giving us the opportunity to provide greater clarity on these points in our revised manuscript.

First, we respectfully point out that Buchberger's algorithm, which computes Gröbner bases, *always* terminates. This is because *all ideals* of a polynomial ring (or any Noetherian ring, for that matter) are finitely generated – a result of central importance in algebraic geometry, formalized by the Hilbert Basis Theorem. As a consequence, we are assured that Buchberger's algorithm will terminate in a finite number of steps. The problem, of course, is that the finite number of steps required could, and often will, be unmanageably large! The computational demands of Buchberger's algorithm are well known, and we do acknowledge in our paper (in both our original and revised submissions) that the problem of determining whether a **general** collection of polynomials constitutes a Gröbner basis (also known as the Gröbner basis detection (GBD) problem) is **NP-Hard**. The computational demands of Buchberger's algorithm are clear from an examination of the algorithm itself; the demands are even greater when a lexicographic monomial ordering is used since this necessarily implements a computationally expensive polynomial division process in comparison with other monomial orders. The complexity class of the GBD has been well known since the 1990s (due mainly to the work of Bernd Sturmfels and others), and has been widely discussed in the literature.

We respectfully point out that our study does, in fact, make clear why RPA-capable CRNs have a special property which guarantees that Buchberger’s algorithm will terminate in polynomial time *if the CRN does indeed exhibit RPA*. Indeed, we demonstrate that the mathematical transformation required to obtain the RPA polynomial from the reaction rate equations is *almost linear* in the sense that *only \mathbb{R} -linear combinations* of these equations are required to produce the ‘subsidiary’ polynomial invariants which are then ‘concatenated’ by monomials. And as long as only *\mathbb{R} -linear combinations* of the polynomials are required, the syzygy polynomials (“S-polynomials”) that are computed at each step of Buchberger’s algorithm are precisely the linear row combinations computed during Gaussian elimination (a polynomial-time algorithm). Indeed, for the simple CRNs considered by Cappelletti et al. [2], for instance, where the RPA polynomials of the CRNs were in the rowspan (no concatenating monomials required), then Buchberger’s algorithm *is*, in fact, Gaussian elimination. Those authors present an efficient method to calculate which particular linear combination of the reaction rates is required to ‘achieve’ this RPA polynomial. But it is clear that for *any* CRN with an RPA polynomial *in the rowspan* of its reaction equations, the RPA polynomial itself can be obtained algorithmically by Gaussian elimination. In particular, if the equations are expressed in the matrix form $dX/dt = Ax$, where x is the vector of monomials, and the two monomials of the candidate RPA polynomial are listed last (at the bottom of the vector), then performing Gaussian elimination on A will result in an echelon form in which the bottom non-zero row has exactly two entries (corresponding to the two coefficients of the constituent terms of the RPA polynomial). Which linear combinations are required to achieve the rows of this echelon form can be stored during the execution of Gaussian elimination (producing an LU-decomposition, for example).

This process readily extends to the more general (nonlinear, but ‘almost linear’) case. The row combinations (corresponding to construction of S-polynomials) that are calculated during the execution of Buchberger’s algorithm may be stored, and retrieved subsequently using the ‘LIFT’ command in Singular. We demonstrate this in the detailed annotations to our code in Section S5. Where only \mathbb{R} -linear combinations of the CRN equations are required to identify the RPA polynomial (as is for the cases considered by Cappelletti et al. [2]), the coefficients retrieved by LIFT will contain only ‘constants’ (polynomial functions of the CRN parameters). If the RPA polynomial is not in the rowspan, the row combinations retrieved by LIFT will contain (only) the concatenating monomials required to combine adjacent ‘complex linear invariants’ (subsidiary polynomials corresponding to topological features of the underlying flow of biochemical information in the CRN), in addition to the constants involving CRN parameters.

In reflecting on the reviewer’s feedback, we have made extensive revisions to our paper to ensure that the technicalities surrounding the computational complexity of Gröbner basis computations for RPA-capable networks are completely clear and explicit. We now make extensive clarifications on these points throughout the SI. In Remark 3 following our proof to Theorem 1, for example, we explain that

“...With a well-defined algebraic elimination problem now at hand, with two candidate variables selected for the geometric projection, the existence of an RPA polynomial as the sole generator of the relevant elimination ideal can always be tested *in principle* via the computation of a suitable Gröbner basis [10]. Unfortunately, the problem of computing Gröbner bases (e.g. via Buchberger's algorithm) for **general polynomial systems** is well known to be NP-Hard [refs], so without any special simplifying structure to the collection of polynomials, we can have no assurance that any Gröbner basis-computing algorithm will terminate on a practical time-frame. However, we will show in the Sections to follow that RPA-capable CRNs **do indeed** have a special structure that allows the relevant Gröbner basis to be computed rapidly, on a time-scale comparable to Gaussian elimination. This is because the fundamental ‘building blocks’ of the RPA polynomial (i.e. its *subsidiary polynomial invariants*), *always* reside in the *rowspan* of the CRN reaction equations (see Sections S3 and S4), and the RPA polynomial can always be constructed from these by multiplying each by a ‘concatenating monomial’ (where necessary, depending on the stoichiometric dependence of the invariants, see Section S3), and adding the resulting polynomials. As a consequence, the most computationally-demanding component of the resulting algorithm for an RPA-capable CRN (cf. general polynomial systems) is the computation of these subsidiary polynomial invariants (also known as ‘complex linear invariants’ [8]) from the CRN reaction equations. Note, in particular, that where only \mathbb{R} -linear combinations of the rows of $YL(G)$ are computed, the S-polynomials of Buchberger's algorithm are identical to the row combinations obtained during Gaussian elimination (applied to the matrix $YL(G)$). Gaussian elimination is well-known to be a polynomial-time algorithm, which terminates rapidly even for relatively large numbers of variables (or complexes, in the case of CRNs).

Later, in Section S4, before providing the detailed mathematical development of our approach for computing RPA-relevant polynomials in the rowspan of the rate equations - for low deficiency CRNs ($\delta = 0, 1$) in the first instance, and then for CRNs of arbitrary deficiency ($\delta > 1$) – we explain that:

“ ... it is only on account of this special ‘almost linear’ coordinate change that the feasibility of this approach is guaranteed: as we noted in our remarks following Theorem 1, the *general* problem of computing Gröbner bases, for *general* collections of polynomials, is known to be NP-Hard. But the fact that the RPA polynomial can always be computed by combining a collection of polynomials *in the rowspan of the system* means that the algorithm that extracts the RPA polynomial from the CRN reaction equations does not differ substantially from Gaussian elimination. Moreover, even for an arbitrary choice of two projection variables, our statement of Theorem 1 offers opportunities for additional computational efficiency when computing a suitable Gröbner basis insofar as only *one* particular elimination ideal is required (i.e. that comprising the two chosen variables). Without prior knowledge of this particular property of the RPA polynomial, a full set of *all* elimination ideals for the system would be required, which necessitates an expensive lexicographic monomial ordering in the execution of the algorithm. With only one elimination ideal required, a more efficient block ordering may be chosen, with a comparatively

fast (e.g. degree reverse lexicographic) ordering imposed on all but the two chosen variables.”

In addition, we provide much more extensive explanations throughout our SI of the mathematical technicalities underpinning this ‘almost-linear’ coordinate change that can always extract the RPA polynomial from the chemical reaction rates of an RPA-capable CRN. The reviewer is quite right to be astounded by this special property of CRN-capable networks. It is truly remarkable (and entirely non-obvious) that the requisite mathematical structure of chemical reactions that are compatible with RPA should be characterised by such a simplicity as to obviate the computational ‘black holes’ that can exist for *general polynomial systems*, including CRN models that do **not** exhibit RPA.

- **Form of the RPA polynomial:** *The whole approach hinges on the RPA polynomial having the form*

$$\rho = g(x, y)(x - c)$$

where $g(x, y) \neq 0$ and y is a non-RPA capable variable which forms the kinetic pair to x . It is unclear why $g(x, y)$ can only be a function of one additional variable, apart from the output variable. The proof given on page 12 in the Supplement does not satisfactorily explain why the ideal $I_f \cap \mathbb{R}[\bar{x}]$ will contain polynomials in x_j and x_m that are not in $I_f \cap I_p$.

As an example consider the following network:

Furthermore assume that species $X_2 \dots X_5$ participate in several reactions that do not involve X_1 but can be catalysed by it. In this scenario the RPA polynomial would be $\rho = x_2 x_3 \left(\frac{c_1}{c_2} - x_1 \right)$. Hence the function g depends on two non-RPA variables x_2 and x_3 . Please explain how this example is consistent with the form of $g(x, y)$ stated above.

Our response: Let us first respond to the reviewer’s question regarding the proposed network, and its relationship to the two-variable requirement, before returning to the reviewer’s concerns about our proof to Theorem 1.

The reviewer proposes two chemical reactions, and adds that we should also ‘assume that species $X_2 \dots X_5$ participate in several reactions that do not involve X_1 but can be catalysed by it’. Indeed, in the specific reactions proposed by the reviewer, X_2 and X_3 are only consumed, not produced, whereas X_4 and X_5 are only produced, not consumed; thus, there must be some additional reactions involving these four variables, so that X_2 and X_3 can be produced and X_4 and X_5 consumed, giving the CRN the *potential* to achieve a steady-state in the positive orthant.

Let us consider several possible ways to ‘close’ the reaction set (i.e. add in reactions that produce X_2 and X_3 and consume X_4 and X_5). First, let us consider two additional closure reactions of the simplest possible form:

We emphasize that these are not suggested to be the ‘best’, or most biologically meaningful, or realistic, closure reactions, only that they constitute a simple solution that allows us to proceed in the first instance. Let x_1, x_2, x_3, x_4, x_5 be the concentrations of the five species. In this case, the mass-action equations for this collection of four reactions contain a **boundary variable**, x_2x_3 – a concept which we clearly and prominently define (“Definition 1”) at the beginning of Section S1.5 in our SI, in the context of defining what we mean by a **variable**, before proceeding to invoke this terminology in our statement and proof of Theorem 1 (the two-**variable** kinetic pairing theorem). Indeed, the rate equations that are induced by this collection of four reactions are:

$$f_1 = c_1x_2x_3 - c_2x_1x_2x_3, \quad (\text{Eq.1})$$

$$f_2 = -c_1x_2x_3 - c_2x_1x_2x_3 + c_3x_4, \quad (\text{Eq.2})$$

$$f_3 = -c_1x_2x_3 - c_2x_1x_2x_3 + c_4x_5, \quad (\text{Eq.3})$$

$$f_4 = c_1x_2x_3 - c_3x_4, \quad (\text{Eq.4})$$

$$f_5 = c_2x_1x_2x_3 - c_4x_5. \quad (\text{Eq.5})$$

In any case, it is clear from these equations (and, indeed, clear from the corresponding reactions themselves) that x_2x_3 is a boundary variable for this CRN, since both x_2 and x_3 **only** appear in the monomials of these rate equations in the form of the factor x_2x_3 . (In other words, neither species occurs either alone or contributes to a monomial that also contains some *other* species (e.g. x_2x_4 or x_5x_3 , say). This CRN thus comprises five species but four variables: $\{x_1, x_2x_3, x_4, x_5\}$. But in this particular CRN, the boundary variable technicality happens to be a moot point since this CRN is **not** capable of RPA (or ACR). This is because $\langle f_1, f_2, f_3, f_4, f_5 \rangle \cap \mathbb{R}[x_1, x_2x_3] = \langle x_2x_3 \rangle$. In other words, the generator of the ideal consisting of all polynomial consequences of f_1, f_2, f_3, f_4 and f_5 containing only the variables x_1 and x_2x_3 **is not an RPA polynomial**. Thus, **for all steady-states** of the system, regardless of parameter choices or initial conditions, $x_2x_3 = 0$, and RPA is thereby not imposed on x_1 , despite the fact that one of the reaction rates (f_1 in this instance) **is** an RPA polynomial. In fact, our Theorem 1 makes this important subtlety crystal clear: as counterintuitive as it may seem, it is **not enough** for an RPA polynomial to be *one of the reaction rates*, or be otherwise *contained* in the ideal generated by a CRN’s reaction equations. Rather, an RPA polynomial **must generate** the ideal $I_f \cap \mathbb{R}[x_i, x_j]$ **and** this ideal **must** be **principal** (i.e. the RPA polynomial must be the **sole** generator of $I_f \cap \mathbb{R}[x_i, x_j]$) in order for the CRN to exhibit RPA. Just to emphasize, $I_f \cap \mathbb{R}[x_i, x_j]$ constitutes the projection of the ideal $\langle f_1, \dots, f_n \rangle$ onto **two** variables (again, *being mindful of how we define a variable*) - the putative RPA variable along with one non-RPA variable.

Although we carefully define and explain the key concept of a boundary variable (see Definition 1 and ensuing discussion in Section S1.5), and elaborate on the concept again in Example 2 in Section S3.2, where we carefully explain that the CRN in question has nine species but eight variables due to the presence of a boundary variable (see discussion after Equation (34)), we recognise that these points must not have been sufficiently clear to the reviewer in our original submission. In our extensively revised SI, we have now provided the reader with a brief overview of the reviewer’s CRN example in order to further highlight and explain the concept of a boundary variable in Section S1.5. We have also supplemented our explanations with the alternative terminology “power product” (another word for monomial) just to be absolutely sure that the concept will be absolutely clear to the general reader. We thank the reviewer for providing such a simple and clear example of a CRN that contains a boundary variable to enhance our explanations of this key concept.

Please allow us the opportunity to expand even further on the subtleties that RPA imposes on the detailed underlying structures of CRNs via some additional examples, since the reviewer has been kind enough to offer such a fruitful and illuminating CRN example. In particular, let us now consider two other possible alternatives for closing the reaction set provided by the reviewer. The reviewer specifically suggests a scenario where the added reactions involving $X_2 \dots X_5$ “do not involve X_1 but can be catalysed by it”. In line with this suggestion, let us suppose that either one, or possibly both, of the two additional reactions include X_1 as both a reactant and a product, thereby endowing it with the properties of a catalysing enzyme that is neither produced nor consumed by the reaction. (We assume this is what the reviewer means by not involving X_1 “but can be catalysed by it”? The wording is not 100% clear as to what the reviewer actually had in mind here. We apologise if we have misunderstood the reviewer’s intentions.)

Suppose, then, that the reaction associated with rate constant c_3 is now replaced by

This altered c_3 reaction leaves f_1 , f_3 and f_5 unaffected, but f_2 and f_4 now become:

$$f_2 = -c_1 x_2 x_3 - c_2 x_1 x_2 x_3 + c_3 x_1 x_4, \quad (\text{Eq.6})$$

$$f_4 = c_1 x_2 x_3 - c_3 x_1 x_4, \quad (\text{Eq.7})$$

We observe that $x_2 x_3$ is still a boundary variable for this altered CRN. When we project the ideal generated by f_1, f_2, f_3, f_4, f_5 (using the updated versions of f_2 and f_4) onto x_1 and $x_2 x_3$, we still obtain $\langle x_2 x_3 \rangle$. Thus, the CRN is again thwarted in any attempt to impose RPA on x_1 , since $x_2 x_3 = 0$ at **all** steady-states for this CRN (regardless of parameters or initial conditions).

Finally, let us consider the possibility that one of the species that appears in the boundary variable (say X_2 , for argument’s sake) catalyses one of the additional closure reactions, rather than X_1 . In this case, the reaction associated with rate constant c_3 is replaced by

and f_2 and f_4 now become:

$$f_2 = -c_1 x_2 x_3 - c_2 x_1 x_2 x_3 + c_3 x_2 x_4, \quad (\text{Eq.8})$$

$$f_4 = c_1 x_2 x_3 - c_3 x_2 x_4. \quad (\text{Eq.9})$$

In this case, $x_2 x_3$ is no longer a boundary variable since x_2 now also features in the monomial $x_2 x_4$. Thus, this CRN comprises five variables (all species): $\{x_1, x_2, x_3, x_4, x_5\}$. So, to test for RPA, we now attempt to project onto x_1 and x_2 . In this case, $\langle f_1, f_2, f_3, f_4, f_5 \rangle \cap \mathbb{R}[x_1, x_2]$ contains only zero (the additive identity for the ring). As expected, the ideal $\langle f_1, f_2, f_3, f_4, f_5 \rangle \cap \mathbb{R}[x_1, x_3]$ also contains only zero. Thus, the CRN remains incapable of RPA. If, instead, we attempt to compute $\langle f_1, f_2, f_3, f_4, f_5 \rangle \cap \mathbb{R}[x_1, x_2, x_3]$, projecting onto the *three* variables that appear in the apparent RPA polynomial in the f_1 reaction rate, we find that this elimination ideal is given by $\langle x_2 x_3 \rangle$. This confirms that for this CRN, either x_2 or x_3 (or both) is zero for **all possible steady-states** of the system. Therefore, this CRN fails to exhibit RPA (at x_1 or otherwise).

We warmly invite the reviewer to verify for himself/herself that the three versions of this particular CRN cannot, under any parametric circumstances, exhibit RPA. To assist, we have simulated the second version of the CRN discussed above for 10,000 randomised parameter sets, $c_i \in (1,100)$, and initial conditions, $x_i(0) \in (1,100)$, see Figure below. From these histograms, the significance of having one rate equation (f_1) in the form of an RPA polynomial is clear: If, for a specific random choice of parameters and initial conditions, x_1 reaches the value c_1/c_2 before x_2 or x_3 reaches zero, then the x_1 -coordinate stops evolving and x_1 does attain the 'apparent' setpoint at steady-state. But if either x_2 or x_3 reaches zero first (before x_1 reaches c_1/c_2), then x_1 remains at whatever value it reached at the moment x_2 or x_3 reaches zero. Thus, the steady-state value of x_1 is a matter of chance, dependent upon the vagaries of parameters in comparison with initial conditions. There is a bias in the steady-state value for x_1 , with mode of the distribution at c_1/c_2 , but x_1 can achieve any value in principle at steady-state. Thus, the CRN does not exhibit RPA.

Figure: Steady-state concentrations for x_1 , and for $x_2 x_3$, obtained from 10,000 simulations of the second of the three CRNs discussed above. Simulations for the other two CRNs are qualitatively very similar. For each simulation, rate constants (c_1, \dots, c_5) and initial conditions ($x_1(0), \dots, x_5(0)$) are randomly selected from the interval $(1,100)$. We gratefully acknowledge the assistance of my (RA's) graduate student, Cailan Jeynes-Smith, in the preparation of this Figure.

The fact remains that our Theorem 1 provides a definitive and universal condition that **must** be satisfied by **any** RPA-capable CRN – no exceptions, not even for CRNs that happen to have an RPA polynomial in the rowspan!! (Interesting, the authors of the Cappelletti et al. paper [2] appear to have overlooked this subtlety). We hope that the reviewer can now begin to appreciate that these remarkable and highly-restrictive conditions on the detailed graph structures of RPA-capable CRNs could never possibly have been gleaned just from our prior work in [1], or by any other prior work (including the work of Cappelletti et al [2]). All three examples we analyse above are **incapable** of RPA since they violate the conditions specified in our Theorem. Specifically, in each case, the intersection of the ideal generated by the reaction equations with a suitable polynomial ring in two variables (ie. one that comprises *one* RPA variable and *one* non-RPA variable, *once again being mindful of what we mean by a variable*), is not **generated** by a *single polynomial that takes the form of an RPA polynomial*. The fact that each CRN induces a reaction rate (f_1) that happens to be in the form of an RPA polynomial is (surprisingly, perhaps) **irrelevant**.

We warmly encourage the reviewer to consider other chemical reactions to add to the pair he/she has proposed. Unlike the Cappelletti et al. [2] study (for example), our study goes far beyond simply ‘testing’ for the RPA capacity of particular CRN examples (and identifying integral variable(s) if the test is successful). Indeed, our article clarifies in detail (and even more explicitly and incisively now due to our extensive revisions, thanks to the reviewer’s helpful feedback and suggestions) how

the **deficiency** of an RPA-capable CRN relates to topological features of the CRN, and concretely delineates the role of this key algebraic invariant in controlling the formation of RPA-relevant ‘complex linear invariants’ in the CRN’s rowspan. For this reason, we can go much further than simply proving that the three specific examples discussed above (encompassing the reviewer’s suggested reactions) cannot exhibit RPA: The two reactions proposed by the reviewer **cannot** impose RPA on X_1 *no matter which reactions are added to the original two*. This is because the two given reactions are linearly independent, and the CRN formed from these reactions has deficiency zero. Moreover, the two complexes that contribute to the apparent RPA-polynomial in the reaction rate f_1 are not members of the same terminal SCC. It is clear from our discussion of the decomposition of CRNs into algebraically independent subnetworks – a decomposition which is closely tied to the partition of deficiency among independent subsets of reactions (see SI Section S1.3) - as well as our extensive discussions of rowspan polynomials (complex linear invariants) throughout the remainder of our supplement, that *no possible addition* of extra reactions can turn the original two into an RPA-conferring set. We briefly note in Section S4.4.1 (see also footnote in that section) the special case where a deficiency-zero collection of reactions can engender RPA – namely when the two relevant complexes reside in the same terminal SCC, and correspond topologically to a ‘trivial’ (isolated) connector node. The pair of reactions suggested by the reviewer are clearly not of this type. No previously published work – not Shinar and Feinberg [7], not Cappelletti et al. [2], nor any other published work on RPA/ACR – is sufficiently general and all-encompassing as to rigorously and definitively delineate the strict requirements for RPA *in any CRN* – regardless of size or deficiency. We emphasize again, in the most strenuous possible terms, that *precisely none* of these insights follows straightforwardly from our previous study in [1], which gives absolutely no insight whatsoever into viable chemical reaction structures that could impose RPA on any molecule consumed or produced by those chemical reactions.

Having thoroughly addressed the reviewer’s query regarding “*how this example is consistent with the form of $g(x,y)$ stated above*” we now return to the reviewer’s queries about our proof to Theorem 1. The reviewer states that “the proof given on page 12 in the Supplement does not satisfactorily explain why the ideal $I_f \cap \mathbb{R}[\bar{x}]$ will contain polynomials in x_j and x_m that are not in $I_f \cap I_p$.” Just to clarify, what our proof to Theorem 1 actually says is that, given **two** variables x_j and x_m that **do not** exhibit RPA, in addition to a variable x_i that **does** exhibit RPA, then $I_f \cap \mathbb{R}[\bar{x}] = I_f \cap \mathbb{R}[x_i, x_j, x_m]$ will contain polynomials in x_j and x_m that are **not** contained in $I_f \cap I_p$ (the ideal consisting of all polynomial consequences of the rate equations for the CRN that also vanish at $x_i = c$, the setpoint for x_i). The reviewer is not specific as to why he/she feels that this technicality unclear, but we note first and foremost that should $I_f \cap \mathbb{R}[x_i, x_j, x_m]$ contain a polynomial in x_j and x_m **only** (not also including x_i), then this polynomial is *ipso facto* not in I_p (since all polynomials in I_p contain x_i by supposition). We also emphasize that the ideal I_f consists of **all polynomials** in the ring $\mathbb{R}[x_1, \dots, x_n]$ that *vanish* on the variety $V(f_1, \dots, f_n)$. In any case, our theorem is fundamentally concerned with which

projections (i.e. choices for \bar{x}) are guaranteed to be *entirely contained* within the ideal $I_f \cap I_p$.

Our proof to Theorem 1 succinctly points out that if we project the polynomial consequences of the system (I_f) onto three variables, **two** of which (x_j, x_m) **do not** exhibit RPA, then there will be a *generator* for the associated elimination ideal involving only x_j and x_m (and not x_i – the RPA-capable variable). The reason for the existence of such a generator is that a system whose steady-states can be regulated by any *number of independent* disturbances (or input stimuli) must be able to adapt to each disturbance individually, corresponding to a single degree of freedom. Once the value of any one non-RPA variable (x_j , say) is specified (as a result of setting the magnitudes of the various possible disturbances), then the value of *any other* non-RPA variable (e.g. x_m) is thereby also specified. The value of x_i , by contrast, is not determined through an alteration to any single degree of freedom (corresponding to one of the independent disturbances) since it exhibits RPA, and is independent of these disturbances. Thus, it follows that some polynomial involving the two non-RPA-variables x_j and x_m is **a** generator of $I_f \cap \mathbb{R}[x_i, x_j, x_m]$.

Now if, for the sake of argument, there were a single generator of $I_f \cap \mathbb{R}[x_i, x_j, x_m]$ containing all three variables, this would mean that the value of x_m could only be determined once the values of *both* x_i and x_j were specified. On the other hand, if the generator(s) of $I_f \cap \mathbb{R}[x_i, x_j, x_m]$ were to contain a polynomial in just x_j , then this would imply that the univariate polynomial either has no positive real roots, or possibly no real roots at all (both of which scenarios corresponding to the non-existence of a steady state for x_j), or has at least one positive real root whose value is a rational function of the system rate constants (a scenario that corresponds to RPA in x_j). All of these conditions violate the assumption that x_j is a non-RPA variable – a contradiction.

Of course, Theorem 1 could not possibly be true in general if variables are simply taken to be species. We point out in the preamble to Theorem 1 that in most cases, the variables of a CRN *are* the species, however should a boundary variable exist, then this must be accounted for in the identification of variables for the CRN (since the component species of the boundary variable *cannot* be considered variables in that case).

We hope these clarifications have been able to assure the reviewer of the correctness of this Theorem and its proof. We have now added a condensed version of our explanations above as a ‘Remark’ following the proof to Theorem 1 (and following Definitions 2 and 3). We also decided to remove the reference to ACR in the statement of the theorem (and hence in the proof), and instead added a comment on this point to the list of Remarks following the Theorem. (The reviewer’s helpful suggestion to give a clear and explicit definition for RPA at the outset now makes it unnecessary to add in descriptions of ACR, and its relationship to the more general case of RPA, to our statement and proof of Theorem 1. We cannot thank the reviewer enough for such supportive advice). We do, of course, recognise that not

all mathematical details of Theorem 1 could possibly be obvious, or clear, to the entire readership of a multidisciplinary journal. It is for this reason that we provide many illustrative examples throughout our SI to demonstrate how the fundamental mathematical principles encapsulated by Theorem 1 are reflected in a range of simple RPA-capable CRNs.

Now to the reviewer's next concern:

Moreover, it is not immediately clear how existence of multiple opposer/balancer modules translates into existence of corresponding RPA polynomials. This has not been explained in sufficient detail.

Our response:

We thank the reviewer for this very helpful advice that we make a clearer connection between the existence of multiple topological modules and the corresponding existence of associated RPA polynomials.

Due to our extensive revisions, our manuscript (and especially our SI) now makes this connection very explicit, and explains the underlying technicalities in considerable detail. In particular, our extensive explanations on the decomposition of RPA-capable CRNs into algebraically-independent subnetworks underscores the fact that such independent subnetworks can be analysed separately as to their RPA-capacity (or otherwise), and that the RPA-capacity of 'multi-modular' networks corresponds to the existence of multiple *independent* RPA-capable subnetworks within such a decomposition. We state explicitly in Section S4.2, for example, as we begin to introduce the mathematical process by which higher deficiency ($\delta > 1$) CRNs can be accommodated within our framework, that

*"... for CRNs for which multiple independent subnetworks contribute independently to RPA, these independent subnetworks correspond to distinct **modules** from a topological perspective (see [1])."*

We also emphasize again in Section S4.5, where the general principles are summarised, that

*"...It is clear from the analysis in the preceding sections that a CRN can be decomposed into **lower deficiency** independent subnetworks corresponding to the individual modules of a multi-modular network **and, in the case of Opposer modules, the controller portion of the module,** which can be analysed separately."*

Our study thus makes it clear that each topological module, which directly corresponds to an algebraically independent subset of the CRN reactions, engenders its own RPA polynomial, specific to that module/subnetwork. Nevertheless, Theorem 1 guarantees that for a multi-modular CRN, choosing one RPA-capable variable (associated with *any* RPA-conferring module) and one non-RPA-capable variable (from *any* module, including one that is distinct from the one containing the RPA-capable variable), and projecting the **entire** system onto those two variables,

will yield an RPA polynomial containing those two variables. Thus, Theorem 1 is valid for *any* choice of two variables with the property that one is RPA-capable and one is non-RPA-capable.

We do acknowledge that in our discussion of the topological consequences of Theorem 1 in Section S3 (see for example Figure S4, and Figure 4 in the main paper), we are referring primarily to RPA in a CRN corresponding to a single module. This is mostly from the point of view of (i) highlighting the existence of an overarching mechanism by which a non-RPA capable variable *ultimately* imposes RPA on a variable, *not necessarily directly*; and (ii) distinguishing the (topological) significance of an RPA-capable variable being a *regulating*, or a *regulated*, variable.

- **Connection to existing works:** In [4] the authors consider RPA systems that are maximally robust and find simple linear-algebraic structural conditions that characterise this property in both deterministic and stochastic settings. How do the results in this paper connect to the results in [4]?

Our response:

We are grateful to the reviewer for introducing us to such an interesting and impressive recent article [4], of which we were hitherto unaware.

An obvious point of difference between this article [4] and ours is that these authors also consider RPA in the stochastic setting, while our study is focused squarely on the deterministic setting in the interests of delineating the full space of RPA-capable CRNs in the greatest possible generality. Beyond this, the article by Gupta and Khammash focus on a particular type of RPA they refer to as *maximal RPA*, or “maxRPA”, where ‘the setpoint for the key output variable depends on the least number of network parameters, and is insensitive to all the others’ – arguing that ‘it makes strong evolutionary sense for intracellular RPA networks to attain (such a) maximal robustness’. In this connection, the authors claim to ‘work with a stronger notion of robustness’ and that their ‘characterising conditions for RPA networks are more restrictive than the topological requirements expounded in (our previous work [1], for example)’. The study is able to identify, concretely, a number of mathematical conditions required for the existence of the maxRPA property, which includes (among other requirements) that the setpoint should be a function of (only) **two** biochemical parameters. In any case, the authors succeed in their goal to ‘mathematically characterise and study the structure of biomolecular networks that constitute such maximal RPA (or maxRPA) systems’ – a most impressive achievement – and propose a novel *internal model principle* (IMP) pertaining to these maxRPA networks.

This focus on maxRPA networks alone constitutes the major distinction between [4] and our study, since our goal is to characterise **all possible** RPA-capable CRNs – as opposed to specific classes of RPA-capable CRNs (e.g. the maxRPA class [4], the deficiency-one Shinar-Feinberg class [7], the class containing a ‘linear constrained integrator’ [2], etc). As we noted earlier, this generality is important in the context of understanding, at an abstract level, the ability of RPA-capable networks to cope

with new disturbances (i.e. those that were not present in the original CRN – e.g. the addition of small-molecule enzyme-inhibitors). This is particularly crucial for signalling networks for which the detailed structures of the underlying CRNs are unknown (in the context of signal transduction in metazoan cells, for instance). Moreover, we respectfully question the authors' point of view on the suggested evolutionary advantages of a setpoint involving only two network parameters. We contend that it is not the *number* of parameters defining the setpoint, but the *nature* of the parameters that is most significant to the functional robustness of the system, and ultimately, the survival of the cell/organism. A CRN could have a setpoint determined by a hundred parameters, but if all these parameters are 'stable' in the sense that they are determined entirely by the chemical properties of the interacting molecules (e.g. association/dissociation constants, or catalytic constants), alterable only by mutation or by factors such as temperature that vary very slowly on the timescale of adaptation, then this may still constitute an entirely useful form of RPA from a biological standpoint. By contrast, a CRN could have a setpoint determined by only two parameters (the minimum, as in the antithetic integral control motif), but if these are highly labile parameters relating to, for example, the rate of synthesis of a particular molecule, or the total expression level of a particular molecule, then this could potentially be problematic.

Consider, for instance, the deficiency-two Shinar-Feinberg model of the EnvZ-OmpR motif. The setpoint for *pOmpR* due to this CRN involves **eleven** parameters (calculated from two independent setpoints arising from the balancer and connector contributions). But all of these parameters are either association/dissociation constants for pairs of molecules, or catalytic constants, and are thus highly stable in magnitude. By contrast, for the antithetic control mechanism, the setpoint for the sensor molecule involves just two parameters. But both of these parameters capture the rate of synthesis of an antithetic molecule which could, in principle, vary quite significantly on the timescale of adaptation (depending on the regulatory mechanisms governing the transcription and translation of the molecules in question).

Moreover, perturbations to the setpoint that are either stable (e.g. due to a mutation in an RPA-relevant signalling protein) or vary on a timescale that is significantly longer than the timescale of adaptation, may not alter the functionality of the biological system whatsoever. One of the most widely-discussed examples of unaltered biological function in the face of a stably varying setpoint is the robustness of tumbling frequency, and precision of adaptation, in *E.coli* chemotaxis (see Alon et al. [12]). In that study [12], genomic alterations that affected the expression levels of various chemotaxis proteins were introduced, which stably altered the setpoint for the steady-state tumbling frequency as well as the adaptation time. But exact adaptation was still observed (albeit with an altered setpoint), and the chemotaxis response of the cells was unaffected. In this context we recognise that, for single-celled organisms, significant temperature variations could certainly alter biochemical parameters; but provided as temperatures vary slowly in comparison with adaptation (RPA), this might not affect the functionality of the organism at all.

Moreover, aside from what type of setpoints are most evolutionarily advantageous, we respectfully point out that the RPA responses frequently observed in highly complex signalling networks, such as cancer signalling networks in human cells (on the basis of time-course data – see [13], for instance) are not necessarily attributable to maxRPA networks; there is simply no evidence to support this. In fact, it is currently entirely unclear *what* CRN architectures are generally responsible for RPA in complex signal transduction networks (in either normal or diseased states).

We hope the reviewer will not object to a friendly and respectful debate on these points, in the interests of moving scientific frontiers forward. We contend that, to make sense of the highly complex signalling networks that arise in nature, it is absolutely essential to have access to a completely general description of the mechanisms by which RPA could be implemented in CRNs. This is the brass-ring objective that we achieve in the present work.

In raising all these points, we are by no means dismissing the study by Gupta and Khammash [4], which is incredibly interesting, and extremely impressive (mathematically and otherwise); we only point out that it is not completely *general* vis-à-vis RPA-promoting CRN structures (and makes no claims to be, since its focus is on the particular class of RPA-capable CRNs known as ‘maxRPA’). It is a beautiful paper, and we are proud to include it in our reference list in the main manuscript.

Concluding Comments to Reviewer #1: We sincerely thank the reviewer again for such a generous investment of time and painstaking effort, and for his/her many supportive and helpful suggestions for making our scientific goals, and our findings, much clearer and more explicit. We are truly grateful to have this opportunity to defend our work, and to provide an extensively revised version of our manuscript that offers a much more comprehensive exposition of all the relevant supporting technicalities, and a much more compelling presentation for a general scientific audience.

References (for response to Reviewer 1)

- [1] R. P. Araujo and L. A. Liotta. The topological requirements for robust perfect adaptation in networks of any size. *Nature communications*, 9(1):1-12, 2018.
- [2] D. Cappelletti, A. Gupta, and M. Khammash. A hidden integral structure endows absolute concentration robust systems with resilience to dynamical concentration disturbances. *Journal of the Royal Society Interface*, 17(171):20200437, 2020.
- [3] G. Craciun, A. Dickstein, A. Shiu, and B. Sturmfels. Toric dynamical systems. *Journal of Symbolic Computation*, 44(11):1551-1565, 2009.
- [4] A. Gupta and M. Khammash. Universal structural requirements for maximal robust perfect adaptation in biomolecular networks. *bioRxiv*, 2022.
- [5] M. Perez Millan, A. Dickstein, A. Shiu, and C. Conradi. Chemical reaction systems with toric steady states. *Bulletin of mathematical biology*, 74(5):1027-1065, 2012.
- [6] A. Sadeghimanesh and E. Feliu. Gröbner bases of reaction networks with intermediate species. *Advances in Applied Mathematics*, 107:74-101, 2019.

- [7] G. Shinar and M. Feinberg, M. Structural sources of robustness in biochemical reaction networks. *Science*, 327(5971): 1389-1391, 2010.
- [8] R. L. Karp, et al. Complex-linear invariants of biochemical networks. *Journal of theoretical biology*. 311: 130-138, 2012.
- [9] M. Feinberg. *Foundations of chemical reaction network theory*. Springer, 2019.
- [10] D. Cox, J. Little, and D. OShea. *Ideals, varieties, and algorithms: an introduction to computational algebraic geometry and commutative algebra*. Springer, 2013.
- [11] J.M. Eloundou-Mbebi, et al. A network property necessary for concentration robustness. *Nature communications*, 7(1): 1-7, 2016.
- [12] U. Alon, et al. Robustness in bacterial chemotaxis. *Nature*, 397(6715): 168-171, 1999.
- [13] A.J. VanMeter, et al. Laser capture microdissection and protein microarray analysis of human non-small cell lung cancer: differential epidermal growth factor receptor (EGFR) phosphorylation events associated with mutated EGFR compared with wild type. *Molecular & Cellular Proteomics*, 7(10): 1902-1924, 2008.

Response to Reviewer #2

Araujo and Liotta develop a set of precise criteria that a system needs to fulfil in order to perfect adaptation. This is remarkable and an extremely valuable contribution to the literature in three ways: (i) the search for "design principles" is taking centre stage in synthetic (but also e.g. developmental systems) biology; the first author has proven a leader in distilling precise mathematical criteria for robust perfect adaptation, and I hope that this study will inspire more work in this area, especially in considering the mathematical properties of chemical reaction networks to guarantee different types of behaviour; (ii) in Mathematical biology there are few instances where the mathematical statements can be so precise that biology will have to "obey" these statements. This is one such instance. Finally (iii) the maintenance and control of robust adaptation is of considerable and far reaching biological relevance. I like this manuscript for each of these three points a lot.

The most interesting mathematical aspects of the work have, unfortunately but predictably been relegated to the supplementary information. The results in Figure 2, for example, are very clearly explained and easy to follow in the SI. I personally enjoyed the SI a lot, and while the discussion e.g. in lines 207-215 is clear, it may be too terse for some readers to follow. The SI by contrast was very clear. Fig 3 does a nice job, however, to get the message of kinetic pairing across.

I think it would help readers to understand how translatable to other phenotypes this type of analysis is, or if RPA is particular in this regard of allowing such general mathematical statements to be derived.

Our response:

We are truly indebted to this reviewer for his/her incredible generosity in considering our work in such careful detail, and for such genuinely helpful feedback and thoughtful suggestions for further improvement. We are immensely grateful to have the benefit of an 'extra set of eyes' cast over our work, to identify additional opportunities to make our exposition more compelling. We have noted all revisions to the main manuscript, and to our SI, with red text, to make it easy to identify which parts of our article have been updated since the original submission.

Regarding the issue of relegating much of our detailed mathematical development to the Supplementary Information, it is true that the severe word limits in *Nature Communications* (6000 words for main text, excluding figure captions) have forced us to limit our exposition in the paper itself to an accessible overview of the major conceptual innovations of the study, with extensive references to all the technical details in our Supplementary Information. A significant additional challenge stems from the fact that solving the RPA problem in complete generality, at the level of intermolecular interactions, has required us to draw together a number of distinct mathematical languages – chemical reaction network theory (CRNT), engineering control theory, and algebraic geometry – into a unified framework.

We greatly value the reviewer's insightful suggestion that "it would help readers to understand how translatable to other phenotypes this type of analysis is, or if RPA is

particular in this regard of allowing such general mathematical statements to be derived.” We now include following text at the conclusion of our article:

The quest to uncover the fundamental ‘design principles’ that constrain complex signalling networks in nature to implement biologically important functions is considered to be one of the most important and far-reaching ‘grand challenges’ in the life sciences [1-6]. On the basis of the present study, along with our earlier topological study at the network macroscale [7], RPA currently stands alone as a keystone biological response for which there now exists a universal explanatory framework – one that imposes strict and inviolable design criteria on arbitrarily large and complex networks, and one that now accounts for the subtleties of intricate intermolecular interactions at the network microscale. These universal RPA-permissive design principles now represent a launching-point for future explorations of more complex phenotypes - including some classes of embryonic patterning problems, for instance, where integral control is known to play a role in promoting adaptation of segmentation boundaries to variations in organism size [8,9]. The identification of universal design principles for many other complex phenotypes, such as Turing patterns [4, 10,11] and multistability/switching-responses [12,13], is likely to prove more challenging due in part to the central role of equilibrium stabilities, or instabilities, in these responses. These ‘grand challenges’ remain open, and we hope that our study will inspire bold new mathematical thinking in these vitally important directions.

Small points:

- much of chemical reaction network theory is obscure to most readers. I would like to see a clear and easy definition of integral control that is accessible to non-expert audiences. Deficiency is maybe another such concept and it would be good to define it on line 146.

Our response: We truly appreciate such helpful suggestions to improve the clarity of our exposition. In our newly revised manuscript, we give a thorough yet accessible explanation of the concept of integral control, and the internal model principle (IMP), via the introduction of a new Figure in our Introduction section (Fig. 1 in our revised submission) – see below. We have also carefully edited our explanations of integral control in-text to complement this new Figure, and to ensure that the concept (and its relationship to the goals of our study) are clear to the readership of a multidisciplinary journal.

We have also vastly expanded our clarifications on the concept of deficiency, especially in our Supplementary Information where we give far greater detail on the concept where we first define it in SI Section S1.2, and particularly its relationship to the flow of biochemical information (and ultimately topology) in RPA-capable CRNs. We have also added in an entirely new section (SI Section S1.3) on the key concept of algebraically-independent subnetworks of a CRN (which we didn’t discuss until much later in the Supplement, in our original submission), and clearly outline the role of deficiency in partitioning the reactions of a CRN into such independent subsets. In the main paper, we also introduce the concept of deficiency much earlier

than previously, and point out early that this is a key algebraic invariant which has fundamental (and previously unrecognised) consequences for the implementation of integral control by CRNs. Later in the main paper, where we give further detail on the underlying mathematical technicalities, we give a clear definition of deficiency, and give an example of how to calculate it, in Figure 5.

Fig. 1. The internal model principle (IMP) and its application to RPA-capable CRNs. The class of constant disturbances ($I(t) = \text{const}$) is generally the disturbance class of most fundamental interest to the study of biological systems. (a) In order to exhibit RPA (i.e. ‘adapt’ to constant disturbances), the dynamical system Σ should be decomposable, via a coordinate transformation if needed, into an ‘output-driven internal model’, Σ_{IM} (generating all the constant signals corresponding to solutions of $\dot{z}_2 = 0$), and the remainder of the system, Σ_0 . The variable z_2 thereby computes the integral of the output error. (b) A suitable coordinate change should be able to recast an RPA-capable system into integral feedback form, even if there is no feedback present in the network. As shown, a linear transformation is sufficient to identify an output-driven internal model for this particularly simple incoherent feedforward motif (Balancer module); $y = \alpha_4/\alpha_3$ (setpoint) at steady-state. (c) A model that employs feedback is frequently simpler to recast in ‘integral feedback’ form, with an output-driven internal model; here $y = \alpha_2$ (setpoint) at steady-state. Note that the reaction rates selected for illustrative purposes in (b) or (c) cannot be induced, under the law of mass action, by any CRN²⁰.

- The authors show that their criteria hold for all instances of RPA. I was wondering how easily in practice these mechanisms could be lost? Is it easy to identify points that disrupt RPA?

Our response: With access to a universal and complete solution to the RPA problem in CRNs, we can now delineate, precisely and definitively, the circumstances under which various classes of disturbances, or network alterations, can either preserve the RPA property or cause RPA to fail. We have not commented on this matter in detail in our manuscript, as this is the subject of ongoing work. One of the most interesting classes of disturbances against RPA-capable CRNs is the addition of

enzyme-inhibitors of various types (competitive, non-competitive, mixed, uncompetitive, etc.). Addition of one more enzyme-inhibitors to a CRN, whether highly specific to its target or not, corresponds to the addition of extra linkage class(es) to the CRN. We now know that certain inhibitor mechanisms are deficiency preserving, while others aren't, and their effects on the RPA property (depending on the relationship of their target protein(s) to the overarching structure of the CRN) are complex and counterintuitive, and in some cases very surprising. Generally, the RPA property is very difficult to perturb via molecular-targeted inhibitors, although we have been able to identify very specific and definitive conditions in which this is possible. Other classes of disturbances (such as those we itemize in our newly revised Supplementary Information, at the beginning of Section S1) are much easier to study from the point of view of altering the RPA property. In any case, it is only by having a completely general mathematical description of RPA-permissive reaction structures, that accounts for all possible RPA-capable CRNs, that we can make such definitive conclusions on the preservation or destruction of the RPA property in the abstract. This is particularly valuable in the context of particularly complex signalling networks (e.g. in cancer signal transduction) for which detailed CRN structures are unknown (and possibly will never be known).

In any case, we are planning to publish this work (separately) very soon, and look forward to sharing this.

- there is a vast literature on design principles which could be touched upon at least in passing (limit cycles, multistability, switch-like behaviour, Turing patterns), especially if there is scope for applying similar concepts in these contexts.

Our response: This is such a wonderful suggestion, which we greatly value. As we noted in our first point above, we have added an extra paragraph that comments on this very issue at the conclusion to our manuscript. **"The quest to uncover the fundamental 'design principles' that constrain complex signalling networks in nature to implement biologically important functions is considered to be one of the most important and far-reaching 'grand challenges' in the life sciences [1-6] Etc."**

Concluding Comments to Reviewer #2: We cannot thank the Reviewer enough for such a generous investment of time and effort, for such incredibly supportive feedback, and for so many insightful suggestions for additions and clarifications to our exposition. We are truly grateful.

References (for response to Reviewer 2)

- [1] Alon, U. *An introduction to systems biology: design principles of biological circuits*. (Chapman and Hall/CRC, 2006).
- [2] Green, S. Revisiting generality in biology: systems biology and the quest for design principles. *Biology & Philosophy* **30**, 629-652 (2015).

- [3] Novák, B. & Tyson, J. J. Design principles of biochemical oscillators. *Nature reviews Molecular cell biology* **9**, 981-991 (2008).
- [4] Vittadello, S. T., Leyshon, T., Schnoerr, D. & Stumpf, M. P. Turing pattern design principles and their robustness. *Philosophical Transactions of the Royal Society A* **379**, 20200272 (2021).
- [5] Stumpf, M. P. Statistical and computational challenges for whole cell modelling. *Current Opinion in Systems Biology* **26**, 58-63 (2021).
- [6] Lim, W. A., Lee, C. M. & Tang, C. Design principles of regulatory networks: searching for the molecular algorithms of the cell. *Molecular cell* **49**, 202-212 (2013).
- [7] Araujo, R. P. & Liotta, L. A. The topological requirements for robust perfect adaptation in networks of any size. *Nat Commun* **9**, 1757 (2018).
<https://doi.org:10.1038/s41467-018-04151-6>
- [8] Ben-Zvi, D. & Barkai, N. Scaling of morphogen gradients by an expansion-repression integral feedback control. *Proceedings of the National Academy of Sciences* **107**, 6924-6929 (2010). <https://doi.org:10.1073/pnas.0912734107>
- [9] Eldar, A. *et al.* Robustness of the BMP morphogen gradient in Drosophila embryonic patterning. *Nature* **419**, 304-308 (2002). <https://doi.org:10.1038/nature01061>
- [10] Lander, A. Pattern, growth, and control. *Cell* **144**, 955-969 (2011).
- [11] Krause, A. L., Gaffney, E. A., Maini, P. K. & Klika, V. Introduction to 'Recent progress and open frontiers in Turing's theory of morphogenesis'. *Philosophical Transactions of the Royal Society A* **379**, 20200280 (2021).
- [12] Ullner, E., Zaikin, A., Volkov, E. I. & García-Ojalvo, J. Multistability and clustering in a population of synthetic genetic oscillators via phase-repulsive cell-to-cell communication. *Physical review letters* **99**, 148103 (2007).
- [13] Duddu, A. S., Sahoo, S., Hati, S., Jhunjhunwala, S. & Jolly, M. K. Multi-stability in cellular differentiation enabled by a network of three mutually repressing master regulators. *Journal of the Royal Society Interface* **17**, 20200631 (2020).

Review 2: Universal structures for embedded integral control in biological adaptation

Robyn P. Araujo and Lance A. Liotta

Overview: The aim of this paper is to provide an algebraic characterisation of the hidden integral controller and construct an algebraic procedure to identify it in adaptation-capable networks. It is shown that this procedure is intimately connected to the structure of the networks via the well-known deficiency theory for chemical reaction networks.

Recommendation: I am grateful to the authors for revising the manuscript to address many of the points that I raised in the first review. Even though the paper has significantly improved, many of the more serious concerns remain. These concerns, which are outlined below, should be fully addressed before the paper can be reconsidered again for *Nature Communications*.

1. **Connection to integral control:** The paper claims to uncover universal structures for embedded “integral control”. However in order to successfully demonstrate this, a coordinate transformation (possibly nonlinear) needs to be constructed that gives rise to the “integrator” variable. It does not matter if this coordinate transformation is done in a single step or in multiple steps (as proposed by this paper), but its existence needs to be shown in order to identify the integral mechanism. In the Supplement the authors write

We note that, although there should always exist some single nonlinear coordinate change that yields the RPA polynomial in principle, for any mass action system, identifying such a single coordinate change in practice may be extremely difficult in all but the very simplest RPA-capable CRNs.

I agree that proving the existence of an integrator may be difficult, but unfortunately it is unavoidable for showing integral control. Let us look into this issue in more details.

Suppose the dynamics is governed by a system of ODEs:

$$\frac{dx}{dt} = f(x),$$

where x is the n -dimensional state of the system and $f(x) = (f_1(x), \dots, f_n(x))$ are the rates of evolution of all the species. In order to exhibit an integrator, we need to find a real-valued function $H(x)$ such the dynamics of $z(t) = H(x(t))$ is given by the ODE:

$$\frac{dz}{dt} = \nabla H(x) \circ f(x) = p(x)(x_i - c) := \rho(x), \tag{0.1}$$

where ∇ is the gradient operator, \circ is the usual dot product and $\rho(x) = p(x)(x_i - c)$ is what the authors call the ‘‘RPA polynomial’’ for the RPA variable x_i . Suppose one is able to find nonlinear polynomials h_1, \dots, h_n (as in Theorem 1 of the paper) such that

$$h_1(x)f_1(x) + \dots + h_n(x)f_n(x) = \rho(x).$$

However this does not necessarily mean that a function $H(x)$ can be constructed so that (0.1) holds. Observe that such a function $H(x)$ would need to have the following partial derivatives

$$\frac{\partial}{\partial x_j} H(x) = h_j(x) \quad \text{for } j = 1, \dots, n.$$

Unless each $h_i(x)$ is a constant (in which case $H(x)$ is linear), it is unclear why such a function $H(x)$ would exist. Since the approach in the paper uses nonlinear elimination steps (via concatenating monomials), the existence of $H(x)$ cannot be ascertained. In fact if for a pair j, k we have

$$\frac{\partial}{\partial x_j} h_k(x) \neq \frac{\partial}{\partial x_k} h_j(x)$$

then certainly $H(x)$ cannot be constructed with partial derivatives h_1, \dots, h_n . This discussion shows that while existence of a RPA polynomial may be necessary for integral control, it is not sufficient and hence this property does not characterise integral controllers (a main claim of the paper). This gap between necessity and sufficiency can be bridged if the authors can show that for RPA networks one always has

$$\frac{\partial}{\partial x_j} h_k(x) = \frac{\partial}{\partial x_k} h_j(x) \quad \forall j, k.$$

2. **Questions about the Kinetic Pairing result:** In the previous review round, I had proposed a possible RPA network and asked the authors how it satisfies the Kinetic Pairing result (Theorem 1). The network has two reactions given by

and there may be several other reactions involving species $\mathbf{X}_2, \dots, \mathbf{X}_5$ that can be catalysed by \mathbf{X}_1 but they do not change \mathbf{X}_1 . In the rebuttal letter, the authors consider certain instances of such a network, and claim that in these networks all possible steady states lie on the boundary on the positive orthant (e.g. x_2 or x_3 is zero) and hence the networks are not RPA. In particular, the authors state in their rebuttal letter that:

The two reactions proposed by the reviewer **cannot** impose RPA on \mathbf{X}_1 no matter which reactions are added to the original two. This is because the two given reactions are linearly independent, and the CRN formed from these reactions has deficiency zero. Moreover, the two complexes that contribute to the apparent RPA-polynomial in the reaction rate f_1 are not members of the same terminal SCC. It is clear from our discussion of the decomposition

of CRNs into algebraically independent subnetworks a decomposition which is closely tied to the partition of deficiency among independent subsets of reactions (see SI Section S1.3) - as well as our extensive discussions of row-span polynomials (complex linear invariants) throughout the remainder of our supplement, that *no possible addition* of extra reactions can turn the original two into an RPA-conferring set. We briefly note in Section S4.4.1 (see also footnote in that section) the special case where a deficiency zero collection of reactions can engender RPA - namely when the two relevant complexes reside in the same terminal SSC, and correspond topologically to a trivial (isolated) connector node. The pair of reactions suggested by the reviewer are clearly not of this type.

To test the authors' claim above, I tried to come up with instances of such networks (i.e. (0.2) + some reactions) which are RPA and yet do not have this issue of steady-state not being in the positive orthant. Please explain how the Kinetic Pairing result fits these networks.

I start with a simpler version of this network with only three species (i.e. species \mathbf{X}_4 and \mathbf{X}_5 are absent) where I add inflow and outflow for species \mathbf{X}_2 and \mathbf{X}_3 . So the overall network becomes

This network has no boundary variables and so the variables are the same as species. It seems that this network will indeed show RPA for x_1 with set-point c_1/c_2 . In Figure 1 I plot the simulated dynamics (rescaled to have the set-point of x_1 as 1) for three randomly chosen values of initial states and rate constants, and one can see that RPA holds.

Next I modify this network to have the production of \mathbf{X}_2 and \mathbf{X}_3 catalysed by \mathbf{X}_1 . So the new network becomes

This network also appears to be RPA as shown by the simulations in Figure 2.

Finally, I consider another variant of this network where we have species \mathbf{X}_4 and \mathbf{X}_5 that reversibly bind to each other to produce an inactive compound, and they catalyse the production of \mathbf{X}_2 and \mathbf{X}_3 respectively. Therefore the new network becomes

This network also seems to be RPA, as shown in the simulations in Figure 3.

These networks do not appear to satisfy the RPA characterisation given by this paper. Also consider Example 6.5.3 in [5] which does not seem to fit this result either. These

Figure 1: Plots of the dynamics of network (0.3) with mass-action kinetics, with initial state $x(0)$ and rate constants $c = (c_1, c_2, \dots)$. Each component of $x(0)$ and c was randomly generated from the interval $[1, 10]$ and here we show plots for three realisations of $x(0)$ and c . The corresponding steady-state (unscaled) is stated and note that the plots are shown for the dynamics which is rescaled so that the steady-state value of x_1 is 1.

counter-examples cast doubt on the correctness of the Kinetic Pairing Theorem on which the entire paper rests.

3. **The set-point may not be a rational function of parameters:** In many places in the main paper and the supplement, the system's set-point is said to be a rational function of biochemical parameters. While this holds for most examples, consider the following birth death network

The dynamics is given by

$$\frac{dx_1}{dt} = c_1 - 2c_2x_1^2$$

and so the set-point

$$c = \sqrt{\frac{c_1}{2c_2}}$$

is not a rational function of c_1 and c_2 ? For more examples, see Examples 2.11 and 2.12 in [3]. Please revise as necessary.

Figure 2: Plots of the dynamics of network (0.4) with mass-action kinetics, with initial state $x(0)$ and rate constants $c = (c_1, c_2, \dots)$. Each component of $x(0)$ and c was randomly generated from the interval $[1, 10]$ and here we show plots for three realisations of $x(0)$ and c . The corresponding steady-state (unscaled) is stated and note that the plots are shown for the dynamics which is rescaled so that the steady-state value of x_1 is 1.

4. **Questions on the Buchberger’s Algorithm:** In the previous review round, I had raised the following issue regarding Buchberger’s Algorithm

The Gröbner basis algorithm to find the RPA polynomial may not terminate. It is mentioned that failure to terminate for a chemical reaction network (CRN) is a *prima-facie* evidence that the CRN does not exhibit RPA. However this is not mathematically shown. In any case, checking for non-termination of a method is impractical.

In response the authors state in their rebuttal letter that the termination is guaranteed because “all ideals of a polynomial ring (or any Noetherian ring, for that matter) are finitely generated a result of central importance in algebraic geometry, formalized by the Hilbert Basis Theorem”. While I agree that termination is guaranteed, I must point out that the reason for my confusion and for raising this termination issue is the following excerpt from the previous version of the Supplement (submitted in round 1):

Failure of the Buchberger (or other Grobner basis-computing) algorithm to terminate could thus be adduced as *prima facie* evidence that the CRN under consideration does not, in fact, have the capacity for RPA.

It seems that in this statement that authors meant “terminate in practical time”. I understand that due to the “almost linear” nature of coordinate change, this algorithm

Figure 3: Plots of the dynamics of network (0.5) with mass-action kinetics, with initial state $x(0)$ and rate constants $c = (c_1, c_2, \dots)$. Each component of $x(0)$ and c was randomly generated from the interval $[1, 10]$ and here we show plots for three realisations of $x(0)$ and c . The corresponding steady-state (unscaled) is stated and note that the plots are shown for the dynamics which is rescaled so that the steady-state value of x_1 is 1.

would work well for RPA networks. However, does the final RPA polynomial produced by the method depend on the monomial ordering or other choices made by the method? More importantly, if a network is **not** RPA, how many steps would the method need to confirm this non-RPA property? Such practical considerations must be discussed in the main text, and they are crucial for applying these ideas for characterising RPA in high-dimensional networks.

5. **Connection of examples to existing works:** The authors start the Results section with “two simple examples that have eluded all previous systematic methods to detect RPA”. Please specify which systematic methods are being referred to here. Also, in the example on Figure 2 it should be mentioned that \mathbf{X}_3 is maxRPA and it can be checked from the characterisation result in [2]. Secondly since there is the reaction

which does not involve \mathbf{X}_1 I do not understand why the term $k_6 X_2 X_3$ does not enter the expression for

$$\frac{d(X_1 - X_2)}{dt}$$

in Figure 2C. Please check. If the term $k_6 X_2 X_3$ is present then it cannot be eliminated with the concatenating monomial O_1 . On the other hand, if this reaction is removed

(i.e. this term is absent) then the overall network simply becomes a trivial RPA network where the output of one RPA network (i.e. the network comprising $\mathbf{X}_3 - \mathbf{O}_1$) is passed as an input to another RPA network (i.e. the antithetic network with $\mathbf{X}_1 - \mathbf{X}_2$). It is straightforward that connecting RPA networks in series (with catalytic reactions) would still result in a RPA network. Such examples are not appropriate for demonstrating the novel results in this paper.

The second example in Figure 3 seems to be taken straight from [4] (see Fig. 2). This should be clearly stated when the example is being introduced in the main text and also in the caption of Figure 3. Also mention that the linear invariants shown in Fig. 3 can be deduced from the approach in [4] (this is stated in passing in the conclusion but it should be stated more prominently when the example is being discussed.)

6. **Many claims without proofs in the Supplement:** The Supplement has been considerably revised, but still many arguments are unclear because proper proofs have not been provided or referenced:

- Why should eq. (8) hold when eq. (9) holds? Can the networks always be partitioned this way? Please explain.

- On page 21 in the Supplement it states that:

Since a perturbation to the CRN that alters the steady state of x_j will also alter the steady-states of other non-RPA capable variables (eg. x_m), $\mathbb{R}[\bar{x}]$ will contain polynomials in x_j and x_m , and that are not contained in $I_f \cap I_p$.

Why should such a perturbation always exist?

- Also on page 21 in the Supplement it says that

The set \bar{x} now contains two independent (uncoupled) variables in the sense that a perturbation to the CRN that alters the steady-state of one of the variables does not affect the steady-state of the other.

Why does this hold? Please elaborate.

- In general, in the proof of the Kinetic Pairing Theorem the authors work over the ring of polynomials over species-variables x_1, \dots, x_n . Shouldn't the system parameters (i.e. rate constants) be included in this ring, as they would appear in the RPA polynomial? This inclusion of parameters is there in the Singular code but not in the proof. However simply adding the parameters in the ring is not sufficient as the n -th roots of the parameter would be added, as the examples mentioned in point 3 show.

7. **Other minor issues:**

- The definition of RPA must be shifted to the main text due to its centrality in understanding the message of the paper.
- Why is the variable x_i missing in $g(x_j)$ in figure 4 (main text) and figure S4 (supplement)?

- Replace “consistutes” with “constitutes” on line 72 in the main text. Please run a spell check.
- The paper says that if a network is RPA the integrator is guaranteed to exist. For example the following on page 22 in the main text

In principle, there should always exist some single nonlinear coordinate change to extract a single output-driven internal model (Fig. 9a) from systems rate equations, corresponding to a single integral of the systems tracking error (Fig. 9b)

or the following on page 16 in the Supplement

All classes of RPA, including ACR, thus require some form of integral control.

Please explain which version of IMP can be used to verify this existence. See [1] for a recent review on IMP.

- On page 20 the authors state that “It is striking to note that the original form of the CRN (Fig. 8a) eludes the Shinar-Feinberg theorem, even though the CRN exhibits ACR and has a deficiency of one.” However does it fit the results in [4]? Please comment on this. Also explain why the authors found the method for finding linear invariants “ad hoc” (lines 568-569 on page 25).
- On the Supplement page 15 it is stated that

Mass-conservative CRNs therefore have no external stimuli or inputs, and can only be perturbed by altering the total abundances (or concentrations) of the constituent molecules - i.e. by altering the initial conditions.

Why cannot the perturbation come in the form of parameter variation, e.g. of a conversion reaction (that conserves mass).

References

- [1] M. Bin, J. Huang, A. Isidori, L. Marconi, M. Mischiati, and E. Sontag. Internal models in control, bioengineering, and neuroscience. *Annual Review of Control, Robotics, and Autonomous Systems*, 5:55–79, 2022.
- [2] A. Gupta and M. Khammash. Universal structural requirements for maximal robust perfect adaptation in biomolecular networks. *Proceedings of the National Academy of Sciences*, 119(43):e2207802119, 2022.
- [3] N. Meshkat, A. Shiu, and A. Torres. Absolute concentration robustness in networks with low-dimensional stoichiometric subspace. *Vietnam Journal of Mathematics*, 50(3):623–651, 2022.
- [4] M. Pérez Millán, A. Dickenstein, A. Shiu, and C. Conradi. Chemical reaction systems with toric steady states. *Bulletin of mathematical biology*, 74(5):1027–1065, 2012.

- [5] M. S. Pérez Millán. *Métodos algebraicos para el estudio de redes bioquímicas*. PhD thesis, Universidad de Buenos Aires. Facultad de Ciencias Exactas y Naturales, 2011.

Reviewers' Comments:

Reviewer #1:

Remarks to the Author:

See attached.

Reviewer #2:

Remarks to the Author:

The reviewers have comprehensively addressed my questions and points. This paper addresses a fundamentally important question in biology and it does so using a powerful mathematical framework that the authors have developed and fine-tuned.

The particular appeal of this approach is that it sets a template (and I imagine that in the future we will see many more examples of this type) is that the mathematical analysis has a level of generality that means that the authors results are robust and reliable. Having solid mathematical foundations for a complex biological phenotype is still rare, but the current paper shows that the search for such theoretical underpinnings can be fruitful.

Overview: The aim of this paper is to provide an algebraic characterisation of the hidden integral controller and construct an algebraic procedure to identify it in adaptation-capable networks. It is shown that this procedure is intimately connected to the structure of the networks via the well-known deficiency theory for chemical reaction networks.

Recommendation: I am grateful to the authors for revising the manuscript to address many of the points that I raised in the first review. Even though the paper has significantly improved, many of the more serious concerns remain. These concerns, which are outlined below, should be fully addressed before the paper can be reconsidered again for *Nature Communications*.

Our response:

Once again, we are truly grateful for the huge investment of time the reviewer has clearly made in considering our work in such careful detail. We have reflected thoughtfully on all the reviewer's points and respond to each in turn below.

All changes made to our paper and Supplementary Information (SI) are noted in **red text**.

1. **Connection to integral control:** The paper claims to uncover universal structures for embedded "integral control". However in order to successfully demonstrate this, a coordinate transformation (possibly nonlinear) needs to be constructed that gives rise to the "integrator" variable. It does not matter if this coordinate transformation is done in a single step or in multiple steps (as proposed by this paper), but its existence needs to be shown in order to identify the integral mechanism. In the Supplement the authors write

We note that, although there should always exist some single nonlinear coordinate change that yields the RPA polynomial in principle, for any mass action system, identifying such a single coordinate change in practice may be extremely difficult in all but the very simplest RPA-capable CRNs.

I agree that proving the existence of an integrator may be difficult, but unfortunately it is unavoidable for showing integral control.

Our response:

What we show in this paper is that a particular type of nonlinear transformation is always able to decompose an RPA-capable set of CRN reactions into an organized collection (a 'topological hierarchy') of *linear coordinate transformations*. Each of these linear coordinate transformations corresponds to a subsidiary integral controller within the CRN. This is a fundamentally different type of integral control implementation from the 'conventional' interpretation of integral control in engineering control theory. Indeed, the nonlinear transformation we propose is **not** a **coordinate** transformation. Rather, it is a map from the system reaction rates to a function space with the same algebraic structure as the CRN reactions (i.e. that of a *commutative ring*), which offers a **universal description** of the coordinated interactions of (integral computing) internal models – obtained via linear coordinate changes - that holds for **all possible** RPA-capable CRNs.

We do acknowledge that in our previous rebuttal letter (as opposed to our paper), we had referred to the mathematical transformation in several places as a coordinate change, which was not correct. We apologise for unwittingly side-tracking the discussion through such sloppy (and incorrect) wording in our previous response. We have now made absolutely certain that this incorrect terminology does not appear anywhere in our paper or in our accompanying SI. The reviewer is entirely correct to point out that the nonlinear transformation we present in this paper is **not** a nonlinear **coordinate** change.

The **conventional** view of the ‘internal model principle’ (IMP) in engineering control theory (particularly for application to problems in systems biology and bioengineering [1,8,9]) suggests that, for robust rejection of constant disturbances, there must exist *some* decomposition of the system (obtained via a coordinate transformation, if needed) into two distinct subsystems: (i) an ‘*output-driven internal model*’ (of the constant disturbance), and (ii) the remainder of the system ([8,9]). This decomposition will generally require a nonlinear coordinate transformation, although many simple CRNs have been identified (as we point out in our paper and accompanying Supplementary Information) for which a linear coordinate transformation is sufficient. In particular, if the system at hand is not already in *feedback* form, then the internal model-identifying coordinate transformation must be able to recast the system into an **integral feedback** form (see references [1] and [9] below, as well as Figure 1b in our paper, for some simple examples of this requirement). If a network can indeed exhibit RPA, then according to this version of the IMP (see [8] and [9], for example, as well as Section 3 in [1]), there should always exist **some (single) coordinate transformation** that recasts the system into such a *feedback* system, with an embedded *output-driven internal model* – even if the system is topologically feedforward structured. This is the concept we are referring to in the excerpt from our Supplementary Information quoted by the reviewer above (‘... *there should always exist some single nonlinear coordinate change that yields the RPA polynomial ... for any mass action system*’).

Note, incidentally, that (in contrast to the formulation we present in our paper) this conventional interpretation of the IMP ([8,9]) requires that certain properties of the disturbances be known *a priori*, including which model variables can be directly regulated by the disturbance, and also requires the prior identification of an ‘output’ for the model (which must be able to ‘adapt’ to, or *reject*, those disturbances). For an RPA-capable system, being able to identify such a nonlinear coordinate transformation, if it exists, simply confirms to us that the system can indeed exhibit robust asymptotic tracking of the output variable’s ‘setpoint’ when subjected to the specified (constant in time) disturbances. It also identifies the specific transformed variable that actually computes the integral of the ‘error’ (where the error in question is the difference between the instantaneous value of the output variable and its setpoint). *But if we already know* (using some *other* analytical method, say) that RPA obtains for certain variable(s) **in a specific CRN**, then explicitly identifying the nonlinear coordinate change that transforms that particular system into integral feedback form does not actually provide any further useful information. The fundamental point we make in our Discussion is that there **is no general way** to

propose a *single nonlinear coordinate change* that can reveal the fundamental mechanisms by which *all* RPA-capable CRNs actually implement RPA, and thereby provide a concrete characterization of the entire solution space to the RPA problem for CRNs.

Consider, for example, Shinar and Feinberg's deficiency-two model of the EnvZ-OmpR osmoregulation network (first presented in the Supplementary Materials of [10], and considered in Figure 3 of our main paper). For this *specific network*, the existence of RPA (and ACR more specifically) in the molecule phospho-OmpR can be shown using a variety of methods. Shinar and Feinberg, in their Supplementary Materials [10], show through a sequence of manual algebraic substitutions that phospho-OmpR has a steady-state value that is a rational function of the CRN parameters, independent of the total abundances of the various interacting molecules. Karp et al. [6], on the other hand, demonstrate how 'complex-linear' invariants may be computed, and show that for this particular CRN, two linearly-independent such invariants may be combined to demonstrate RPA in phospho-OmpR, with the same setpoint as previously calculated by Shinar and Feinberg [10]. Now, since this CRN exhibits RPA (at phospho-OmpR), undoubtedly, there must undoubtedly exist some nonlinear coordinate transformation that could recast this system (a 'balancer' module, as we show, with a feedforward structure) into an integral *feedback* system with a single output-driven (i.e. phospho-OmpR-driven) internal model, where a single (transformed) variable computes the error in phospho-OmpR in comparison with its (known) setpoint. But even if we were to identify the requisite coordinate transformation, explicitly and analytically, for this particular CRN, what would this coordinate map actually tell us (vis-à-vis the RPA-capacity of the CRN) that we didn't already know? More importantly, what could this specific coordinate map possibly tell us about the space of *all possible RPA-capable CRNs*, or the *general (universal)* properties of collections of chemical reactions that can confer RPA on a subset of the interacting molecules? Absolutely nothing! The integral-feedback-recasting *nonlinear coordinate transformation* for an RPA-capable CRN *is unique to that particular CRN*. In other words, the single nonlinear coordinate transformation required to identify an internal model in the deficiency-two Shinar-Feinberg EnvZ-OmpR model discussed above, if such can be explicitly identified, will necessarily be a *different* nonlinear map from the one required for the Cappelletti et al. [11] toy model (Example 1 in Section S3.1 of our SI) – *despite the fact that the two models are fundamentally alike*: both are Balancer modules (topologically speaking) for which a single balancer invariant and a single connector invariant are obtained via *linear coordinate transformations*, and require a *single concatenating monomial* to obtain an RPA polynomial.

By contrast, our nonlinear transformation (Eq. 6) is of a fundamentally different character. Nowhere in our paper do we claim that this nonlinear transformation is a *coordinate transformation*. Importantly, this nonlinear map relaxes the *feedback* requirement of the IMP in its conventional interpretation (as discussed above, see also Section 6.2 in [9]), and instead preserves the underlying topological structure of the network. Thus, RPA-capable CRNs of Balancer type retain their feedforward characteristics under this transformation. **We maintain that referring to an**

embedded integral control is entirely justified since the nonlinear map in question is always able identify, for any RPA-capable CRN, a topologically organized collection of *linear coordinate changes*, each one of which identifies a subsidiary internal model which does recapitulate the dynamical structure of the disturbance (i.e. constant-in-time), and which thereby imposes RPA on some characteristic of the network, with its own setpoint. Each such linear coordinate change thereby corresponds to subsidiary control problem with its own linear integral variable.

For the Cappelletti et al. [11] toy model, for instance, the nonlinear transformation which projects the system onto two variables (A and B – see Singular Code in S5.3 of our Supplementary Information) can be decomposed into two linear coordinate changes, corresponding to two key invariants. One of these is a ‘balancer’ invariant, $k_2BC - k_3B^2$, which represents the ‘error’ in the **ratio of the two ‘proportioner’ molecules**, C and B, in comparison with their ‘setpoint’: i.e. $k_2B^2 \left(\frac{C}{B} - \frac{k_3}{k_2} \right)$. The other is a ‘connector’ invariant, $k_1AB - k_7C$, which represents the error in the **ratio of the ‘upregulating’ contributions** (in this case, AB) **to the ‘downregulating’ contributions** (in this case, C), for the ‘connector’ calculation: i.e. $k_1C \left(\frac{AB}{C} - \frac{k_7}{k_1} \right)$. Together, these two independent computations, each with a linear integral variable, and each conferring RPA on *some* (topologically important) feature of the CRN, confer RPA on the molecule A. The setpoint for A is a combination of the setpoints from the two contributing subsidiary (linear) problems. Of course, many CRNs will have much more complicated invariants than these, involving more than two terms per invariant, but the same fundamental principles hold. (This is the *universality* of the framework we present).

The feedforward model presented by Bin et al. [1] provides a nice illustration of the distinction between the conventional interpretation of the IMP and the one we develop here, as applicable to RPA in CRNs. In Section 3 of [1] the authors review a very simple two-variable incoherent feedforward loop (IFFL) model (Eq. 25) that has been widely studied in the systems biology literature (including in [9]). The two model equations are

$$\frac{dx}{dt} = \alpha u - \delta x,$$

and

$$\frac{dy}{dt} = \beta \frac{u}{x} - \gamma y.$$

The authors show that a nonlinear *coordinate* map that transforms these two equations into a partitioned form, corresponding to an integral feedback structure, is

$$(x, y) \mapsto (z_1, z_2) = (y, \varphi(x, y)) = (y, \alpha y - \beta \log x),$$

which produces, in the new coordinates (z_1, z_2) :

$$\frac{dz_1}{dt} = \beta u e^{(z_2 - \alpha z_1)/\beta} - \gamma z_1,$$

$$\frac{dz_2}{dt} = \beta\delta - \alpha\gamma z_1.$$

This transformed system has an output $y = z_1$, and has the desired internal model form, where z_2 is the transformed variable that integrates the error (in $z_1 = y$). This nonlinear coordinate change nicely confirms that RPA is possible at y , and that its setpoint is $\beta\delta/\alpha\gamma$. But a **different** nonlinear transformation of the system (*not* a *coordinate* transformation) is also possible:

$$\beta \frac{dx}{dt} - \alpha x \frac{dy}{dt} = \alpha\gamma x \left(y - \frac{\delta\beta}{\alpha\gamma} \right)$$

Unlike the coordinate transformation considered previously, this particular transformation underscores the fact that this simple two-variable reaction system is *fundamentally of the same type* as both the Cappelletti et al. [11] toy model (Example 1 in our Supplementary Information S3.1) and the deficiency-two Shinar-Feinberg EnvZ-OmpR model we discussed above: a Balancer module, with two independent (linear) invariants – a balancer invariant, and a connector invariant – combined via a single concatenating monomial (in this case, x). This transformation also reveals the setpoint, $y = \beta\delta/\alpha\gamma$. Of course, we do acknowledge that the rate equations above are not polynomials, the only consequence of which, in the context of our paper, is that we cannot use the powerful automated algorithms of algebraic geometry that compute Gröbner bases (which pertain to polynomials); nevertheless, rational functions are C^1 -smooth functions on $(0, \infty)$, and thus possess the algebraic properties of a ring as required for the application of our Theorem 1. The RPA polynomial so obtained involves two variables (one RPA-capable (i.e. y) and one non-RPA-capable (i.e. x)) and is thus the single generator of the relevant ideal. By contrast, the nonlinear coordinate transformation discussed in [1], as noted above, is unique to this particular set of rate equations, and provides no insight into the **general properties** of all RPA-capable CRNs (or even its fundamental connection to the RPA-conferring properties of the Cappelletti et al. [11] or Shinar-Feinberg [10] models discussed above). Shoval et al. [9] acknowledge that the coordinate transformations that recast systems into integral feedback form ‘may well be merely a mathematical construct with no biological meaning’ (see Section 6.3 in [9]).

Again, we maintain emphatically that this is a fundamentally different way of looking at integral control from the IMP as it is usually understood in engineering control theory. This approach identifies where, within the collection of reactions themselves, the computations relevant to RPA actually occur at the level of the interacting molecules of the CRN. Identifying a (single) nonlinear **coordinate** change that transforms the system into an integral feedback system, with a **single** embedded internal model, can only confirm that RPA will obtain; it does not explain *how* it is implemented by the *original variables* of the system (ie. the molecular concentrations).

Being able to characterize the entire space of all possible RPA-capable CRNs in such a general way has enormous practical implications because it provides a completely comprehensive and universal understanding of how to either destroy or maintain the presence of RPA – through evolution, or via experimental or clinical interventions, where new mutational events or exogenous enzyme inhibitors *exert their effects at the molecular level*. In particular, we can destroy the RPA property in a CRN by disrupting *any one* linear integral controller. By contrast, retaining the RPA property requires that *all* linear integral controllers already present be preserved. Cappelletti et al. [11], for instance, consider how a linear integral controller can be preserved when adding in a new (exogenous) stimulus to the network (see Theorem 5.1 in [11]); but there are many other types of perturbations that could occur in CRNs in either an evolutionary setting, or in experimental/clinical settings. Given that many of the most complex signalling networks that arise in nature, such as signal transduction networks in mammalian cancer cells, will likely never be delineated in complete intricate detail in terms of elementary chemical reactions, the robustness-conferring mechanisms that self-assemble in biology require a completely new approach, an entirely new language, for understanding the fundamental structures of life’s networks. In this paper we wish to offer just such an approach.

Let us look into this issue in more details.

Suppose the dynamics is governed by a system of ODEs:

$$\frac{dx}{dt} = f(x),$$

where x is the n -dimensional state of the system and $f(x) = (f_1(x), \dots, f_n(x))$ are the rates of evolution of all the species. In order to exhibit an integrator, we need to find a real-valued function $H(x)$ such the dynamics of $z(t) = H(x(t))$ is given by the ODE:

$$\frac{dz}{dt} = \nabla H(x) \cdot f(x) = p(x)(x_i - c) := \rho(x), \quad (0.1)$$

where ∇ is the gradient operator, \cdot is the usual dot product and $\rho(x) = p(x)(x_i - c)$ is what the authors call the “RPA polynomial” for the RPA variable x_i . Suppose one is able to find nonlinear polynomials h_1, \dots, h_n (as in Theorem 1 of the paper) such that

$$h_1(x)f_1(x) + \dots + h_n(x)f_n(x) = \rho(x).$$

However this does not necessarily mean that a function $H(x)$ can be constructed so that (0.1) holds. Observe that such a function $H(x)$ would need to have the following partial derivatives

$$\frac{\partial}{\partial x_j} H(x) = h_j(x) \quad \text{for } j = 1, \dots, n.$$

Unless each $h_j(x)$ is a constant (in which case $H(x)$ is linear), it is unclear why such a function $H(x)$ would exist. Since the approach in the paper uses nonlinear elimination steps

(via concatenating monomials), the existence of $H(x)$ cannot be ascertained. In fact if for a pair j, k we have

$$\frac{\partial}{\partial x_j} h_k(x) \neq \frac{\partial}{\partial x_k} h_j(x)$$

then certainly $H(x)$ cannot be constructed with partial derivatives h_1, \dots, h_n . This discussion shows that while existence of a RPA polynomial may be necessary for integral control, it is not sufficient and hence this property does not characterise integral controllers (a main claim of the paper). This gap between necessity and sufficiency can be bridged if the authors can show that for RPA networks one always has

$$\frac{\partial}{\partial x_j} h_k(x) = \frac{\partial}{\partial x_k} h_j(x) \quad \forall j, k.$$

Our response: Yes, the reviewer is entirely correct to point out that the nonlinear map we propose in this paper (Eq. 6) is not a *coordinate* transformation. But again, we don't actually seek a diffeomorphism $H(x)$ of the type the reviewer describes, and we don't claim that our collection of elimination polynomials gives rise to a *coordinate* transformation (other than in the special case of constant elimination polynomials). We do apologise again for incorrectly referring to the nonlinear map as a nonlinear coordinate change in our previous rebuttal letter. This was sloppy (and incorrect) wording, and we have made absolutely certain that no such inaccuracies appear anywhere in our manuscript or in our SI.

It is true that the existence of an RPA polynomial, ρ , contained in the ideal $I_f = \{h_1 f_i + \dots + h_n f_n : h_i \in \mathbb{R}[x_1, \dots, x_n]\}$ is a necessary but insufficient condition for RPA. As our Theorem 1 makes clear, $\rho = g(x_i, x_j)(x_i - c)$ must **generate** the **principal** ideal in two variables, $I_f \cap \mathbb{R}[x_i, x_j]$, where x_j is (any) non-RPA-capable variable. This condition is both necessary and sufficient for RPA at x_i , under the assumption of stability. But nowhere do we claim that a CRN for which $\rho(x_i, x_j)$ generates $I_f \cap \mathbb{R}[x_i, x_j]$ contains "an" integrator. This can only be true if the elimination polynomials are constants, as the reviewer points out. If non-constant elimination polynomials are required, our framework requires there to be *multiple* independent (linear) integrators, connected via concatenating monomials; the nonlinearity is thereby relegated entirely to the concatenating process, not to the identification of internal models. In general, the output for one internal model will be an input for another. This collection of linear coordinate maps is always obtainable from a decomposition of the special nonlinear transformation we identify. That is the viewpoint we develop here.

Again, we are re-interpreting the IMP from a completely different standpoint from the one considered in prior work, in which the existence of a *nonlinear coordinate transformation*, and an associated **output-driven internal model**, is normally at issue. Note that the RPA polynomial we reference in Eq. 6 (and Theorem 1 – the Two-Variable Kinetic Pairing Theorem) contains strictly two variables in contrast to $\rho(x)$ in

(0.1) above, which involves an unspecified number of variables. We simply do *not* claim that the transformation that identifies the (two-variable) RPA polynomial is associated with a single integrator. Rather, we claim that this transformation is always able to obtain ρ via a connected collection of *linear integrators*, each requiring a *linear coordinate change*, and each imposing RPA on some topologically important feature of the CRN.

Bin et al. in [1] offer a particularly insightful comment. They state in Section 3: ‘The question that the IMP asks is, If a system ... is seen experimentally to regulate against all inputs (in some class of functions, e.g. constants), then what can be said about its internal structure? Answers to this question may help guide experimentalists and modelers by ruling out putative mechanisms and suggesting a search for components responsible for adaptation’. The version of the IMP those authors go on to discuss in that paper (involving a coordinate transformation that identifies an output-driven internal model, within a feedback structure) provides one answer as to what can be said about the internal structure of RPA-capable networks. Our paper provides a different answer – one that reveals the *universal properties*, at the *molecular level*, of all RPA-capable networks.

2. Questions about the Kinetic Pairing result: In the previous review round, I had proposed a possible RPA network and asked the authors how it satisfies the Kinetic Pairing result (Theorem 1). The network has two reactions given by

and there may be several other reactions involving species X_2, \dots, X_5 that can be catalysed by X_1 but they do not change X_1 . In the rebuttal letter, the authors consider certain instances of such a network, and claim that in these networks all possible steady states lie on the boundary on the positive orthant (e.g. x_2 or x_3 is zero) and hence the networks are not RPA. In particular, the authors state in their rebuttal letter that:

The two reactions proposed by the reviewer cannot impose RPA on X_1 no matter which reactions are added to the original two. This is because the two given reactions are linearly independent, and the CRN formed from these reactions has deficiency zero. Moreover, the two complexes that contribute to the apparent RPA-polynomial in the reaction rate f_1 are not members of the same terminal SCC. It is clear from our discussion of the decomposition of CRNs into algebraically independent subnetworks a decomposition which is closely tied to the partition of deficiency among independent subsets of reactions (see SI Section S1.3) - as well as our extensive discussions of rowspan polynomials (complex linear invariants) throughout the remainder of our supplement, that no possible addition of extra reactions can turn the original two into an RPA-conferring set. We briefly note in Section S4.4.1 (see also footnote in that section) the special case where a deficiency zero collection of reactions can engender RPA namely when the two relevant complexes reside in the same terminal SSC, and correspond topologically to a trivial (isolated) connector node. The pair of reactions suggested by the reviewer are clearly not of this type.

To test the authors' claim above, I tried to come up with instances of such networks (i.e. (0.2) + some reactions) which are RPA and yet do not have this issue of steadystate not being in the positive orthant. Please explain how the Kinetic Pairing result fits these networks.

Our response: Yes, in the previous round of review the reviewer provided us with the two reactions given in (0.2) above and, pointing out that such a reaction pair would produce a reaction rate $\frac{dx_1}{dt} = c_1x_2x_3 - c_2x_1x_2x_3$, asked us how this could be reconciled to the claims of our **Two-Variable Kinetic Pairing Theorem** since this equation has the apparent form of an RPA polynomial, yet appears to have three rather than two variables. We pointed out that it is possible to have an RPA polynomial involving three (or more) species, but only two variables, in CRNs containing a 'boundary variable'. But more importantly, we went on to show that, for the particular reaction pair proposed by the reviewer equipped with the needed 'closure' reactions (for which we provided three different examples), the projection of the ideal generated by the collection of CRN rate equations (I_f) onto **two variables** (one RPA-capable, and one non-RPA-capable) produced the elimination ideal $\langle x_2x_3 \rangle$. Hence, none of the three CRN examples could exhibit RPA.

Our Theorem 1 (the 'two-variable kinetic pairing theorem') states, simply, that a CRN is RPA-capable in some variable x_i exactly when the projection of the steady-state ideal, I_f , onto x_i and x_j (for *any* non-RPA-capable x_j) is **generated by a single polynomial** of the form $\rho = g(x_i, x_j)(x_i - c)$. We call such polynomials 'RPA polynomials'. As we pointed out in our earlier rebuttal, it is not enough for a set of CRN equations to contain a reaction rate that has the form of an RPA polynomial (as is the case for all the reviewer's examples, both here and in the previous round of review, which all have $\frac{dx_1}{dt} = c_1x_2x_3 - c_2x_1x_2x_3$). Rather, the elimination ideal $I_f \cap \mathbb{R}[x_1, x_j]$ must be **generated** by an RPA polynomial.

In the reviewer's new report, he/she provides three alternative collections of reactions, all of which contain the reaction $\frac{dx_1}{dt} = c_1x_2x_3 - c_2x_1x_2x_3$, and all of which do, in fact, exhibit RPA. As the reviewer rightly points out, none of these examples contain any boundary variables. According to our Theorem 1, all three of these new examples will indeed exhibit RPA since the elimination ideal $I_f \cap \mathbb{R}[x_1, x_j]$ in each case (with $x_j \in \{x_2, x_3\}$ for the first two CRN examples, and $x_j \in \{x_2, x_3, x_4, x_5\}$ for the third example) is generated by a **two-variable RPA polynomial**. In particular, $\rho = x_1 - c_1/c_2$ in the first and third CRN examples (where the pairing function is zero-order in both its arguments), and $\rho = x_1^2(x_1 - c_1/c_2)$ in the second CRN example (where the pairing function is zero-order in the non-RPA variable). Thus, all relevant **two-variable** projections for these CRNs produce an ideal generated by a single polynomial of the requisite form, hence the CRN exhibits RPA. Notice that the three-variable reaction rate for x_i is not the **generator** of $I_f \cap \mathbb{R}[x_1, x_j]$ in any of these examples (although it is obviously *contained* in I_f). As we explained previously, the fact that a CRN contains a reaction rate of the form $\frac{dx_1}{dt} = c_1x_2x_3 - c_2x_1x_2x_3$ is, in and of itself, *irrelevant* vis-à-vis the RPA capacity of the CRN (and hence, the claims of our Theorem 1). What matters is

what actually generates the ideal $I_f \cap \mathbb{R}[x_i, x_j]$ – i.e. the projection of I_f onto the two variables x_i and x_j (where one of these is RPA-capable, and one is not). The three CRNs discussed in the previous round of review, as well as the three new CRNs offered by the reviewer, *all* contain the reaction rate $\frac{dx_1}{dt} = c_1x_2x_3 - c_2x_1x_2x_3$. But for the previous set of three CRNs (considered in the last review round), the relevant elimination ideal is $\langle x_2x_3 \rangle$, and as a consequence, those CRNs cannot exhibit RPA in x_1 (or anything else). For the new set of CRNs, the relevant elimination ideal is $\langle x_1 - c_1/c_2 \rangle$ for the first and third examples, and $\langle x_1^2(x_1 - c_1/c_2) \rangle$ for the second example, and as a consequence, these CRNs *do* exhibit RPA at x_1 , with setpoint c_1/c_2 .

We make some additional observations for each of the three new CRNs in turn below. As the reviewer notes, we had observed in our previous rebuttal that the two specific reactions originally proposed by the reviewer together have deficiency zero, and that the addition of the necessary closure reactions will not be able to increase this deficiency. As a consequence, that CRN could not exhibit RPA. In other words, the fact that those particular CRNs were not RPA capable was not tied to the specific choices of closure reactions that we provided as illustrative examples. Of course, if one begins with a *different* pair of reactions (i.e. not involving X_4 and X_5), as is the case for the first two new examples the reviewer provides, or adds in *more than* the needed closure reactions, as is the case for the third new example, then one can certainly increase deficiency, and thereby arrive at an RPA-capable set. But in this case, the RPA-capacity is tied inextricably to the altered structure arising from the different starting pair and/or the redundant closure reactions, with the attendant increase in deficiency associated with that altered structure, and is not due solely to the mere presence of the original pair (which appear to produce an apparent ‘RPA polynomial’ for $\frac{dx_1}{dt}$ in all cases). We explain this point in more detail in our ensuing analysis. We apologise if we unwittingly side-tracked the discussion by raising the issue of the original pair being characterized by a deficiency of zero, and that adding in the necessary ‘closure’ reactions to allow us to work with that original pair cannot increase deficiency. The point we were really trying to emphasize was that the mere presence of $\frac{dx_1}{dt} = c_1x_2x_3 - c_2x_1x_2x_3$ in the rowspan of the system, even for cases where x_2x_3 is a boundary variable, does not, in and of itself, imply RPA. Our Theorem 1 explains why this is the case.

Again, the fundamental issue in all six CRN examples (the three here, as well as the three from the previous round of review) is that the reaction rate for x_1 does not actually *generate* the two-variable elimination ideal for these CRNs. In general, CRNs for which one of the reaction equations is an ‘apparent’ RPA polynomial may or may not exhibit RPA.

I start with a simpler version of this network with only three species (i.e. species X_4 and X_5 are absent) where I add inflow and outflow for species X_2 and X_3 . So the overall network becomes

This network has no boundary variables and so the variables are the same as species. It seems that this network will indeed show RPA for x_1 with set-point c_1/c_2 . In Figure 1 I plot the simulated dynamics (rescaled to have the set-point of x_1 as 1) for three randomly chosen values of initial states and rate constants, and one can see that RPA holds.

Our response: Here, the reviewer starts with a *different* version of the initial two reactions (rather than the two given in (0.2)), so this CRN can have a deficiency of one, even with only the minimum needed closure reactions. In any case, this CRN does indeed exhibit RPA because $\langle \frac{dx_1}{dt}, \frac{dx_2}{dt}, \frac{dx_3}{dt} \rangle \cap \mathbb{R}[x_1, x_2] = \langle x_1 - \frac{c_1}{c_2} \rangle$. We provide our Singular code and output below so that the reviewer can verify (see GI[1]):

```

Last login: Wed Dec 14 11:35:16 on ttys000
/Applications/Singular.app/Contents/MacOS/./bin/SINGULAR.sh ; exit
(base) araujo@SEF-PA00144783 ~ % /Applications/Singular.app/Contents/MacOS/./bin/SINGULAR.sh ; exit
SINGULAR / Development
A Computer Algebra System for Polynomial Computations / version 4.1.2
by: W. Decker, G.-M. Greuel, G. Pfister, H. Schoenemann 0< Feb 2019
FB Mathematik der Universitaet, D-67653 Kaiserslautern \
> ring R = (0, c1, c2, c3, c4, c5, c6), (x3, x2, x1), lp;
> poly f1 = c1*x2*x3 - c2*x1*x2*x3;
> poly f2 = c3 - c5*x2 - c1*x2*x3 - c2*x1*x2*x3;
> poly f3 = c4 - c6*x3 - c1*x2*x3 - c2*x1*x2*x3;
> ideal I = f1, f2, f3;
> ideal GI = groebner(I);
> GI;
GI[1]=(c2*c3*c4)*x1+(-c1*c3*c4)
GI[2]=(2*c1*c5)*x2^2+(-2*c1*c3+2*c1*c4+c5*c6)*x2+(-c3*c6)
GI[3]=(c6)*x3+(-c5)*x2+(c3-c4)
> █

```

Whereas (0.2) includes two additional species, X_4 and X_5 , each assigned to a different reaction in the pair, thereby guaranteeing the linear independence of the reactions once the requisite closure reactions are added, this property is no longer present in the new reaction set. When organized correctly into its linkage classes, this CRN has six complexes, two linkage classes and a rank of three, resulting in a deficiency of one.

The CRN has just two non-terminal complexes, from which it follows that there is a polynomial of the form $a_1x_2x_3 - a_2x_1x_2x_3$ in the rowspan of the system – a feature that was already obvious from the form of $\frac{dx_1}{dt}$. But note that this fact, in and of itself, does not guarantee that RPA will obtain. The three examples considered in the previous round of review also contained $\frac{dx_1}{dt} = c_1x_2x_3 - c_2x_1x_2x_3$ but did not exhibit RPA. Note that the Shinar-Feinberg theorem presupposes that the system admits a positive steady-state; it does not, in and of itself, provide any method for checking that this will indeed be the case. Note also that for the cases previously considered in which x_2x_3 was a boundary variable, it was *possible* (in principle) for the generating RPA polynomial (referenced in our Theorem 1) to take the form $c_1x_2x_3 - c_2x_1x_2x_3$, since this is a *two-variable* polynomial in that case. (But again, since $I_f \cap \mathbb{R}[x_1, x_2x_3] = \langle x_2x_3 \rangle$ for those examples, the CRNs could not actually exhibit RPA.) Here, where there are no boundary variables, it is not possible for the (generating) RPA polynomial to take the form $\frac{dx_1}{dt} = c_1x_2x_3 - c_2x_1x_2x_3$ since this contains three variables. An RPA polynomial contains (at most) **two** variables.

The reason this particular CRN has a deficiency of one is that a linear combination of the reactions involving c_2, c_3, c_4 (in the second linkage class only) produces $X_1 \rightarrow \emptyset$, whereas a linear combination of the reactions involving c_1, c_3, c_4 (involving the first linkage class, and thus involving the new pair of complexes $X_2 + X_3$ and X_1) produces $\emptyset \rightarrow X_1$. Thus, X_2 and X_3 regulate both the production and the degradation of X_1 . This makes the CRN a balancer module, where X_2 and X_3 play the role of diverter species. Note that although the reaction involving c_5 could be used in place of the reaction involving c_3 in the argument above, these two reactions involve the same complexes (\emptyset and X_1), so the ‘redundancy’ in the reactions does not result in an increase in deficiency (nor any distinct new cycles or feedforward actions). The same holds for using the reaction involving c_6 in place of the reaction involving c_4 .

Next I modify this network to have the production of X_2 and X_3 catalysed by X_1 . So the new network becomes

This network also appears to be RPA as shown by the simulations in Figure 2.

Our response: Yes, this CRN does indeed exhibit RPA because $\langle \frac{dx_1}{dt}, \frac{dx_2}{dt}, \frac{dx_3}{dt} \rangle \cap \mathbb{R}[x_1, x_2] = \langle x_1^2 \left(x_1 - \frac{c_1}{c_2} \right) \rangle$. Our Singular code and output are provided below for easy verification:

```

Last login: Thu Dec 15 08:32:11 on console
/Applications/Singular.app/Contents/MacOS/./bin/SINGULAR.sh ; exit
(base) araujo@SEF-PA00144783 ~ % /Applications/Singular.app/Contents/MacOS/./bin/SINGULAR.sh ; exit
SINGULAR / Development
A Computer Algebra System for Polynomial Computations / version 4.1.2
by: W. Decker, G.-M. Greuel, G. Pfister, H. Schoenemann \ Feb 2019
FB Mathematik der Universitaet, D-67653 Kaiserslautern \
> ring R = (0, c1, c2, c3, c4, c5, c6), (x3, x2, x1), lp;
> poly f1 = c1*x2*x3 - c2*x1*x2*x3;
> poly f2 = -c1*x2*x3 - c2*x1*x2*x3 + c3*x1 - c5*x2;
> poly f3 = -c1*x2*x3 - c2*x1*x2*x3 + c4*x1 - c6*x3;
> ideal I = f1, f2, f3;
> ideal GI = groebner(I);
> GI;
GI[1]=(c2*c3*c4)*x1^3+(-c1*c3*c4)*x1^2
GI[2]=(c2*c5)*x2*x1+(-c1*c5)*x2+(-c2*c3)*x1^2+(c1*c3)*x1
GI[3]=(2*c1*c5)*x2^2+(-2*c1*c3+2*c1*c4)*x2*x1+(c5*c6)*x2+(-c3*c6)*x1
GI[4]=(c6)*x3+(-c5)*x2+(c3-c4)*x1
> █

```

Once again, this particular CRN does not actually contain (0.2) but a different pair of starting reactions, so its deficiency readily exceeds zero (unlike the three CRNs we considered in the previous round of review). When correctly organized into linkage classes, this CRN has eight complexes, two linkage classes, and a rank of three, giving a deficiency of three. The reason for the two extra units of deficiency in comparison with the previous case is that the reaction $X_2 \rightarrow \emptyset$ is replicated using a different set of complexes via the reaction $X_1 \rightarrow X_1 + X_2$ (accounting for the first additional unit of deficiency), while the reaction $X_3 \rightarrow \emptyset$ is replicated using a different set of complexes via the reaction $X_1 \rightarrow X_1 + X_3$ (accounting for the second additional unit of deficiency).

Finally, I consider another variant of this network where we have species X_4 and X_5 that reversibly bind to each other to produce an inactive compound, and they catalyse the production of X_2 and X_3 respectively. Therefore the new network becomes

This network also seems to be RPA, as shown in the simulations in Figure 3.

Our response: Yes, for the same reason as the previous two examples, this CRN does indeed exhibit RPA. This particular CRN does actually employ (0.2), and the reviewer has discovered a clever way to separate the necessary closure reactions into three distinct steps, $X_4 \xrightarrow{c_3} X_4 + X_2$, $X_5 \xrightarrow{c_4} X_5 + X_3$ and $X_4 + X_5 \xrightarrow{c_7} \emptyset$, giving a deficiency of one, which creates the needed parallel routes for the production/degradation of X_1 as discussed in the previous two examples. The reactions involving c_5, c_6 replicate the reactions involving c_3, c_4 (respectively) using *different* sets of complexes, thereby increasing the deficiency from one to three. The reaction involving c_8 replicates the reaction involving c_7 using the *same* set of complexes, and therefore does not contribute to any further deficiency increases. Our Singular code and output is provided below:

```
Last login: Thu Dec 15 12:11:13 on ttys000
/Applications/Singular.app/Contents/MacOS/./bin/SINGULAR.sh ; exit
(base) araujo@SEF-PA00144783 ~ % /Applications/Singular.app/Contents/MacOS/./bin/SINGULAR.sh ; exit
SINGULAR / Development
A Computer Algebra System for Polynomial Computations / version 4.1.2
0<
by: W. Decker, G.-M. Greuel, G. Pfister, H. Schoenemann \ Feb 2019
FB Mathematik der Universitaet, D-67653 Kaiserslautern \
> ring R = (0, c1, c2, c3, c4, c5, c6, c7, c8), (x5, x4, x3, x2, x1), lp;
[> poly f1 = c1*x2*x3 - c2*x1*x2*x3;
[> poly f2 = -c1*x2*x3 - c2*x1*x2*x3 + c3*x4 - c5*x2;
[> poly f3 = -c1*x2*x3 - c2*x1*x2*x3 + c4*x5 - c6*x3;
[> poly f4 = c1*x2*x3 - c7*x4*x5 + c8;
[> poly f5 = c2*x1*x2*x3 - c7*x4*x5 + c8;
[> ideal I = f1, f2, f3, f4, f5;
[> ideal GI = groebner(I);
[> GI;
GI[1]=(c2*c3*c4*c8)*x1+(-c1*c3*c4*c8)
GI[2]=(4*c1^2*c7)*x3^2*x2^2+(2*c1*c6*c7)*x3^2*x2+(2*c1*c5*c7)*x3*x2^2+(-c1*c3*c4+c5*c6*c7)*x3*x2+(-c3*c4*c8)
GI[3]=(c3)*x4+(-2*c1)*x3*x2+(-c5)*x2
GI[4]=(c4)*x5+(-c3)*x4+(-c6)*x3+(c5)*x2
> █
```

This CRN has eleven complexes, four linkage classes, a rank of four, and therefore a deficiency of three. Once again, the added deficiency arises from the additional replication of the ‘needed’ closure reactions employing different sets of complexes.

These networks do not appear to satisfy the RPA characterisation given by this paper. Also consider Example 6.5.3 in [5] which does not seem to fit this result either. These counter-examples cast doubt on the correctness of the Kinetic Pairing Theorem on which the entire paper rests.

Our response: None of these networks constitute counter-examples. **All** of these networks satisfy the RPA characterization given by our paper. Our Theorem 1 (the ‘Two-Variable Kinetic Pairing Theorem’) correctly identifies that the three examples suggested here by the reviewer will exhibit RPA at x_1 , since a two-variable RPA polynomial generates the principal elimination ideal $I_f \cap \mathbb{R}[x_1, x_j]$. In addition, our Theorem 1 correctly identifies that the three similar examples considered in the previous round of review *cannot* exhibit RPA since the generator of the relevant elimination ideal is *not* an RPA polynomial (despite the fact that one of the rate equations has the apparent form of an RPA polynomial).

The reviewer mentions Example 6.5.3 in [5], which is the following:

Example 6.5.3. Consider the following polynomials in two variables:

$$\begin{aligned} f_1(\mathbf{x}) &= x_1[(x_1 - 1)^2 + (x_2 - 2)^2] \\ f_2(\mathbf{x}) &= x_2[(x_1 - 1)^2 + (x_2 - 2)^2]. \end{aligned}$$

Notice that these polynomials also have the shape: $f_i = p_i - x_i q_i, i = 1, 2$, where all the coefficients of p_i, q_i are non negative. As we mentioned before, it is possible to find a reaction network modeled with mass-action kinetics, such that the associated system is $dx_1/dt = f_1, dx_2/dt = f_2$.

It is easy to see in this example that the system shows ACR for both variables, since the only positive solution is $x_1 = 1, x_2 = 2$. However, we will ignore this obvious fact and use a procedure inspired by our previous discussion. All the computations here can be checked using any computer algebra system, such as Macaulay2 [61] and Singular [35].

Although this example claims that ‘the system shows ACR for both variables’, this system does not describe an RPA-capable (or ACR-capable) CRN since there are no parameters, and no possible ‘input’ or disturbance, in this model. So there’s really nothing for the network to adapt **to** here. Thus, there is no non-RPA-capable variable; *both* of the variables *appear* to exhibit ‘RPA’, since the system’s steady-state is a single fixed point in \mathbb{R}^2 (and again, that’s not really what RPA is). The special type of RPA known as ACR normally pertains to CRNs where there are mass conservation relationships linking the molecular concentrations, such that the ‘total’ concentration of a particular type/class of molecule can be varied (by altering the initial conditions of the system, for instance). In any case, in order for a CRN to adapt to *any* kind of disturbance, there must always be at least one non-RPA-capable variable in the CRN (otherwise how can there possibly be any kind of ‘internal model’, or any network components that ‘offset’ the disturbance?) In any case, the definition of RPA adopted in our work (as general as this is!) does not accommodate ‘non-adaptive’ cases like this where there’s simply nothing for the network to adapt to. We must respectfully point out that the author of [5] has been a little hasty in

describing this as an ACR-capable network simply because there is a single positive solution $x_1 = 1, x_2 = 2$.

In addition, the author of Example 6.5.3 claims that ‘it is possible to find a reaction network modeled with mass-action kinetics, such that the associated system is $dx_1/dt = f_1, dx_2/dt = f_2'$. The condition being referred to here ($f_i = p_i - x_i q_i$, with all coefficients of p_i, q_i non-negative) is the so-called *Hungarian Lemma*, which states (roughly) that for a polynomial dynamical system to be inducible by a CRN under the mass-action assumption, any term in a rate equation preceded by a negative sign must be divisible by the subject of the rate equation. In other words, for the dx_1/dt equation, each term preceded by a negative sign must include x_1 as a factor. But the Hungarian Lemma, strictly speaking, pertains to CRNs in which each chemical reaction occurs at a rate (normally noted as a kinetic parameter superposed on the associated reaction arrow), that is independent of the corresponding rates of all other reactions. Because Example 6.5.3 is not suitably parametrized in this sense, there is necessarily an ambiguity introduced in terms of which specific reactions might produce the various terms of the equations. In other words, there is not a one-to-one correspondence between reaction rates and putative CRN structure, as predicted by the Hungarian Lemma under the specified conditions. The rate f_1 , for example, contains the term $-2x_1^2$; this could in principle arise from either $2X_1 \xrightarrow{2} X_1$ or from $2X_1 \xrightarrow{1} \emptyset$. From this point of view, we should not really even accept the two given equations as a valid CRN model. In any case, on both counts (not being ACR-capable, and not being a valid CRN model), this Example simply does not constitute any sort of counterexample to the claims of our paper.

We do very much appreciate the lengths to which the reviewer has gone in order to make absolutely certain that all our claims are 100% accurate and watertight. But we do hope the reviewer is now willing to accept the correctness of our assertions. **Two variables** is always the right number of variables for assessing RPA capacity, the mathematical reason for which is given in our proof to Theorem 1. For **all** RPA-capable CRNs, the geometric projection of their rate equations onto **two variables** is generated by a single polynomial of the form $\rho = g(x_i, x_j)(x_i - c)$. Any CRN that does not satisfy this property cannot exhibit RPA - no exceptions, *un point c'est tout*.

3. The set-point may not be a rational function of parameters: In many places in the main paper and the supplement, the system's set-point is said to be a rational function of biochemical parameters. While this holds for most examples, consider the following birth death network

The dynamics is given by

$$\frac{dx_1}{dt} = c_1 - 2c_2x_1^2$$

and so the set-point

$$c = \sqrt{\frac{c_1}{2c_2}}$$

is not a rational function of c_1 and c_2 ? For more examples, see Examples 2.11 and 2.12 in [3]. Please revise as necessary.

Our response: We sincerely thank the reviewer once again for such a meticulous review of our work and for such careful attention to every possible technical detail. In this particular case, x_1^2 is a boundary variable since x_1 only appears in the CRN rate equations in the form of the monomial x_1^2 . Therefore, the setpoint for this variable is $c_1/2c_2$, which is a rational function of parameters, as expected. With boundary variables accounted for in this way, we can be assured that the setpoint of any RPA-capable variable will be a rational function of parameters, since we are working with a commutative **ring** structure (multivariate polynomials) with coefficients taken over a **field** (real numbers, with all coefficients considered symbolically). We allow only addition and additive inverses (subtraction) and multiplication without inverses (i.e. no division) of the ring elements; the coefficients admit addition/subtraction as well as multiplication/division (due to the field structure). No radicals may be taken.

But we do completely appreciate the reviewer's point, and that the boundary variable technicality might be lost on some readers (despite the fact that we do highlight the concept quite prominently). The setpoint of the **species concentration** may indeed be an algebraic (rather than a rational) function of parameters in these special cases.

Examples 2.11 and 2.12 in [3] also contain a **species** setpoint that is an n -th root of a rational function of parameters simply because the species in question contributes to an RPA-capable **boundary variable** in each case. In Example 2.11 – a continuation of Example 2.3 – x_A appears only in the form x_A^n . Likewise, in Example 2.12 – a continuation of Example 2.8 – x_B appears only in the form x_B^n .

We now add the following clarification of this matter to Remark 3 after the statement and proof to Theorem 1 (immediately following Definition 3, in Section S1.5 in our SI):

3. If the setpoint, c , depends only on biochemical rate constants and not on total concentrations (X_{ktot}) or production/degradation rates of any of the molecules, the system also has the capacity for ACR in the variable x_i .

Moreover, although we note in the statement and proof of Theorem 1 that c is a rational function of biochemical parameters, in the special case of an RPA variable that is a boundary variable, the setpoint of the corresponding RPA species may be an algebraic, rather than a rational, function of parameters. Consider, for instance, the one-species system: $\frac{dx}{dt} = c_1 - 2c_2x^2$; here x appears only in the form of the monomial x^2 , which is therefore a boundary variable. The 'setpoint' for the boundary variable, x^2 , is thus $\frac{c_1}{2c_2}$ - a rational function of parameters - while the setpoint for the species x is $\sqrt{\frac{c_1}{2c_2}}$.

We now also refer to this Remark in the main paper (at the end of the paragraph immediately following Eq. 6), where it is noted that c is a rational function of parameters.

4. **Questions on the Buchberger's Algorithm:** In the previous review round, I had raised the following issue regarding Buchberger's Algorithm

The Gröbner basis algorithm to find the RPA polynomial may not terminate. It is mentioned that failure to terminate for a chemical reaction network (CRN) is a prima-facie evidence that the CRN does not exhibit RPA. However this is not mathematically shown. In any case, checking for nontermination of a method is impractical.

In response the authors state in their rebuttal letter that the termination is guaranteed because "all ideals of a polynomial ring (or any Noetherian ring, for that matter) are finitely generated - a result of central importance in algebraic geometry, formalized by the Hilbert Basis Theorem". While I agree that termination is guaranteed, I must point out that the reason for my confusion and for raising this termination issue is the following excerpt from the previous version of the Supplement (submitted in round 1):

Failure of the Buchberger (or other Gröbner basis-computing) algorithm to terminate could thus be adduced as prima facie evidence that the CRN under consideration does not, in fact, have the capacity for RPA.

It seems that in this statement that authors meant "terminate in practical time". I understand that due to the "almost linear" nature of coordinate change, this algorithm would work well for RPA networks. However, does the final RPA polynomial produced by the method depend on the monomial ordering or other choices made by the method?

Our response: No, whether or not the elimination ideal referenced in our Theorem 1 is generated by an RPA polynomial is completely unrelated to the choice of monomial ordering. The form of the 'final RPA polynomial' is also unrelated to the

choice of monomial ordering: it is the *generator* of a *principal ideal*. But it's important to bear in mind that only monomial orderings that can *achieve* the desired elimination are suitable for testing this property (i.e. an *elimination ordering*). One cannot apply some completely arbitrary monomial ordering and expect to obtain (any) elimination ideal. We do refer to the highly accessible text by Cox et al. [7] in both our main text and in our SI for the benefit of any reader who wants to understand these technicalities fully. More importantly, we provide sample code at the end of our SI, with a discussion of a particularly efficient elimination ordering (using command '(dp(n-2), dp(2))' – see Section S5.1), should this be required for large and complicated CRNs.

Note that the reviewer's quotation from our original submission ('*Failure of the Buchberger (or other Gröbner basis-computing) algorithm to terminate could thus be adduced as prima facie evidence that the CRN under consideration does not, in fact, have the capacity for RPA.*') was removed from our revised submission, and is no longer included in our SI.

More importantly, if a network is not RPA, how many steps would the method need to confirm this non-RPA property? Such practical considerations must be discussed in the main text, and they are crucial for applying these ideas for characterising RPA in high-dimensional networks.

Our response: Of course there is no possible way to be specific about the number of steps that would be required to confirm the *inability* of some *general (non-RPA-capable)* CRN to exhibit RPA. If the algorithm is being applied completely mindlessly to a large and complicated CRN, with a random choice of two variables, then we quite agree that if the algorithm seems to be taking an inordinately long time to terminate, then the RPA-capacity or otherwise of the CRN is entirely unclear.

Now, the algorithm we present *can* be applied mindlessly, without any real understanding of the mathematical principles we discuss at length in our paper, and if the CRN under consideration is indeed RPA-capable, then such a completely mindless approach *will* be able to provide a confirmation of RPA capacity, along with an explicit identification of the setpoint. *If the sole* contribution of our paper were the development of an algorithmic test for RPA capacity, the reviewer would be quite correct to point out that this algorithm cannot easily distinguish between non-RPA-capable CRNs, and extremely large and complicated RPA-capable CRNs that could require a significant (and indeterminate) time-frame for the execution of the algorithm. But it's important to recognize that if the algorithm seems to be taking a long time to terminate, then our paper presents analytical approaches to check whether the CRN at hand has any potential for RPA capacity through a decomposition into independent subnetworks (wherever possible), and an analysis of deficiency in these subnetworks.

As an example, consider the CRN for the mammalian enzyme 6-phosphofructo-2-kinase/fructose-2,6-bisphosphatase (PFK-2/FBPase-2), which operates bifunctionally to both activate and inactivate fructose-2,6-bisphosphate (F2,6BP), as discussed by

Karp et al. [6] in terms of the potential RPA-capacity of F2,6BP. **As we will show, this CRN cannot exhibit RPA.** The CRN (presented in [6]), organized into its linkage classes, with $X_1 \equiv E$, $X_2 \equiv E\text{-ATP}$, $X_3 \equiv F6P$, $X_4 \equiv E\text{-ATP-F6P}$, $X_5 \equiv F2,6BP$, $X_6 \equiv E\text{-F2,6BP}$, $X_7 \equiv E\text{-ATP-F2,6BP}$, and $X_8 \equiv E\text{-ATP-F6P-F2,6BP}$, is given by

Interestingly, Karp et al. [6] claim that the Gröbner basis implementation in Mathematica does not terminate for this CRN, although they give no information as to how they've ordered their variables, or which monomial ordering they chose (although they almost certainly would have used a lexicographic monomial ordering since, prior to our work, it was not known that an elimination ideal involving just two variables is all that's required to test RPA capacity; the lexicographic ordering is generally an extremely computationally expensive elimination ordering since it produces a *full complement* of elimination ideals, in vast excess of what's actually required to solve the RPA problem). We used Singular and used an efficient block monomial ordering (to project onto **two** variables) that allowed the algorithm to terminate in **several seconds, from which we could confirm** that this particular CRN is not RPA-capable (at $X_5 \equiv F2,6BP$ or otherwise). We provide our Singular code below:

way of general advice to the reader, if one allows the algorithm to run for several hours (e.g. overnight) using the most efficient implementation possible (as described above) and it still hasn't terminated after this period, then one should consider undertaking some additional analysis of the CRN (as we provide below) through (i) first decomposing into algebraically-independent subnetworks (if possible), and then (ii) considering where deficiency arises within these independent subnetworks, and whether the deficiency corresponds to the presence of parallel pathways and/or feedback cycles, as required for RPA. Consider also that one should have a reason to suspect that a CRN exhibits RPA. It is clear from the detailed principles we develop here, as well as our previous topological analysis of RPA-capable network architectures at the network macroscale [12], that RPA-capable networks are actually very 'special', with very specific structural requirements, and are therefore extremely rare in the space of all possible networks. If one has a completely arbitrary network, with no particular reason to suspect it might be able to exhibit this special type of robustness, then it almost certainly doesn't. In the specific case we analyse here, the CRN involves a bifunctional enzyme (X_1) which both upregulates and downregulates its target protein (X_5); *a priori*, this fact presents the *possibility* that this CRN might have the capacity for RPA (even if, in the final analysis, it actually doesn't).

Now, recall that deficiency is a measure of the linear independence of the reactions of a CRN, relative to their distribution into the connected components (linkage classes) of a graph. In particular, the deficiency of a CRN is increased by one for every instance of a reaction being 'replicated' elsewhere in the network via a *different* set of complexes. Reversible reactions, for example, which duplicate a *single* reaction using the *same* pair of complexes, do not of themselves contribute to any deficiency increases. Recall also that, for an RPA-capable CRN, these replicated reactions must constitute either (i) a collection of parallel pathways, where the production/degradation (or interconversion between activation states) of the RPA-molecule must be orchestrated via different collections of reactions (involving different sets of complexes), or (ii) a cycle in the production/degradation (or interconversion between activation states) of a non-RPA-molecule (as regulated by the RPA-capable molecule), where the individual components of the cycle must be orchestrated via different collections of reactions (involving different sets of complexes). There's simply no other way for the right linear invariants to emerge! We highlight and discuss these principles repeatedly throughout the analysis of the worked examples in our SI.

Returning now to the non-RPA-capable CRN at hand: Note that the CRN for PFK-2/FBPase-2 has a deficiency of 5, comprising 14 complexes, 3 linkage classes and a rank of 6. By noting which reactions are 'replicated' (using different sets of complexes) to yield each of these 5 units of deficiency, it is straightforward to see that this CRN does not have the right structure to impose RPA on X_5 (or any other molecule). First, we note that the eight species of the model are all intricately interconnected in the nineteen reactions of the CRN, such that no decomposition into independent subsets is possible (see discussion of this point in S1.3 of our Supplementary Information). Moreover, in assessing the linear independence of the

CRN reactions, only one reaction in each pair of reversible reactions need be considered (for the reasons noted above pertaining to the deficiency-preserving nature of reversible reactions). Now, from a routine linear-algebraic analysis of the remaining reactions, it is easy to show that the reactions occurring at rates $k_1, k_3, k_5, k_6, k_{11}$ and k_{14} form a linearly independent set that spans the stoichiometric subspace of this CRN. The linear dependencies that give rise to the five units of deficiency are (with rate constants noted by underbraces to highlight the identity of each reaction):

$$1. \quad \underbrace{X_7 - X_5 - X_2}_{k_8} = - \underbrace{(X_2 - X_1)}_{k_1} + \underbrace{X_6 - X_1 - X_5}_{k_3} + \underbrace{X_7 - X_6}_{k_6}$$

$$2. \quad \underbrace{X_3 + X_2 - X_7}_{k_{10}} = \underbrace{X_2 - X_1}_{k_1} + \underbrace{X_1 + X_3 - X_6}_{k_5} - \underbrace{(X_7 - X_6)}_{k_6}$$

$$3. \quad \underbrace{X_8 - X_7 - X_3}_{k_{16}} = \underbrace{X_2 - X_1}_{k_1} - \underbrace{(X_6 - X_1 - X_5)}_{k_3} - \underbrace{(X_7 - X_6)}_{k_6} + \underbrace{X_4 - X_2 - X_3}_{k_{11}} + \underbrace{X_8 - X_4 - X_5}_{k_{14}}$$

$$4. \quad \underbrace{X_5 + X_6 - X_8}_{k_{18}} = - \underbrace{(X_2 - X_1)}_{k_1} - \underbrace{(X_1 + X_3 - X_6)}_{k_5} - \underbrace{(X_4 - X_2 - X_3)}_{k_{11}} - \underbrace{(X_8 - X_4 - X_5)}_{k_{14}}$$

$$5. \quad \underbrace{X_3 + X_4 - X_8}_{k_{19}} = \underbrace{X_6 - X_1 - X_5}_{k_3} + \underbrace{X_1 + X_3 - X_6}_{k_5} - \underbrace{(X_8 - X_4 - X_5)}_{k_{14}}$$

It is clear that these five distinct replicated reactions have no relationship whatsoever to either a collection of parallel pathways, to form a balancer module, or a cycle (or collection of interconnected cycles corresponding to an opposing set), to form an opposer module. None of these replicated reactions captures the production or degradation of a particular molecule, nor the interconversion between any one molecule and another molecule. This analysis thereby constitutes an independent confirmation that this CRN is not RPA-capable (at X_5 or otherwise).

We now provide some additional clarifications on the computational challenges of applying our algorithm in the main text. In particular, we now state that:

Although this method can be applied with any choice of two variables – one RPA-capable and one non-RPA-capable – we provide guidance in our SI on the decomposition of CRNs into independent subnetworks that can be analysed individually as to their RPA-capacity, and for which a particularly judicious choice of two variables can be made. **In fact, analysis of deficiency in such independent subnetworks can confirm the inability of a CRN to exhibit RPA, if such is the case, since large and complex CRNs may otherwise**

require a long (and potentially indeterminate) timeframe for the algorithm to terminate. We provide a fully analysed example of a non-RPA-capable CRN, to illustrate these principles, in SI Section S4.5.

We have also added in an entire new subsection to our SI (Section S4.5), entitled ‘A note on non-RPA-capable CRNs and computational challenges’, in which we summarise the commentary given above, and provide a full analysis of the PFK-2/FBPase-2 CRN (from Karp et al. [6]) as presented above.

5. Connection of examples to existing works: The authors start the Results section with “two simple examples that have eluded all previous systematic methods to detect RPA”. Please specify which systematic methods are being referred to here.

Our response: We now add in a number of key references at this point in our text to highlight which systematic methods are being referred to. In particular, the Shinar-Feinberg theorem provides a systematic way to determine ACR for deficiency-one networks containing two complexes that differ in a single species, but is silent on all CRNs with deficiency greater than one (and also cannot detect ACR in deficiency-one CRNs that *do not contain two complexes that differ in a single species*, such as the CRN we consider in Figure 8a in our paper). Cappelletti et al. provide a systematic method for detecting RPA, and a (single linear) integral controller, but is only applicable if an RPA polynomial is contained in the *rowspan* of the system. Even exceedingly simple CRNs (like the one we consider as Example 1 in our S3.1 of our SI, or the even simpler version we consider in Figure 8a of the main paper) elude this method if there is no RPA polynomial in the system’s rowspan. Eloundou-Mbebi et al. [13] present a necessary condition for RPA in CRNs of any deficiency, and thereby provide a systematic test that is generally able to identify some CRN species that certainly cannot exhibit RPA (which can thereby be excluded from further analysis). But this method is silent on whether any species that is not excluded by this test actually does exhibit RPA. In fact, all nine species in the deficiency-two Shinar-Feinberg EnvZ-OmpR model (considered in Figure 3 in our paper) satisfy the necessary condition proposed by Eloundou-Mbebi et al. [13], even though only one of these (pOmpR) actually *does* exhibit RPA. Gupta and Khammash [2] provide a universal characterization of maxRPA networks, but this approach is not applicable to any RPA-capable CRN that does not exhibit maxRPA. We discuss the universality of our approach in comparison with previous work in our Discussion section.

Of course, there are many other approaches to the analysis of CRNs, but none of these are ‘*systematic approaches to the study of RPA (or ACR)*’ in the sense that they can be applied to any CRN (of any deficiency, whether there is an RPA in the rowspan or not), without the need for guesses, or prior insight, and provide definitive information on the capacity for RPA and on the mechanisms for implementing it. The method for identifying ‘complex linear invariants’ developed by Karp et al. [6] can detect RPA in the deficiency-two Shinar-Feinberg EnvZ-OmpR model, but this is not a *systematic method* for RPA detection since those authors had to “guess” which complexes would end up in the invariants we identify via our

systematic approach. In fact, those authors state in Section 2.6 of their paper, referring to the above-noted example, in addition to another example they considered: ‘The two examples discussed above had already been analysed by other methods, so we had an idea of which invariants to expect and which subset of complexes to consider. For a new network such information may not be available, so how can non-trivial type 1 complex-linear invariants (simply, “invariants”) be found? The automatic procedure outlined in Section 2.2 can be used in principle but this becomes computationally infeasible when there are many complexes. We have found the following systematic procedure to be helpful on several examples’. The authors go on to provide some practical suggestions to the reader on computing useful ‘complex linear invariants’, which do not constitute a truly *systematic* procedure for rigorous application to *general* CRNs. Moreover, the ‘automatic procedure outlined in Section 2.2’ mentioned by these authors refers to computing a Gröbner basis for the system. But again, those authors did not account for algebro-geometric properties of RPA-capable CRNs that we identify in our paper, including the requirement for projection onto two variables only; thus, as a consequence, this systematic computation is now more widely applicable to CRNs (for the determination of RPA capacity, at least) than previously considered (where a lexicographic monomial ordering was necessary, producing a full collection of elimination ideals, since it was not previously recognized that only a specific elimination ideal is actually required to detect RPA). We emphasize again here the fact that these authors attempted to compute a Gröbner basis for a CRN involving the mammalian bifunctional enzyme PFK-2/FBPase-2, and found that their algorithm did not terminate; by contrast, by exploiting the use of a highly efficient elimination ordering (a consequence of our Theorem 1), we were able to compute a suitable Gröbner basis in under three minutes and thereby confirm that the CRN in question cannot exhibit RPA.

Also, in the example on Figure 2 it should be mentioned that X_3 is maxRPA and it can be checked from the characterisation result in [2]. Secondly since there is the reaction

which does not involve X_1 I do not understand why the term $k_6 X_2 X_3$ does not enter the expression for

$$\frac{d(X_1 - X_2)}{dt}$$

in Figure 2C. Please check.

Our response: We analyze the example in Figure 2 in careful detail in our Supplementary Information, where we provide a full discussion – first in Section S1.6 where we undertake a full algebro-geometric analysis to illustrate the consequences of Theorem 1 (the two-variable kinetic pairing theorem), then again in Section S4.4.2 where we show that the CRN’s set of eight reactions can be partitioned into two algebraically-independent subnetworks, and describe how various polynomial invariants may be determined within these subsets using an analysis of deficiency.

We also provide a complete Singular code for this example in Section S5.2, where we consider several different two-variable projections, and also show how the single concatenating monomial for this CRN can be computed automatically using the *lift* command in Singular.

As indicated in our Singular code (SI Section S5.2), the CRN depicted in Figure 2 of our paper (and analysed from a number of different viewpoints throughout our SI) induces the following set of four reaction equations under the law of mass-action:

$$\begin{aligned}\frac{dX_1}{dt} &= k_1R - k_2X_1X_2, \\ \frac{dX_2}{dt} &= k_3 - k_2X_1X_2 + k_4X_3, \\ \frac{dX_3}{dt} &= k_5 - k_6X_2X_3 - k_7O_1X_3, \\ \frac{dO_1}{dt} &= k_7O_1X_3 - k_8O_1.\end{aligned}$$

From this it follows that

$$\frac{d}{dt}(X_1 - X_2) = k_1R - k_3 - k_4X_3$$

We are unsure on the basis of the reviewer's comment why he/she feels the term $k_6X_2X_3$ should be present in the expression for $d(X_1 - X_2)/dt$. We have carefully checked this example (and all other examples in our paper) and can assure the reviewer that these reaction forms, and the expression for $d(X_1 - X_2)/dt$ in particular, are all completely correct.

On the matter of X_3 being maxRPA, although this is certainly true, this is not quite the point of Figure 2, which is attempting to provide a representation of the general properties of all CRNs that act an Opposer Module (which could contain any number of opposer 'nodes', in principle, organized into an opposing set structure). For this simple example, a two-node opposing set is shown, where the two opposer nodes are as simple as possible (hence the distal opposer is indeed maxRPA). Since X_3 and X_2 regulate each other (and thereby form a feedback loop), and since X_2 contributes to a different opposer node (regulated by R , upon which RPA is ultimately conferred), one cannot generally consider the reactions involving O_1 and X_3 in isolation before ensuring that the CRN as a whole has been correctly decomposed into algebraically independent subnetworks. We undertake this analysis carefully in Section S4.4.2. As shown, this allows us to determine the correct algebraic invariant (involving O_1 and X_3) to be 'passed' to the other opposer node (involving R).

If the term $k_6X_2X_3$ is present then it cannot be eliminated with the concatenating monomial O_1 . On the other hand, if this reaction is removed (i.e. this term is absent) then the overall network simply becomes a trivial RPA network where the output of one RPA network (i.e. the network comprising $X_3 - O_1$) is passed as an input to another RPA network (i.e. the antithetic network with $X_1 - X_2$). It is straightforward that connecting RPA networks in

series (with catalytic reactions) would still result in a RPA network. Such examples are not appropriate for demonstrating the novel results in this paper.

Our response: We offer several additional points of clarification regarding the reaction occurring at rate k_6 , which contributes the term $k_6X_2X_3$ to the rate equations, and the absolute necessity of including this reaction in order to confer RPA on the CRN (including the entire ‘embedded network’).

First, regarding the elimination of $k_6X_2X_3$, and the identity of the requisite concatenating monomial, we’d like to point out that the four mass-action equations corresponding to this CRN (noted above in our previous comment) can be analysed systematically in Singular, from which the concatenating monomial can be computed automatically. This is explained in detail, including the use of the Singular command *lift* (which is used to identify the concatenating monomial) in our analysis of the Singular code at the end of Section S5.2.

Second, we *cannot* remove the reaction $X_3 + X_2 \xrightarrow{k_6} X_2$ as it is *necessary* for RPA in the “full” (i.e. embedded) network if the molecule O_1 is responsible for regulating the embedded network, as indicated in Figure 2(a,d). The reviewer points out that “if this reaction is removed ... then the overall network simply becomes a trivial RPA network where the output of one RPA network (i.e. the network comprising $X_3 - O_1$) is passed as an input to another RPA network (i.e. the antithetic network with $X_1 - X_2$). It is straightforward that connecting RPA networks in series (with catalytic reactions) would still result in an RPA network.” This is not actually true if O_1 is responsible for regulating the embedded network, as we will explain below. But if, instead, X_2 is responsible for regulating the embedded network, then the reviewer is entirely correct. In this latter case, and with the reaction $X_3 + X_2 \xrightarrow{k_6} X_2$ removed, the reactions involving O_1 and X_3 are completely isolated from any disturbances to the embedded network (i.e. there is no ‘signal detection’ in these reactions), and can therefore be considered at their steady-state ($X_3 = k_8/k_7$, $O_1 = k_5/k_8$). The controller mechanism thereby reduces to the well-known antithetic-integral controller structure, with $d(X_1 - X_2)/dt = k_1R - k_\alpha$, where $k_\alpha = k_3 + k_4k_8/k_7$. This constitutes a single opposer node, obtained via a single linear coordinate change. Nevertheless, it is clear that much more complicated opposer mechanisms could arise, and that a single opposer node is insufficient to characterize the space of all possible Opposer modules orchestrated by CRNs. Note also that the molecule O_1 has a fixed concentration (k_5/k_8) in the scenario just considered; thus, if O_1 were to regulate the embedded network (rather than X_2), then RPA would not be possible at R since the antithetic controller would no longer be able to regulate the embedded network via a feedback loop. It’s important to recognize that what we are actually analyzing in Figure 2 is just the *controller* part of the overall network. Considered in isolation, with R (the ‘sensor’ molecule from the ‘embedded network’) as the input, the network will not actually **exhibit** RPA at R unless the CRN is embedded into the requisite feedback structure, as indicated in Figure 2(a,d). (We do discuss the distinction between being *RPA capable* and *exhibiting RPA* in SI Section S1.5 – see Definition 3).

We make it clear in the main text that we consider X_3 to be a *network protein*, which can therefore be affected in some way by disturbances to the embedded network.

This is possible due to the inclusion of the reaction $X_3 + X_2 \xrightarrow{k_6} X_2$, whereby the molecule X_2 downregulates the molecule X_3 ; this reaction permits a controller structure in which (in contrast to the scenario considered above) O_1 is **not** RPA-capable, (unlike X_3 , which *is* RPA-capable either way) and is thereby able to provide the necessary ‘actuation’ to the embedded network. The resulting CRN architecture, corresponds to what we previously described [12] as a *two-node opposing set*. The architecture of an opposing set consists of a collection of *interlinked* feedback loops, embedded into the feedback portion of an overarching feedback loop into which the ‘controlled’ system is embedded. Our current paper now makes clear why the interlinking of feedback loops is required in opposing sets (which was not entirely clear from our prior topological analysis [12]): signal transfer from the ‘distal’ opposer to the ‘proximal’ opposer (see Figure S12 in our Supplementary Information) is required for the *passing of invariants*, using concatenating monomials as needed; signal transfer from the proximal opposer to the distal opposer is necessary to embed these interactions in a feedback loop.

The second example in Figure 3 seems to be taken straight from [4] (see Fig. 2). This should be clearly stated when the example is being introduced in the main text and also in the caption of Figure 3. Also mention that the linear invariants shown in Fig. 3 can be deduced from the approach in [4] (this is stated in passing in the conclusion but it should be stated more prominently when the example is being discussed.)

Our response: No, we most assuredly did not just take our example in Figure 3 straight from [4]. This example was considered by Shinar and Feinberg in the Supplementary Materials to their seminal paper in Science [10], and we refer to this fact in-text in our discussion of Figure 3. We now add in an additional reference to this fact in our caption to the Figure, to ensure that this is absolutely clear. Perez-Millan et al. [4] also discuss the fact that this example is taken from the SM of Shinar and Feinberg’s paper. We are wondering if the reviewer might actually mean Karp et al. [6] here? In any case, Karp et al. [6] also take the example from the SM of Shinar and Feinberg’s paper, and those authors also make it clear that the Shinar-Feinberg paper is the source of the example. This deficiency-two model of the EnvZ-OmpR motif (due to Shinar/Feinberg) has been extensively discussed in the literature (including by [4] and [6], who have also analysed this model using different theoretical viewpoints from the one we consider here). Neither of those prior viewpoints provides a clear and definitive way to connect the properties of this particular example to the properties of all possible RPA-capable CRNs, however. More generally, it has been a subject of tremendous interest to discover robust and systematic ways to handle ACR/RPA in CRNs of deficiency greater than one.

We’re not quite so sure that the two linear invariants can be deduced straightforwardly from the approach in [4]. The connector invariant actually involves three terms, not two, so cannot arise from the theory of toric ideals. Again, we

suspect the reviewer might be referring to the approach by Karp et al. [6]. While it is certainly true that Karp et al. [6] were able to identify the linear invariants relevant to Figure 3, this was only possible once it was clear which complexes/monomials are involved in these invariants (on the basis of the solution originally obtained by Shinar and Feinberg [10] through manual substitution, and possibly also through the analysis of the CRN undertaken by [4], since both papers [4] and [6] have authors in common). But more importantly, it is entirely unclear from the invariants computed by Karp et al. [6] how this solution relates to the complete solution space of all RPA-capable CRNs! *This is the fundamental idea we are attempting to communicate through these simple examples in Figures 2 and 3.* And again, we stand by our claim that prior to our work, there has been no **systematic** way (not involving ‘guesses’ or Ansätze, or prior analysis via other methods) to analyse **general** CRNs, of arbitrary deficiency, for which an RPA polynomial is not contained in the rowspan of the system.

6. Many claims without proofs in the Supplement: The Supplement has been considerably revised, but still many arguments are unclear because proper proofs have not been provided or referenced:

- Why should eq. (8) hold when eq. (9) holds? Can the networks always be partitioned this way? Please explain.

Our response: We have scrutinized our exposition in Section S1.3 of our SI, and agree with the reviewer that Eq. (8) is not guaranteed to hold when Eq. (9) holds. Unlike the rest of the material in this section, this specific detail was not actually part of Martin Feinberg’s treatment in Appendix 6.A of his text. We have now removed this detail from our SI, as it does not follow from the preceding results. In many of the examples we analyse in our paper, the deficiency does partition into the same subsets obtained from a partition of CRN rank; but this might not obtain in general. We have carefully worked through our entire manuscript and SI to ensure that our discussion of deficiency is accurate throughout (and does not make use of what was formerly Eq. (8)). We do thank the reviewer once again for such a meticulous review of every technical detail in our paper.

- On page 21 in the Supplement it states that:

Since a perturbation to the CRN that alters the steady state of x_j will also alter the steady-states of other non-RPA capable variables (eg. x_m), If $\mathbb{R}[\bar{x}]$ will contain polynomials in x_j and x_m , and that are not contained in $I_f \cap I_p$.

Why should such a perturbation always exist?

Our response: When we say *perturbation* here, we are simply referring to a disturbance that the CRN is able to ‘reject’, thereby exhibiting RPA in one or more variables. It’s really only meaningful to talk about RPA in the context of a CRN subjected to some sort of disturbance which alters the steady-state of the system. If the CRN is not subjected to any disturbance at all, the CRN can only exhibit a ‘trivial’ form of RPA (since nothing in the network can ever change).

If the reviewer is querying why there should always be at least one variable that is non-RPA-capable, then we emphasize again that this is a key component of RPA. If all variables in the network are ‘fixed’ at steady-state, and can never be altered by *any* disturbance, then the network is not able to adapt to disturbances (and there can be no ‘internal model’, which recapitulates the disturbance).

Of course, special cases of CRNs could exist with only *one* non-RPA-capable variable, rather than two or more, in which case the statement above vacuously holds (*ex falso quodlibet*). In any event, the selection of variables taken for the projection consists of *one* non-RPA-capable variable, in addition to the putative RPA-capable variable.

- Also on page 21 in the Supplement it says that

The set \bar{x} now contains two independent (uncoupled) variables in the sense that a perturbation to the CRN that alters the steady-state of one of the variables does not affect the steady-state of the other.

Why does this hold? Please elaborate.

Our response: Here we are referring to the fact that \bar{x} now contains two variables – one RPA-capable and one non-RPA-capable. The two variables are *ipso facto* uncoupled since a disturbance to the system that alters the steady-state of the non-RPA-variable necessarily has no effect on the steady-state of the RPA-variable (whose setpoint is independent of the disturbance).

- In general, in the proof of the Kinetic Pairing Theorem the authors work over the ring of polynomials over species-variables x_1, \dots, x_n . Shouldn't the system parameters (i.e. rate constants) be included in this ring, as they would appear in the RPA polynomial? This inclusion of parameters is there in the Singular code but not in the proof. However simply adding the parameters in the ring is not sufficient as the n -th roots of the parameter would be added, as the examples mentioned in point 3 show.

Our response: In general, the ring of interest in these problems is the ring of polynomials in n variables, with real-valued coefficients – i.e. $\mathbb{R}[x_1, \dots, x_n]$. In this case, since we want to handle all coefficients (parameters) symbolically, we adjoin the coefficients to the ring with the algebraic structure of a fraction field. Notice in our Singular code that we declare two sets, each enclosed by round brackets (). The first of these, which declares the symbolic parameters, begins with a zero, which declares the characteristic of the field. Characteristic zero fields include the set of

real numbers, but also the rationals (including fraction fields). This is a standard way to handle symbolic coefficients in algebro-geometric computations of this type. Since these coefficients come from a field, we can apply all field operations (including division/multiplicative inverses) to these during the execution of any algorithms. By contrast, monomials/polynomials can only be added/subtracted and multiplied (no division) since these are defined to be elements of a ring.

Of course, if one particularly wants to define a parameter k to be a variable of the model for analytical purposes, one can certainly make this choice if desired, and consider the ring $\mathbb{R}[x_1, \dots, x_n, k]$. This allows the ring to incorporate k within the power products (monomials), instead of interpreting it as a coefficient in the field over which the ring is defined. This could be useful if one particularly wanted to project the system onto k (as the non-RPA-capable variable) in the application of Theorem 1 – which might be helpful if it is difficult to posit a suitable non-RPA variable *a priori* (for a complicated new CRN that one suspects to be RPA-capable), and if the parameter k is a CRN property that can be used as a disturbance. But even so, it's important to recognize that there's no need to declare specific biochemical parameters to be variables of the model to check if the CRN can adapt to these as disturbances. If the CRN is RPA capable, then it can adapt to any disturbance that is not present in the identified setpoint. By contrast, if the parameter *is* in the setpoint, the CRN **cannot** adapt to perturbations to that parameter. We mention this point explicitly in our definition of RPA.

7. Other minor issues:

- The definition of RPA must be shifted to the main text due to its centrality in understanding the message of the paper.

Our response: We have now shifted the definition of RPA to the main text. We sincerely thank the reviewer for such helpful and supportive suggestions.

- Why is the variable x_i missing in $g(x_i)$ in figure 4 (main text) and figure S4 (supplement)?

Our response: We had stated in the caption to Figure 4 (main text) that 'when x_j is the diverter node of the Balancer module, g will generally be zero-order in x_i ' and in the caption to Figure S4 in our SI that 'for clarity, we depict a functional form for g that is zero order in x_i , since this is the most common form obtained in practice (i.e. provide the RPA variable x_i is not autoregulatory). The main reason for presenting the simplified form of g that usually obtains was to emphasize the topological role of x_j (relative to x_i) in orchestrating a 'pairing' action. This simplification is really not essential, however, and our exposition of the pairing principle is unaffected if we include the general form of the pairing function, $g(x_i, x_j)$. In response to the reviewer's query, we have now updated this figure in both the main paper (Figure 4) and the SI (Figure S4) to feature the general form of the pairing function, $g(x_i, x_j)$, and have updated our captions accordingly.

- Replace “consistutes” with “constitutes” on line 72 in the main text. Please run a spell check.

Our response: We thank the reviewer for spotting this typo, which has now been corrected. We have run a spell check, and there are no additional typos.

- The paper says that if a network is RPA the integrator is guaranteed to exist. For example the following on page 22 in the main text
In principle, there should always exist some single nonlinear coordinate change to extract a single output-driven internal model (Fig. 9a) from systems rate equations, corresponding to a single integral of the systems tracking error (Fig. 9b)

or the following on page 16 in the Supplement

All classes of RPA, including ACR, thus require some form of integral control.

Please explain which version of IMP can be used to verify this existence. See [1] for a recent review on IMP.

Our response: Here we are referencing a long-standing idea in the systems biology literature (see, for example [8] and [9], and Section 3 of [1]) that, where RPA obtains, there ‘should’ exist some (generally nonlinear) coordinate change that recasts the system into integral feedback form (if necessary, if the system in question has a feedforward architecture, and is subjected to a disturbance at its ‘diverter’). The integral in question should operate on the system’s tracking error. Reference [8], in particular, provides a very detailed mathematical elaboration of these ideas, which hold for nonlinear systems under several technical assumptions (which are very mild in the context of CRNs). Both [1] and [8] apply these ideas to identify a nonlinear transformation that identifies an output-driven internal model for a simple CRN with a feedforward structure, through a recasting into feedback form. We reviewed this simple example in our response to the Reviewer’s Point 1. Again, our paper provides a very **different** viewpoint, **distinct** from the notion that *‘there should always exist some single nonlinear coordinate change to extract a single output-driven internal model ... corresponding to a single integral’*. Our reason for referencing the viewpoint of prior control theoretic literature [1,8,9] is to contrast it with our new completely different viewpoint, and not to present that prior viewpoint as axiomatic.

We now add in these three references [1,8,9] at the location noted by the reviewer, to emphasize the prior IMP viewpoint we draw upon here.

- On page 20 the authors state that “It is striking to note that the original form of the CRN (Fig. 8a) eludes the Shinar-Feinberg theorem, even though the CRN exhibits ACR and has a deficiency of one.” However does it fit the results in [4]? Please comment on this. Also explain why the authors found the method for finding linear invariants “ad hoc” (lines 568-569 on page 25).

Our response: As noted in many of our previous responses, we suspect the reviewer is actually referring to Karp et al. [6], rather than Perez Millan et al. [4], in this comment. Certainly, once one knows which particular complexes (and hence, which monomials) one needs to include in any RPA/ACR-relevant invariants, one can certainly use the method developed Karp et al. [6] to compute those invariants. In this specific simple case, however, the invariants in question happen to be the actual rate equations dA/dt (the connector polynomial) and dB/dt (the balancer polynomial), so the method is hardly necessary.

Regarding our description of the Karp et al. [6] method for finding linear invariants as ‘ad hoc’, those authors themselves acknowledge that this method relies on knowing ahead of time which complexes (monomials) are relevant. As we had noted in one of our earlier responses, the authors specifically state that the ‘examples (analysed in that paper) had already been analysed by other methods, so we had an idea of which invariants to expect and which subset of complexes to consider. For a new network such information may not be available, so how can non-trivial type 1 complex-linear invariants (simply, “invariants”) be found?’. The authors go on to provide some practical suggestions for the reader on computing useful ‘complex linear invariants’, which may be helpful in studying new CRNs but certainly do not constitute a truly systematic procedure for rigorous application to general CRNs.

- On the Supplement page 15 it is stated that *Mass-conservative CRNs therefore have no external stimuli or inputs, and can only be perturbed by altering the total abundances (or concentrations) of the constituent molecules - i.e. by altering the initial conditions.*

Why cannot the perturbation come in the form of parameter variation, e.g. of a conversion reaction (that conserves mass).

Our response: Yes, the reviewer certainly does make an interesting point. In practice, conversion reactions are normally mediated by enzymes, so if the enzyme in question is not part of the network, we agree that the corresponding parameter (which reflects the concentration of the enzyme) could be perturbed. Almost any parameter of a CRN could be perturbed in principle. But the framework we develop here emphasizes that a CRN can adapt to perturbations to any parameter that does not appear in the RPA-variable’s setpoint, regardless of whether the CRN is mass-conservative or not.

We did not intend the statement in question to constitute a major technical point. Nevertheless, in response to the reviewer’s comment, we have adjusted the wording of that statement to read: ‘Mass-conservative CRNs therefore have no external stimuli or inputs, and are typically perturbed by altering the total abundances (or concentrations) of the constituent molecules – i.e. by altering the initial conditions’. We do thank the reviewer again for such meticulous attention to every possible technical detail.

Final comments to the reviewer: We would like to sincerely thank the reviewer again for the incredible generosity he/she has shown us by scrutinizing our work in such extraordinary detail. We trust the reviewer will now be satisfied that our claims are all accurate and well-substantiated, and can now agree that our paper is ready for publication.

References

- [1] M. Bin, J. Huang, A. Isidori, L. Marconi, M. Mischiati, and E. Sontag. Internal models in control, bioengineering, and neuroscience. *Annual Review of Control, Robotics, and Autonomous Systems*, 5:55-79, 2022.
- [2] A. Gupta and M. Khammash. Universal structural requirements for maximal robust perfect adaptation in biomolecular networks. *Proceedings of the National Academy of Sciences*, 119(43):e2207802119, 2022.
- [3] N. Meshkat, A. Shiu, and A. Torres. Absolute concentration robustness in networks with low-dimensional stoichiometric subspace. *Vietnam Journal of Mathematics*, 50(3):623-651, 2022.
- [4] M. Perez Millan, A. Dickenstein, A. Shiu, and C. Conradi. Chemical reaction systems with toric steady states. *Bulletin of mathematical biology*, 74(5):1027-1065, 2012.
- [5] M. S. Perez Millan. *Metodos algebraicos para el estudio de redes bioquimicas*. PhD thesis, Universidad de Buenos Aires. Facultad de Ciencias Exactas y Naturales, 2011.
- [6] R.L. Karp, M. Perez Millan, T. Dasgupta, A. Dickenstein, and J. Gunawardena. Complex-linear invariants of biochemical networks. *Journal Theoretical Biology*, 311:130-138, 2012.
- [7] D. Cox, J. Little, and D. O’Shea. *Ideals, varieties, and algorithms: an introduction to computational algebraic geometry and commutative algebra*. Springer Science & Business Media, 2013.
- [8] E.D. Sontag. Adaptation and regulation with signal detection implies internal model. *Systems & Control Letters*, 50:119-126, 2003.
- [9] O. Shoval, U. Alon, and E. Sontag. Symmetry Invariance for Adapting Biological Systems. *SIAM Journal on Applied Dynamical Systems*, 10: 857-886, 2011.
- [10] G. Shinar and M. Feinberg, M. Structural sources of robustness in biochemical reaction networks. *Science*, 327(5971):1389-1391, 2010.
- [11] D. Cappelletti, A. Gupta and M. Khammash. A hidden integral structure endows absolute concentration robust systems with resilience to dynamical concentration disturbances. *Journal of the Royal Society Interface*, 17(171):20200437, 2020.
- [12] R.P. Araujo and L.A. Liotta. The topological requirements for robust perfect adaptation in networks of any size. *Nature Communications*, 9(1):1-12, 2018.
- [13] J.M. Eloundou-Mbebi, A. Küken, N. Omranian, S. Kleessen, J. Neigenfind, G. Basler and Z. Nikoloski. A network property necessary for concentration robustness. *Nature Communications*, 7(1):1-7, 2016,

Reviewer #1 (Remarks to the Author):

I went over the revised version of the paper, and while I truly appreciate the efforts of the authors in preparing the revision, I am still of the opinion that this paper is not suitable for an interdisciplinary journal like Nature Communications. The main issues I have are as follows:

1. As the title of the paper suggests, the aim of this study is to identify embedded integral control in adapting circuits. While the paper does construct a series of subsidiary integrators within the network via linear coordinate transformations, it does not provide a global coordinate transformation that would identify a global integrator. Hence the connection with standard integral controllers, and the standard Internal Model Principle is tenuous. By algebraically showing the existence of an RPA polynomial, only regulation and tracking are established, not integral control.

2. The proof of the Kinetic Pairing Result is hard to understand and verify for non-specialists in algebraic methods (like myself). In fact, when I showed the proof to a specialist in such methods (with the permission of the editor), that person also found the proof particularly difficult to understand. Here I am specifically referring to pages 20-21 in the Supplement. This difficulty was mentioned to the authors in the previous review report, but the authors have not addressed it by simplifying/elaborating on the proof. Therefore, after two rounds of revisions (and more than 2 months of time spent on this paper!) I still cannot ascertain the correctness of the reported results.

3. Even if the Kinetic Pairing result is correct, the final outcome is not a clean characterization of adapting networks. The characterization is done via an algorithm whose termination can take an inordinately large time for even moderately-sized networks. In cases where the algorithm does not terminate in a reasonable time, the authors suggest using analytical approaches like decomposing the large network into several independent networks and then performing deficiency analysis. However, this type of analysis is beyond the scope of most readers who might be interested in this work. If possible, the authors should try to extend their algorithm by incorporating this analysis.

Reviewer #2 (Remarks to the Author):

The manuscript by Araujo and Liotta addresses both a fundamental biological problem, and a fundamental conceptual innovation.

The biological problem is that of organisms' ability to adapt to changes in the environment or their physiological state. This is a hallmark of many signalling processes where adaptation is required to safeguard the appropriate response. This has been a longstanding problem across many biological areas.

The conceptual advance offered by Araujo and Liotta is to derive mathematical results and conditions that guarantee adaptive behaviour. This is a tour de force and highly innovative; I like the idea of generalising integral control to dissect the design principles of adaptive behaviour a lot - there is scope for further applications of this framework. Many areas of biology will benefit from a more mathematically framework, and this work is a major contribution in this direction. Being able to prove formally aspects that biological systems have to obey to exhibit certain behaviour is exciting. In addition to the authors' earlier - outstanding in my view - work (PMID: 29717141), there is a growing body of other exciting work in this domain (see e.g. PMID: 19536158; PMID: 20223989, PMID 30194237); the present manuscript makes a substantial contribution to this important area of research. What sets the current work apart from these earlier studies are the more general biological relevance of adaptive behaviour.

The response to the reviewers' comments now exceeds the manuscript in length and I was unable to follow some of the criticisms voiced by the other reviewer. It is clear that the ideas underlying the alternative formulation of an integral controller are highly technical but the derivations contain the appropriate level of detail and I was able (admittedly with some work and effort) to follow the arguments laid out in the manuscript and especially the now substantial supplementary material.

I can understand that the references to additional mathematical work and proofs (e.g. in Refs 7 and 15 in the supplementary material) does potentially add problems for a reader in reconstructing the full mathematical framework. In my view the authors negotiate the difficulties of exposing and explaining their proof in a clear and laudable manner. Developing a mathematical framework for a multidisciplinary audience is challenging but the work presented here provides important pointers how we can achieve this and find common ground between the often opposing needs of mathematical and biological audiences.

Minor point:

Lines 438-442 contain in my reading one of the essential messages of this paper and I would suggest that the authors stress this and make the innovation in the paper more explicit.

In summary, this is exactly the type of mathematical biology that I and many others want to see.

Response to Reviewers

Reviewer #1

I went over the revised version of the paper, and while I truly appreciate the efforts of the authors in preparing the revision, I am still of the opinion that this paper is not suitable for an interdisciplinary journal like Nature Communications. The main issues I have are as follows:

1. As the title of the paper suggests, the aim of this study is to identify embedded integral control in adapting circuits. While the paper does construct a series of subsidiary integrators within the network via linear coordinate transformations, it does not provide a global coordinate transformation that would identify a global integrator. Hence the connection with standard integral controllers, and the standard Internal Model Principle is tenuous. By algebraically showing the existence of an RPA polynomial, only regulation and tracking are established, not integral control.

Our Response:

We sincerely thank the reviewer again for such a significant investment of his/her time in considering our work. We would like to emphasize once again, however, that there simply is no (and can be no) global coordinate transformation (and hence, no single 'global integrator') that can reveal the general properties of all possible RPA-capable CRNs. As we explained in our most recent response, one of the major points of our article is that the conventional approach to control theory, with its 'standard integral controllers' and 'standard Internal Model Principle' (as the Reviewer calls them) is wholly inadequate to capture the fundamental design principles that organize all forms of biological complexity into robustness-promoting (and ultimately, survival-promoting) structures. What we demonstrate instead is that all RPA-capable chemical reaction networks – without exception - are constructed from a topological hierarchy of building blocks ('invariants'), which independently implement the Internal Model Principle by robustly rejecting disturbances to specific network features (e.g., a ratio in the concentrations of specific molecules constituting a balancing mechanism) and which thereby work together collaboratively to implement RPA on specific molecule(s). Our analysis identifies the remarkable fact that all such RPA-promoting subsidiary invariants are obtained via linear coordinate changes, and recapitulate – *locally* within the topology of the network – the dynamical properties of the disturbance. By proving that all RPA-capable CRNs are decomposable into these well-defined topological hierarchies of internal models, we provide a universal description for the implementation of integral control that holds for all collections of chemical reactions that exhibit RPA (including any special case of RPA, such as absolute concentration robustness (ACR)).

2. The proof of the Kinetic Pairing Result is hard to understand and verify for non-specialists in algebraic methods (like myself). In fact, when I showed the proof to a specialist in such methods (with the permission of the editor), that person also found

the proof particularly difficult to understand. Here I am specifically referring to pages 20-21 in the Supplement. This difficulty was mentioned to the authors in the previous review report, but the authors have not addressed it by simplifying/elaborating on the proof. Therefore, after two rounds of revisions (and more than 2 months of time spent on this paper!) I still cannot ascertain the correctness of the reported results.

Our Response:

We thank the Reviewer for clarifying further the section of the proof they found hard to understand. This gives us the opportunity to restate, highlight, and summarize again the several additional clarifications of the proof we provided in the previous rounds of response. In fact, our extensive additional clarifications on the proof to our Theorem 1 (the Kinetic Pairing theorem) were added after the first round of review, through a series of 'Remarks' following the proof in our Supplementary Information.

In particular, the Reviewer stated in his/her first report that "...The proof given on page 12 in the Supplement does not satisfactorily explain why the ideal $I_f \cap \mathbb{R}[\bar{x}]$ will contain polynomials in x_j and x_m that are not in $I_f \cap I_p$ ", and proceeded to give an example of a CRN that he/she believed to be a counterexample to our Theorem. We responded with a detailed explanation as to why 'the ideal $I_f \cap \mathbb{R}[\bar{x}]$ will contain polynomials in x_j and x_m that are not in $I_f \cap I_p$ ', along with a detailed analysis to demonstrate that the Reviewer's example is entirely consistent with our Theorem. The Reviewer responded in the following round by challenging us with three *new* CRN examples which he/she proposed were counterexamples to our Theorem. We responded by providing a careful and detailed analysis of these new examples, and demonstrating that these were entirely consistent with our Theorem - not counter-examples. In fact, we used our detailed analysis of the Reviewer's various CRNs to underscore the fact that these illustrations demonstrate how (and why) the Theorem works.

The Reviewer also now provides new information about consulting with a 'specialist' third-party ('...when I showed the proof to a specialist in such methods ... that person also found the proof particularly difficult to understand'). Nevertheless, we respectfully point out that the Reviewer has not been very specific as to what he/she does not understand in our proof to the Kinetic Pairing Theorem beyond the initial question regarding "polynomials in x_j and x_m that are not in $I_f \cap I_p$ " (which we addressed thoroughly, as noted above). It is also possible that extensive clarifications of our proof in past rounds of responses, along with our lengthy and detailed analyses of the CRN examples, might not have been shown *in extenso* to the Reviewer's consultant.

3. Even if the Kinetic Pairing result is correct, the final outcome is not a clean characterization of adapting networks. The characterization is done via an algorithm whose termination can take an inordinately large time for even moderately-sized networks. In cases where the algorithm does not terminate in a reasonable time, the

authors suggest using analytical approaches like decomposing the large network into several independent networks and then performing deficiency analysis. However, this type of analysis is beyond the scope of most readers who might be interested in this work. If possible, the authors should try to extend their algorithm by incorporating this analysis.

Our Response:

We respectfully emphasize once again that within the original and revised manuscript versions we have provided detailed and rigorous mathematical arguments to support the correctness of the Kinetic Pairing result, and have additionally provided detailed analyses of all the new CRN examples suggested to us by the Reviewer to demonstrate how they reflect the essential underlying principles of this key Theorem.

In addition, the Reviewer suggests that “the final outcome is not a clean characterization of adapting networks” and that the “characterization is done via an algorithm ...”. We feel it is important to clarify for the record that the universal characterisation of adaptation-capable CRNs presented in our study is not obtained from an algorithm, but from our carefully-developed mathematical arguments that all such adapting networks are constructed from topological building blocks that are obtained by linear coordinate changes (associated with linear integral controllers). Since all RPA-capable CRNs must be constructed from these fundamental building blocks, and because the topological principles governing RPA capacity are now known in complete generality, our findings definitively characterise the full set of chemical reaction structures that can implement RPA. In other words, through identifying the fundamental building blocks of RPA in CRNs and the algebraic structures (in terms of CRN deficiency, etc.) by which these building blocks are constructed, as well as the general principles by which the building blocks are connected together into larger networks, we are able to provide a comprehensive (i.e. universal) description all possible RPA-capable CRNs at the level of intermolecular interactions.

Now, for a specific RPA-capable CRN under consideration, we present an algorithmic test that can demonstrate the ability of a particular molecule to exhibit RPA, in addition to identifying the decomposition (of the requisite nonlinear transformation of the system) into subsidiary linear controllers, as well as the ‘setpoint’ of the system as a function of parameters. This is a completely separate matter from the issue of the ‘universal characterisation’ of RPA-capable CRNs. Although the reviewer suggests that the termination of the algorithm ‘can take an inordinately large time for even moderately-sized networks’, we respectfully emphasize once again that it is, in fact, only non-RPA-capable CRNs that may require an indeterminate (and potentially impractical) timeframe for the algorithm to terminate. We clearly explain the technical reasons for this fact in our paper (in both the main article, as well as in the Supplementary Information), and provide an alternative analytical approach to establish the *inability* of a CRN to exhibit RPA, should our ‘general’ algorithm take too long to terminate. We respectfully point out that this ‘deficiency analysis’ requires only linear algebraic techniques, and is thus

accessible to the mainstream scientific community. We provide a fully-worked example of the application of this alternative analytical approach to a specific CRN in our Supplementary Information.

The complete generality of our findings on RPA-capable chemical reaction networks, along with the clarity and comprehensiveness of our analysis, will now enable the scientific community to consider fundamental questions about the evolution of biochemical reaction networks at the whole-cell level, and at the level of entire organisms. As such, we believe strongly that this study will be of interest to researchers from a huge variety of different fields, including evolutionary biology, bioengineering, cancer research, developmental biology, pharmacology, whole-cell modelling, and many others.

Reviewer #2

The manuscript by Araujo and Liotta addresses both a fundamental biological problem, and a fundamental conceptual innovation. The biological problem is that of organisms' ability to adapt to changes in the environment or their physiological state. This is a hallmark of many signalling processes where adaptation is required to safeguard the appropriate response. This has been a longstanding problem across many biological areas.

The conceptual advance offered by Araujo and Liotta is to derive mathematical results and conditions that guarantee adaptive behaviour. This is a tour de force and highly innovative; I like the idea of generalising integral control to dissect the design principles of adaptive behaviour a lot - there is scope for further applications of this framework. Many areas of biology will benefit from a more mathematically framework, and this work is a major contribution in this direction. Being able to prove formally aspects that biological systems have to obey to exhibit certain behaviour is exciting. In addition to the authors' earlier - outstanding in my view - work (PMID: 29717141), there is a growing body of other exciting work in this domain (see e.g. PMID: 19536158; PMID: 20223989, PMID 30194237); the present manuscript makes a substantial contribution to this important area of research. What sets the current work apart from these earlier studies are the more general biological relevance of adaptive behaviour.

The response to the reviewers' comments now exceeds the manuscript in length and I was unable to follow some of the criticisms voiced by the other reviewer. It is clear that the ideas underlying the alternative formulation of an integral controller are highly technical but the derivations contain the appropriate level of detail and I was able (admittedly with some work and effort) to follow the arguments laid out in the manuscript and especially the now substantial supplementary material. I can understand that the references to additional mathematical work and proofs (e.g. in Refs 7 ad 15 in the supplementary material) does potentially add problems for a reader in reconstructing the full mathematical framework. In my view the authors negotiate the difficulties of exposing and explaining their proof in a clear and laudable manner. Developing a mathematical framework for a multidisciplinary

audience is challenging but the work presented here provides important pointers how we can achieve this and find common ground between the often opposing needs of mathematical and biological audiences.

Minor point:

Lines 438-442 contain in my reading one of the essential messages of this paper and I would suggest that the authors stress this and make the innovation in the paper more explicit.

In summary, this is exactly the type of mathematical biology that I and many others want to see.

Our Response:

We are indebted to Reviewer 2 for such an authoritative and thorough consideration of our work – both in terms of our admittedly complicated mathematical arguments and in terms of the broad conceptual contributions of our study – and for such generous suggestions to strengthen our exposition even further. We are truly grateful that the Reviewer has been willing to invest the time required to carefully review all our technical arguments to verify their mathematical soundness.

We have taken on board the Reviewer's helpful suggestion to further emphasize the content of Lines 438-442, in order to 'make the innovation in the paper more explicit'. (Lines 438-442 contain the text: "Until now strategies for identifying an internal model, and an associated integral, via a nonlinear coordinate change have only been applicable to exceedingly simple CRNs. By contrast, our approach identifies a well-defined nonlinear map between reaction rates of the model variables f_1, \dots, f_n , and a defining algebraic invariant, ρ (Eq. 6), which exists for all adaptation-capable CRNs"). With this suggestion in mind, we have now amplified the ideas contained in lines 438-442 of the previous version and more explicitly highlighted the innovation in our paper by adding in an extensive new paragraph at the end of the Introduction section (now lines 130-142 in this final version). We now explain-

"As we will show in the sections to follow, a mathematical transformation may always be applied to the reaction rates of any RPA-capable CRN to produce a special two-variable invariant called an 'RPA polynomial'. This distinguished algebraic invariant encodes the robust asymptotic tracking of a molecular setpoint, no matter how complex or intricate the intermolecular interactions, nor how vast the network of interacting molecules. Unlike the nonlinear coordinate transformations invoked in the 'standard' IMP, where a global coordinate transformation (unique to each RPA-capable network) is required to identify a single internal model, the nonlinear transformation we identify here has a special 'almost linear' structure (and, in special cases, exactly linear) and is decomposable into a topologically organised collection of linear integral controllers, each with its own independent internal model. In this way, we are able to identify the fundamental building blocks of all possible RPA-capable CRNs,

thereby revealing definitive and universal structural requirements that characterize all adaptation-capable collections of interacting molecules.”

This new material now leads directly to the detailed and careful analysis of our two simple running examples (which exemplify the properties of all RPA-capable CRNs), which lead, in turn, to a careful exposition of the underlying technicalities (comprehensively supported by our detailed Supplementary Information).